

**Reconstructing coupled time series in climate systems by machine learning**
Yu Huang[1], Lichao Yang[1], Zuntao Fu[1*]
[1]Lab for Climate and Ocean-Atmosphere Studies, Dept. of Atmospheric and Oceanic Sciences,
School of Physics, Peking University, Beijing, 100871, China
*Correspondence to:* Zuntao Fu (fuzt@pku.edu.cn)
**Abstract.**

7        Despite the great success of machine learning, its applications in climate dynamics have not been

well developed. One concern might be how well the trained neural networks could learn a dynamical
system and what can be the potential applications of this kind of learning. Detailed studies show that
the coupling relations or dynamics among variables in linear or nonlinear systems can be well learnt
by reservoir computer (RC) and long short-term memory (LSTM) machine learning, and these learnt
coupling relations can be further applied to reconstruct one series from the other dominated by
common coupling dynamics. In order to validate the above conclusions, toy models are applied to
address the following three questions: (i) what can be learnt from different dynamical time series by
machine learning; (ii) what factors significantly influence machine learning reconstruction; and (iii)
how to select suitable explanatory or input variables for the reconstructed variable for machine
learning. The results from these toy models show that both RC and LSTM can indeed learn coupling
relations among variables, and the learnt implicit coupling relation can be applied to accurately
reconstruct one series from the other. Both linear and nonlinear coupling relations between variables
can influence the quality of the reconstructed series. If there is a strong linear coupling between
variables, all of variables can be taken as explanatory variables for the reconstructed variable, and the



reconstruction can be bi-directional. However, when the linear coupling among variables is much
weaker, but with stronger nonlinear causality among variables, the reconstruction quality is direction-
dependent and it may be only uni-directional. We propose using convergent cross mapping causality
(CCM) index $\rho_{a \to b}$ to determine which variable can be taken as the reconstructed one and which can
be taken as the explanatory variable. For example, the Pearson correlation between the average
Tropical Surface Air Temperature (TSAT) and the average Northern Hemispheric SAT (NHSAT) is as
weak as 0.08, but the CCM index of NHSAT cross maps TSAT is $\rho_{N \to T} = 0.70$, it means that NHSAT
could be taken as the explanatory variable. Then we find that TSAT can be well reconstructed from
NHSAT by means of RC. However, the reconstruction quality in the opposite direction is poor, because
the CCM index of TSAT cross maps NHSAT is only $\rho_{T \to N} = 0.24$. These results also provide insights
on machine learning approaches for paleoclimate reconstruction, parameterization scheme, and
prediction in related climate studies.
**Key words:** Reconstruction, Climate time series, Machine learning, Causality, Reservoir computer,
Surface air temperature





**Highlights:**
i)  Learnt coupling dynamics between series by machine learning can be used to reconstruct series.
ii)  Reconstruction quality is direction- and variable-dependent for nonlinear systems.
iii) The CCM index is a potential indicator to choose reconstructed and explanatory variables.
iv) The tropical average SAT can be well reconstructed from the average Northern Hemispheric SAT.



## 1   Introduction

Making use of observed climatic time series is very important to solve climate problems, such as paleoclimate reconstruction (Brown, 1994; Emile-Geay and Tingley, 2016), interpolation for the missing points in measurements (Hofstra et al., 2008), parameterization schemes (Wilks, 2005; Vissio and Lucarini, 2018), and seasonal climate prediction (Comeau et al., 2017; Wang et al., 2017). Neural network-based machine learning, which recently attracts great attention in climate studies (Reichstein et al., 2019), is useful for these problems. First, machine learning has been widely applied to downscaling and data mining (Mattingly et al., 2016; Racah et al., 2017). Second, the machine learning frameworks for time series were used for predicting climate variables, such as temperature, humidity, runoff and air pollution (Zaytar and Amrani, 2016; Biancofiore et al., 2017; Kratzert et al., 2019; Feng et al., 2019). Recently it is demonstrated the large potentials for machine learning to simulate the temporal dynamics of complex systems (Pathak et al., 2017; Du et al., 2017; Watson, 2019), while the physics of systems is suggested for consideration. For example, the results of applying machine learning to Lorenz system (Lorenz, 1963) and Rossler models showed that their chaotic attractors were able to be well simulated (Pathak et al., 2017; Lu et al., 2018; Carroll, 2018). Chaos is the characteristic of climatic series (Lorenz, 1963; Patil et al., 2001), so that success in chaotic time series offers deep understanding for the application of machine learning.

Applying machine learning to climatic series attracts much attention, but it is still unclear what can be learnt by machine learning during the training process, and if there exist physical characteristics in climatic series, what is the key factor determining the performance of machine learning. This is crucial for investigating why machine learning performs not well with some datasets, and how to improve the performance for them. The cross-correlation of climatic time series might be first thought





of, because linear correlation is the implicit assumption for traditional statistical methods, and it is
known that they often fail if linear correlation is weak (Brown, 1994; Sugihara et al., 2012; Emile-
Geay and Tingley, 2016). However, previous studies (Sugihara et al., 2012; Emile-Geay and Tingley,
2016) suggest that a nonlinear correlation could be useful even though the linear correlation is weak.
For instance, the linear cross-correlations of sea surface temperature series observed in different
tropical areas are unstable and vary with time, which leads to an overall weak correlation, but this
nonlinear correlation could result in better El Niño predictions (Ludescher et al., 2014; Conti et al.,
2017). The phase plots of the ENSO/PDO index and some proxy variables are not linear lines but
nonlinear trajectories, which contributes greatly to reconstructing longer climate series (Mukhin et al.,
2018). These studies indicate that nonlinear relations can contribute to better analysis, reconstruction,
and prediction (Hsieh et al., 2006; Donner, 2012; Schurer et al., 2013; Badin et al., 2014; Drótos et
al., 2015; Van Nes et al., 2015; Comeau et al., 2017; Vannitsem and Ekelmans, 2018). Accordingly,
when applying machine learning to reconstruct climatic series, is it necessary to give attention to the
linear or nonlinear relations induced by the physical couplings? This is an open question.

To this question, we first discuss a real-world example from climate system. As we know, there

exists coupling in the atmospheric motions between the tropics and the Northern Hemisphere, which
is realized through the transfer of atmospheric energy (Farneti and Vallis, 2013). Due to the underlying
complicated processes, it is difficult to use a formula to cover this coupling between the tropical
average surface air temperature (TSAT) series and the Northern Hemispheric surface air temperature
(NHSAT) series. In the result part, it will be shown that the linear Pearson correlation between the
TSAT and NHSAT is very weak, but we can still use machine learning to well reconstruct the TSAT
series from the NHSAT series. However, the NHSAT series cannot be reconstructed from the TSAT



series. Such a contrasting result means that the machine learning approach is not the same as the

traditional statistical methods. Accordingly, is there any physical feature hided in the NHSAT-TSAT

coupling influencing this result? Furthermore, one might wonder if the similar phenomenon will be

universal for some other coupled climate series.

In this paper, to make progress towards understanding how machine learning approach is

influenced by the physical couplings of climatic series, we will investigate the machine learning

performances and their dependence on different coupling dynamics in climatic conceptual models. It

will be demonstrated that machine learning is able to "learn" these couplings, and the "learnt" coupling

relation are very useful for reconstructing climatic series. Moreover, our study also analyzes the

features of linear relations and nonlinear relations in climate systems, and a method to select

explanatory variables for machine learning is proposed. Hence, the underlying mechanism of the

abovementioned NHSAT-TSAT reconstruction will be also explained.

Our paper is organized as follows. In section 2, the methods about reconstructing time series and

data are introduced. In section 3, the ability of machine learning to "learn" different coupling relations

of climate system, and the potential application of machine learning to reconstruct climate series, are

shown. Then the performance dependence of machine learning on different factors is presented. The

underlying coupling strength is demonstrated to play a key role. In section 4, the application to real-

world climate series is shown. Summary is made in section 5.

**2    Methods and data**
**2.1    Leaning coupling relations and reconstructing coupled time series**

Firstly, we introduce our workflow for learning couplings of dynamical systems by machine


learning, and reconstructing the coupled time series. The total time series can be divided into two parts:
the training series (time lasting denoted as $t$) and the testing series (time lasting denoted as $t'$). For the
systems of toy models, the coupling relation or dynamics is stable and unchanged with time, i.e., there
is the stable coupling or dynamic relation $b(t) = F[a_1(t), a_2(t), ..., a_n(t)]$ among inputs
$a_1(t), a_2(t), ..., a_n(t)$ and output $b(t)$. If this inherent coupling relation can be "learnt" by machine
learning in the training series, the "learnt" coupling relation should be reflected by machine learning
in the testing series. Therefore, the workflow of our study can be summarized as follows (see Fig. 1):
(i) During the training period, $a_1(t), a_2(t), ..., a_n(t)$ and $b(t)$ are input into the machine learning
frameworks to learn the coupling or dynamic relation $b(t) = F[a_1(t), a_2(t), ..., a_n(t)]$. The "learnt"
coupling relation is denoted as $b(t) = \hat{F}[a_1(t), a_2(t), ... a_n(t)]$. Then it is tested whether this coupling
relation can be learnt by machine learning.
(ii) The second step is accomplished with the testing series to apply the learnt coupling relation $\hat{F}$
together with only $a_1(t'), a_2(t'), ..., a_n(t')$ to derive $b(t')$, denoted as $\hat{b}(t')$. $\hat{b}(t')$ is called "the
reconstructed $b(t')$" since only $a_1(t'), a_2(t'), ..., a_n(t')$ and the learnt coupling relation $\hat{F}$ have been
taken into account.
(iii) The first objective of this study is to answer whether the coupling relation
$b(t) = F[a_1(t), a_2(t), ..., a_n(t)]$ can be learnt by machine learning, i.e., whether the learnt coupling
relation $\hat{F}$ can well approximate the real coupling relation $F$. Since we do not intend to reach an
explicit formula of the learnt coupling relation $\hat{F}$, we will answer this question indirectly by
comparing the reconstructed series $\hat{b}(t')$ with the original series $b(t')$. If $\hat{b}(t') \approx b(t')$, then it can be
regarded as $\hat{F} \approx F$, and the machine learning framework can indeed learn the intrinsic coupling
relation among $a_1(t), a_2(t), ..., a_n(t)$ and $b(t)$.



(iv) If the machine learning framework can learn the intrinsic coupling relation between
$a_1(t), a_2(t),..., a_n(t)$ and $b(t)$, the learnt coupling relation $\hat{F}$ can be applied to reconstruct output
$b(t')$ even if only $a_1(t'), a_2(t'),..., a_n(t')$ are available.

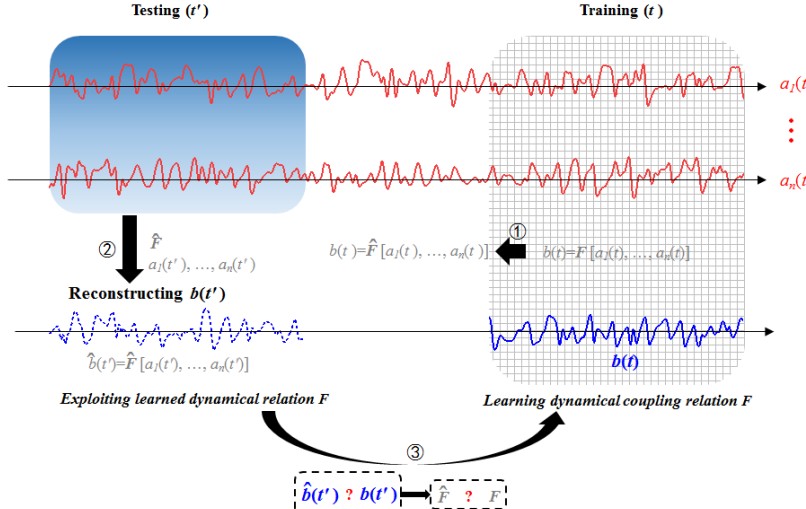


**Figure 1** Diagram illustration for reconstructing time series by machine learning. (1) The available part of the
dataset $\{a_1(t), …, a_n(t), b(t)\}$ is used to train the neural network ($a_1(t), …, a_n(t)$ and $b(t)$ are the time series of the
variables $a_1, …, a_n, b$). So that the inherent coupling relation $F$ among these variables can be learnt by the neural
network, and the learnt coupling relation is noted as $\hat{F}$. (2) $b(t')$ is unknown, but the dataset $\{a_1(t'), a_2(t'), …,$
$a_n(t')\}$ is available which is input into the trained neural network, and the unknown series $b(t')$ can be
reconstructed, denoted as $\hat{b}(t')$. (3) If $\hat{b}(t') \approx b(t')$, then $\hat{F} \approx F$ can be derived, and it indicates that the
machine learning framework have learnt the intrinsic coupling relation.

## 2.2   Machine learning frameworks

Inspired by several recent studies (As discussed in the introduction), we focus on employing the
commonly-used machine learning frameworks for time series to accomplish our study: reservoir





computer (RC), back propagation (BP), and long short-term memory (LSTM) neural networks. The
RC (Du et al., 2017; Lu et al., 2017; Pathak et al., 2018) has three layers: listening reservoir,
synchronization reservoir and prediction reservoir (see Fig. 2).

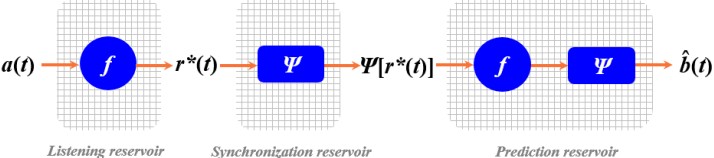


**Figure 2** Schematic of the RC neural network: the three parts of the RC neural network are the listening reservoir,
the synchronization reservoir and the prediction reservoir. A time series $a(t)$ is input into the RC neural network.
After the training processes of the three parts, the time series of $b$ variable can be reconstructed by machine learning,
denoted as $\hat{b}(t)$.

**The first layer is the listening reservoir.** A time series $a(t)$ (here for brevity, only one

dimensional series is considered) is first input into the listening reservoir. The function "$f$" governing
the listening reservoir training process is defined as follows:

$f: \quad r^*(t) = \tanh[M \cdot r(t) + W_{in} \cdot a(t) + E].$                          (1)

Here $r(t)$ represents the initial state of the listening reservoir, and $r^*(t)$ is the updated state of $r(t)$.
The listening reservoir has $N$ neurons, and these neurons are sparsely connected in the Erdös-Rènyi
matrix graph configuration (Lu et al., 2017). The connectivity of these neurons is represented by the
adjacency matrix $M$ of size $N{\times}N$, whose values are drawn from a uniform random distribution on the
interval [−1, 1]. The typical reservoir size used in this study is $N = 1000$. The other two components
of the listening reservoir are an input-to-reservoir weight matrix $W_{in}$, and a unit matrix $E$. The
elements of $W_{in}$ are randomly chosen from a uniform distribution in [−1, 1]. In the training process,
and $E$ is crucial for modulating the bias.





**The second layer is the synchronization reservoir.** After the training process of listening
reservoir, the updated state $r^*(t)$ is transported to a synchronization function "$\Psi$". The function is
as follows:
$$\Psi: \quad \Psi[r^*(t)] = W_{out} r^*(t). \tag{2}$$
When using the trained RC neural network make prediction or reconstruction, the role of Eq. (2) will
be making $\Psi[r^*(t)]$ close to the target series $b(t)$. This is a procedure to make $\Psi[r^*(t)]$ close to the
target variable $b(t)$. The reservoir-to-output matrix $W_{out}$ denotes a linear fitting process. Here it
should be noted that when using training data to train the RC neural network, $W_{out}$ needs to be trained
by the input data through Eq. (3) so that the elements of $W_{out}$ can be determined.
$$W_{out} = \arg\min_{W_{out}} \left\| W_{out} r^*(t) - b(t) \right\| + \alpha \left\| W_{out} \right\| \tag{3}$$
Here the symbol $\left\| \cdot \right\|$ is the $L_2$-norm of a vector ($L_2$ denotes the least square method) and $\alpha$ is the $L_2$
regularization (ridge regression) coefficient.
**The third layer is the prediction reservoir.** As Fig.2 shows, the prediction reservoir consists
of the function "$f$" and "$\Psi$". Actually this is a feedback loop to the former two layers. In this reservoir
layer, the input data is $\Psi[r^*(t)]$ rather than $a(t)$. Finally, the output of this layer is the reconstruction
for the target series $b(t)$, which is denoted as $\hat{b}(t)$.
Both BP and LSTM have widely used by many fields (Reichstein et al., 2019), and their
algorithms can be open access in https://ww2.mathworks.cn/help/deeplearning. The LSTM requires
to update its network every step, for both training and testing (Zaytar and Amrani, 2016; Kratzert et
al., 2019). Regarding this point, we stop the LSTM updates after training has been accomplished, and
this case is denoted as LSTM*.
**2.3   Evaluation of reconstruction quality**



To evaluate the quality of reconstruction by machine learning, the root mean squared error
(RMSE) of residual series (Hyndman and Koehler, 2006) is adopted, which represents the difference
between the original series $b(t')$ and the reconstructed series $\hat{b}(t')$. In order to fairly compare the
errors of reconstructing different processes (Pennekamp et al., 2018), we normalize the RMSE. The
RMSE and normalized RMSE (nRMSE) are defined as
$$RMSE = \sqrt{\frac{1}{k}\sum_{t}[b(t')-\hat{b}(t')]^2}, \qquad (4)$$
$$nRMSE = \frac{RMSE}{\max[b(t')]-\min[b(t')]}. \qquad (5)$$
In the results section, it can be observed that the good reconstruction quality corresponds to the
nRMSE value which is smaller than 0.1.

### 2.4    Data preparation

#### 2.4.1    The time series from conceptual climate models

**A linearly coupled model:** The autoregressive fractionally integrated moving average
(ARFIMA) model (Granger and Joyeux, 1980) maps a Gaussian white noise $\varepsilon(t)$ into a correlated
sequence $x(t)$ (Eq. (6)), which could simulate the linear dynamics of oceanic-atmospheric coupled
system (Hasselmann, 1976; Franzke, 2012; Massah and Kantz, 2016; Cox et al., 2018).
$$\varepsilon(t) \xrightarrow{ARFIMA(p,d,q)} x(t) \qquad (6)$$
The parameters $p$, $d$ and $q$ denote the order $p$ of the autoregressive process (AR $(p)$ process), the Hurst
exponent $H=d+0.5$ of long-term memory process, and the order $q$ of the moving average process (MA
$(q)$ process).The coefficients in this model are set as follows: (1) $p=3$ (we set the AR(3) coefficients
as 0.6, 0.2 and 0.1 respectively); (2) $d=0.2$ (the Hurst exponent is 0.7); (3) $q=3$ (we set the MA(3)



coefficients as 0.3, 0.2 and 0.1 respectively). Finally $x(t)$ is a complex time series which is composited
from all of the above three processes.
**A nonlinearly coupled model:** The Lorenz 63 (in the following referred as to Lorenz63) chaotic
system (Lorenz, 1963) depicts the nonlinear coupling relation in a low-dimensional chaotic system.
The system reads
$$\frac{dx}{dt} = -\sigma(x-y)$$
$$\frac{dy}{dt} = \mu x - xz - y \tag{7}$$
$$\frac{dz}{dt} = xy - Bz$$

When the parameters are fixed at $(\sigma, \mu, B) = (10, 28, 8/3)$, the state in the system is chaotic. We
employed the Runge-Kutta integral of the fourth order to acquire the series output from Lorenz63.
The time steps were 0.01.
**A high-dimensional model:** The two-layer Lorenz 96 (in the following referred as to Lorenz96)
model (Lorenz, 1996) is a high-dimensional chaotic system, which is generally employed to mimic
mid-latitude atmospheric dynamics (Chorin and Lu, 2015; Hu and Franzke, 2017; Vissio and Lucarini,
2018; Chen and Kalnay, 2019; Watson, 2019). It reads
$$\frac{dX_k}{dt} = X_{k-1}(X_{k+1} - X_{k-2}) - X_k + F - \frac{h_1}{J}\sum_{j=1}^{J} Y_{j,k}$$
$$\frac{dY_{k,j}}{dt} = \frac{1}{\theta}[Y_{k,j+1}(Y_{k,j-1} - Y_{k,j+2}) - Y_{k,j} + h_2 X_k]. \tag{8}$$

In the first layer of the Lorenz 96 there are 18 variables marked as $X_k$ ($k$ is any integer of interval [1,
18]), and each $X_k$ is coupled with $Y_{k,j}$ ($Y_{k,j}$ is from the second layer). The parameters are set as fellows:
$J = 20$, $h_1 = 1$, $h_2 = 1$, and $F=10$. The scale parameter $\theta$ controls the scale separation of the two layers.
When $\theta > 1$, processes in the second layer will be slower than processes in the first layer because





the increment of $Y_{k,j}$ is decreased by the term of $\theta$. The time scale of $Y_{k,j}$ can be also close to that of
$X_k$ by modulating the value of $\theta$; especially, the coupling strength will be amplified when $\theta$ is much
smaller than 1. The Runge-Kutta integral of the fourth order and periodic boundary condition are
adopted (that is: $X_0 = X_K$ and $X_{K+1} = X_1$ ; $Y_{k,0} = Y_{k-1,J}$ and $Y_{k,J+1} = Y_{k+1,1}$), and the integral time unit was
taken as 0.05.

### 2.4.2   Real-world climatic series

TSAT, NHSAT and the Nino3.4 index are chosen to represent real-world climatic time series.
The original data is from National Centers for Environmental Prediction (NCEP,
https://www.esrl.noaa.gov/psd/data/gridded/data.ncep.reanalysis2.html) and KNMI Climate Explorer
(http://climexp.knmi.nl/selectdailyindex.cgi?id=someone@somewhere). The series of TSAT and
NHSAT are from the regional average of gridded daily data in NCEP Reanalysis 2. The selected
spatial range is $20^0$N–$20^0$S for the tropics and $20^0$N–$90^0$N for the Northern Hemisphere. The selected
temporal range is from 1981/09/01 to 2018/12/31.
**Training and testing the datasets:** Before analysis, all the time series are standardized to take
zero mean and unit standard deviation. We divide the total series into two parts: 60% of the time series
training the neural network and 40% being the testing series. Specific data lengths of the training
series and testing series will be also listed in the results section.

## 3   Results

## 3.1   Coupling relation learning

### 3.1.1   Linear coupling relation and machine learning



First of all, we consider the simplest case: the linear coupling relation between two variables.
**Quantifying the linear coupling strength:** To quantify the linear correlation between two series
$a(t)$ and $b(t)$, the Pearson correlation coefficient is adopted and is defined as
$$corr. = \frac{mean[(a-\bar{a})\cdot(b-\bar{b})]}{std(a)\cdot std(b)}. \tag{9}$$
In Eq. (9), the symbols "*mean*" and "*std*" denote the average and standard deviation for series $a(t)$ and
$b(t)$, respectively.
Here, the ARFIMA (3, 0.2, 3) model is taken as an example for "learning" strong linear coupling
between two variables. Obviously, there are different temporal structures in $x(t)$ and $\varepsilon(t)$, especially
for their large-scale trends (Fig. 3a) and power spectra (Fig. 3b). The marked difference between $x(t)$
and $\varepsilon(t)$ is in their low-frequency variations, and there are more low-frequency and larger-scale
structures in $x(t)$ than in $\varepsilon(t)$. We employ machine learning (RC, LSTM, LSTM*, and BP) to learn the
dynamics of this model (Eq. (6)) by the procedure shown in Fig. 1. The training parts of $\varepsilon(t)$ are
selected from the gray shadow in Fig. 3a. RC, LSTM, LSTM*, and BP are used to learn the coupling
relation between $x(t)$ and $\varepsilon(t)$. Then, the learnt coupling together with $\varepsilon(t')$ is used to reconstruct $x(t')$.
The reconstruction results and the performance of different machine learning frameworks are listed in
Table 1. It shows that there is a strong linear correlation (0.88) between $x(t')$ and $\varepsilon(t')$. This
reconstruction result indicates that strong linear coupling can be well captured by all kinds of machine
learning frameworks since the values of nRMSE are low for all of the machine learning frameworks.



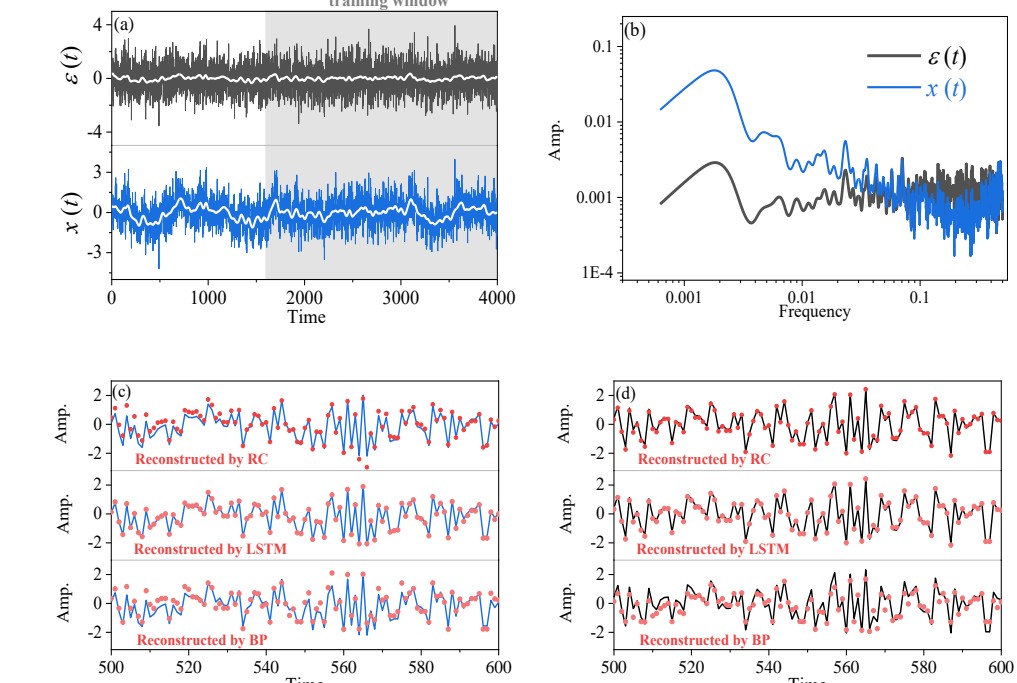



**Figure 3** (a) The $x(t)$ time series (blue) and the $\varepsilon(t)$ time series (black) of the ARFIMA(3,0.2,3) model. White lines

depict the large-scale trends of these time series acquired by 50-step smoothing average. (b) Comparison of the

power spectrum of $x(t)$ (blue) with the power spectrum of $\varepsilon(t)$ (black). (c) Comparison of the reconstructed time

series of $x(t)$ by RC, LSTM and BP respectively (red dots), and the original $x(t)$ time series are presented by the blue

lines. (d) Comparison of the reconstructed time series of $\varepsilon(t)$ by RC, LSTM and BP respectively (red dots), and the

original $\varepsilon(t)$ time series are presented by the black lines. Only partial segments of the reconstructed series are shown.







Detailed comparisons between the original and reconstructed series are shown in Fig. 3c. When

inputting $\varepsilon(t')$, the learnt linear coupling can be applied to accurately reconstruct the original $x(t')$ by
the RC or LSTM. When $x(t')$ is reconstructed from $\varepsilon(t')$ by LSTM, the minimum value of nRMSE
(0.01) is reached; all of the reconstructed data are nearly overlapped with the original ones and cannot
be visually differentiated (see Fig. 3c). When reconstructing $x(t')$ from $\varepsilon(t')$ by the RC, the





reconstruction quality is also well. The best performance of LSTM among the given machine learning
frameworks benefits from its network updating all the time. When updating is stopped, then the
reconstruction error of LSTM* is no longer better than that of the RC, and is just a little bit better than
that of BP (see Table 1). Therefore, if the training of machine learning frameworks is only carried out
on the training series and there is no further updating, the RC is the best machine learning framework.

**Table 1** Details of reconstructing ARFIMA (3, 0.2, 3)

| Input (a) | Output (b) | corr. | Data length (training/testing) | Neural network | RMSE | nRMSE |
|-----------|------------|-------|--------------------------------|----------------|------|-------|
| $\varepsilon(t')$ | $x(t')$ | 0.88 | 2400/1600 | RC | 0.31 | 0.04 |
|  |  |  |  | LSTM | 0.07 | 0.01 |
|  |  |  |  | LSTM* | 0.46 | 0.06 |
|  |  |  |  | BP | 0.52 | 0.07 |
| $x(t')$ | $\varepsilon(t')$ | 0.88 | 2400/1600 | RC | 0.09 | 0.01 |
|  |  |  |  | LSTM | 0.08 | 0.01 |
|  |  |  |  | LSTM* | 0.45 | 0.06 |
|  |  |  |  | BP | 0.50 | 0.07 |

### 280    3.1.2    Nonlinear coupling relation and machine learning

From the above results, it is known that a strong linear correlation is of great importance for
series reconstruction and coupling learning. For a nonlinear coupled system, it has been reported that
the coupling strength between two variables cannot be quantified by linear correlation (Brown, 1994;
Sugihara et al., 2012). Hence, the convergent cross mapping (CCM, Sugihara et al., 2012) method is
adopted.
**Quantification of the nonlinear coupling strength:** CCM index can quantify the coupling or
causality between two variables (Sugihara et al., 2012; Tsonis et al., 2018; Zhang et al., 2019). When
two series are output from the same dynamical system or coupled to each other, there will be shared
information in their dynamical phase spaces (Takens, 1981). The stronger the coupling or forcing, the



more information they share with each other. The shared information can be quantified by CCM index
$\rho$. For two time series $a(t)$ and $b(t)$, there are two CCM indices (Tsonis et al., 2018): $\rho_{a \to b}$ and $\rho_{b \to a}$:
(i) If variable $b$ does influence variable $a$, the information of $b(t)$ will be stored in $a(t)$, and thus we
will acquire a high value of cross-mapping skill $\rho_{a \to b}$ ; (ii) if variable $a$ does influence variable $b$, the
information of $a(t)$ will be stored in $b(t)$, and thus we will acquire a high value of cross-mapping skill
$\rho_{b \to a}$ . The implementation of that $a(t)$ cross maps $b(t)$ is based on a nonlinear operator known as the
empirical dynamics model (Sugihara, 1994), and detailed algorithm for CCM can be found in a
previously reported study (Tsonis et al., 2018).
When the linear correlation between variables is very weak, could machine learning frameworks
still be applied to learn the underlying coupling dynamics? To address this question, we turn to the
Lorenz 63 system (Lorenz, 1963). There is a very weak linear correlation between variables $X$ and $Z$
(with a Pearson correlation coefficient of 0.002) in the Lorenz63 model (see Table 2), and such a weak
linear correlation is induced by the unstable local correlation between variables $X$ and $Z$ (see Fig. 4a):
For example, $X$ and $Z$ are negatively correlated in the interval of 0–200, but positively correlated in
200–400. This alternation of negative and positive correlation appears over the whole processes of $X$
and $Z$, which leads to an overall weak linear correlation. In this case, it cannot obtain a feasible linear
regression model between $X$ and $Z$ to reconstruct one from the other, since there is no such good linear
dependency as found in the ARFIMA (p, d, q) model (see Figs. 5a and 5b). However, there is indeed
a well-defined coupling relation between $X$ and $Z$ in the Lorenz63 system (with a high CCM index
$\rho_{X \to Z} = 0.91$), so that it can be possible to reconstruct $Z$ from $X$ by some tools.

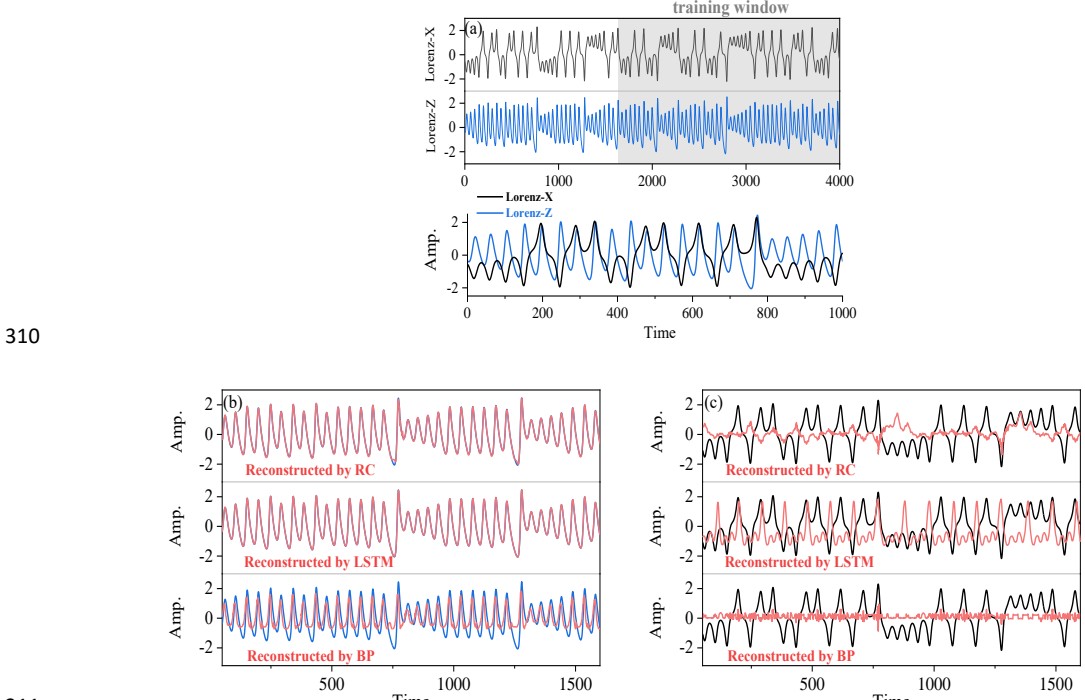



**Figure 4** (a) The $X$ time series (black) and the $Z$ time series (blue) of the Lorenz 63 model. (b) Comparison of the

reconstructed time series of $Z$ by RC, LSTM and BP respectively (red lines), and the original $Z$ time series are

presented by the blue lines. (c) Comparison of the reconstructed time series of $X$ by RC, LSTM and BP respectively

(red lines), and the original $X$ time series are presented by the black lines.

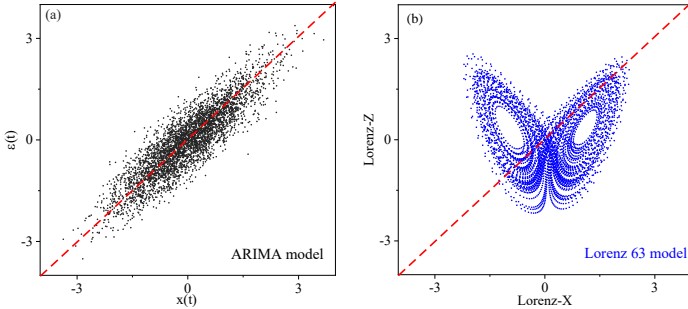


**Figure 5** (a) Scatter plot of $x(t)$ versus $\varepsilon(t)$ from ARFIMA(3,0.2,3) model (black dots). (b) Scatter plot of $X$ time



series and *Z* time series of the Lorenz 63 model (blue dots).

**Table 2** Details of Lorenz63 system reconstruction

| Input (a) | Output (b) | *corr.* | $\rho_{a \to b}$ | Data length (training/testing) | Neural network | RMSE | nRMSE |
|---|---|---|---|---|---|---|---|
| Lorenz -X | Lorenz-Z | **0.002** | 0.91 | 2400/1600 | **RC** | **0.04** | **0.008** |
|  |  |  |  |  | LSTM | 0.02 | 0.004 |
|  |  |  |  |  | LSTM* | 1.02 | 0.24 |
|  |  |  |  |  | BP | 0.77 | 0.17 |
| Lorenz -Z | Lorenz-X | **0.002** | **0.03** | 2400/1600 | **RC** | **1.13** | **0.34** |
|  |  |  |  |  | LSTM | 1.03 | 0.31 |
|  |  |  |  |  | LSTM* | 1.08 | 0.33 |
|  |  |  |  |  | BP | 1.01 | 0.31 |

Different from the case of linear system with strong linear coupling, the reconstruction for the
time series of the Lorenz63 system depends on the used machine learning frameworks (Fig. 4b). The
series reconstructed by LSTM nearly overlaps with the original series, and has the minimum nRMSE
(0.004); moreover, the RC performs quite well, with only a little difference found at some peaks and
dips. These reconstruction results indicate that, even though the linear correlation is very weak, the
strong nonlinear correlation will allow RC and LSTM to fully capture the underlying coupling
dynamics. However, BP and LSTM* perform poorly, and their reconstruction results have large errors
(nRMSE = 0.17 for BP, and nRMSE = 0.24 for LSTM*). The reconstructed series heavily departs
from the original series, especially for all peaks and dips, and the reconstructed values for each
extreme point are underestimated (Fig. 4b). This means that both of BP and LSTM* cannot learn the
nonlinear coupling.
**3.2    Reconstruction quality and impact factors**
From the above results, it is found that machine learning framework is able to learn different





(linear or nonlinear) coupling relations well. Just as mentioned in the introduction section, under the
condition that machine learning framework can learn the coupling relation between or among
variables within the same system, the learnt coupling relation can be applied to reconstruct one
variable's time series from others (see Fig.1). We further investigate the ability of such time series
reconstruction, including what factors might influence the reconstruction quality.

### 3.2.1   Direction dependence and variable dependence

When reconstructing the time series of linear model of Eq. (6), it can be found that the
reconstruction is invertible (see Fig. 3d and Table 1): one variable can be taken as explanatory variable
to reconstruct another variable's time series well; oppositely, it can be also well reconstructed by
another variable. In fact, when there is a strong linear correlation between variables, the invertible (or
bi-directional) construction can also be accomplished by using a traditional regression approach
(Brown, 1994). Hence, when the linear coupling is weak but the nonlinear coupling is strong, will the
bi-directional reconstruction still be allowed? The answer is usually no. For example, when comparing
the reconstruction quality of reconstructing variable $Z$ from variable $X$ (Fig. 4b) with that of
reconstructing variable $X$ from variable $Z$ (Fig. 4c), it is obvious that all of the machine learning
frameworks fail (large values of nRMSE are all close to 0.3) for reconstructing $X$ from $Z$. The
reconstruction direction is no longer invertible in this nonlinear system, where the reconstruction
quality is direction-dependent and variable-dependent.
Hence, we further discuss how to select the suitable explanatory variable or reconstruction
direction. Tables 1 and 2 show that the reconstruction quality in a linear coupled system highly
depends on the Pearson correlation, however it might be different for a nonlinear system. For the



Lorenz 63 system, the two-direction CCM coefficients between the variable $X$ and variable $Z$ is
asymmetric (with a stronger $\rho_{X \to Z} = 0.91$ and weaker $\rho_{Z \to X} = 0.03$), and then variable $Z$ can be well
reconstructed from variable $X$ by machine learning but variable $X$ cannot be reconstructed from
variable $Z$ (Fig. 4b and 4c). CCM index can be taken as a potential precursor to determine the
explanatory variable and reconstructed variable for this nonlinear system. Actually, for most of
nonlinearly coupled variables (Here called $a$ and $b$), the two-direction causal forces might be
asymmetric, and the values of $\rho_{a \to b}$ and $\rho_{b \to a}$ are often unequal to each other (Tosnis et al., 2019).
Here the asymmetric reconstruction quality is just caused by the asymmetric causality between
variables.
**The cases of high-dimensional system:** It is noted that the above conclusions and results are
derived from the low-dimensional chaotic system (Eq. (7)). It is also interesting to know whether they
are also applicable to a high-dimensional chaotic system of Lorenz 96 model. Since BP and LSTM*
fully fail in reconstructing for Lorenz63 system (Not shown here), which suggests that they are not
suitable for handling nonlinear systems, so the results from LSTM and RC will be shown next.

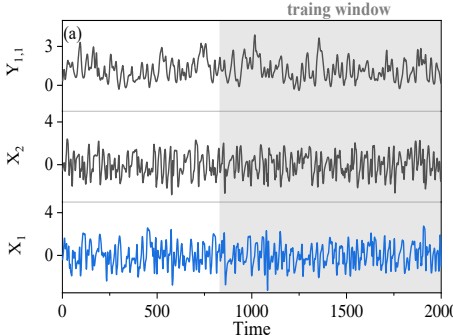



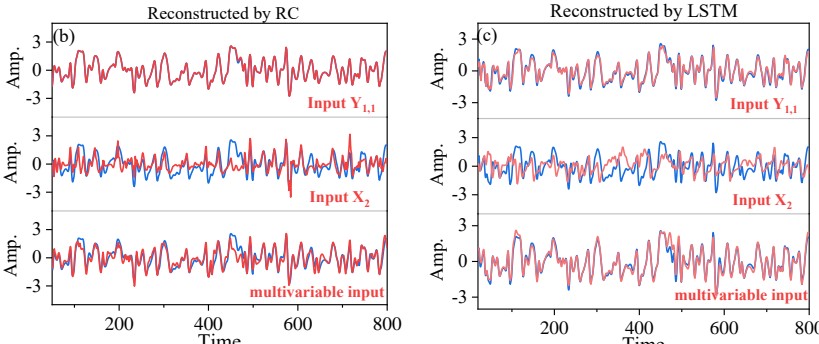


**Figure 6** (a) The $Y_{1,1}$ time series(black), $X_2$ time series (black) and $X_1$ time series(blue)of the Lorenz 96 model. (b)
By means of the RC machine learning, when using $Y_{1,1}$, $X_2$ and multivariate to be the explanatory variable
respectively, the corresponding reconstructed $X_1$ time series are showed respectively from the top panel to the bottom
panel (red lines), and the original $X$ time series are presented by the blue lines. (d) By means of the LSTM machine
learning, when using $Y_{1,1}$, $X_2$ and multivariate to be the explanatory variable respectively, the corresponding
reconstructed $X_1$ time series are showed respectively from the top panel to the bottom panel (red lines), and the
original $X$ time series are presented by the blue lines.

Firstly, we use variables $X_1$ and $Y_{1,1}$ in Eq.(8) to illustrate the direction dependence in the high-

dimensional system. Details of $X_1$ and $Y_{1,1}$ are shown in Fig. 6a, and the Pearson correlation between
$X_1$ and $Y_{1,1}$ is weak (only -0.11, see Table 3). For the given value of each parameter in Eq. (8), the
forcing from $X_1$ to $Y_{1,1}$, is much stronger than the forcing from $Y_{1,1}$ to $X_1$. Hence, CCM index shows
the asymmetric causality: $\rho_{Y_{1,1} \to X_1} = 0.98$ and $\rho_{X_1 \to Y_{1,1}} = 0.61$. It means that reconstructing $X_1$ from $Y_{1,1}$
might obtain a better quality than the opposite direction. Hence, by means of RC, the error of
reconstructing $X_1$ from $Y_{1,1}$ is nRMSE=0.01, and in the opposite direction it is nRMSE=0.06 (Table
3), which is consistent with the indication of CCM index (Similar results can be found in Figs. 6b and
6c).



**Table 3** Details of reconstructing the Lorenz 96 model

| Input (a) | Target (b) | *corr.* | $\rho_{a \to b}$ | Data length (training/testing) | Neural network | RMSE | nRMSE |
|---|---|---|---|---|---|---|---|
| $Y_{1,1}$ | $X_1$ | -0.11 | 0.98 | 1200/800 | RC | 0.03 | 0.01 |
| | | | | | LSTM | 0.34 | 0.05 |
| $X_1$ | $Y_{1,1}$ | **-0.11** | **0.61** | **1200/800** | **RC** | **0.35** | **0.06** |
| | | | | | **LSTM** | **0.42** | **0.08** |
| $X_2$ | $X_1$ | -0.06 | 0.37 | 1200/800 | RC | 0.69 | 0.13 |
| | | | | | LSTM | 1.09 | 0.20 |
| $X_1$ | $X_2$ | **-0.06** | **0.25** | **1200/800** | **RC** | **0.95** | **0.17** |
| | | | | | **LSTM** | **0.84** | **0.16** |
| $X_2, X_{17}, X_{18}$ | $X_1$ | -0.06, -0.24, 0.06 | 0.37, 0.29, 0.41 | 1200/800 | RC | 0.41 | 0.08 |
| | | | | | LSTM | 0.32 | 0.06 |

The reconstruction between $X_1$ and $X_2$ in the same layer of Lorenz96 is also shown. There is an
asymmetric causal relation ($\rho_{X_2 \to X_1} = 0.37$ and $\rho_{X_1 \to X_2} = 0.25$) between $X_1$ and $X_2$, and their linear
correlation is very weak (see Table 3). The RC gives better result of reconstructing $X_1$ from $X_2$
(nRMSE=0.13) than reconstructing $X_2$ from $X_1$ (nRMSE=0.17). LSTM also has different results for
$X_1$ and $X_2$ (Table 3), where the quality of reconstructing from $X_1$ to $X_2$ (nRMSE=0.16) is better than
reconstructing from $X_2$ to $X_1$ (nRMSE=0.20). The reconstruction quality of LSTM is worse than the
RC, and the reconstruction results by LSTM are not consistent with the coupling strengths. This might
indicate that LSTM will perform worse in some cases than RC, and the best choice of machine learning
framework in data reconstruction is RC.
Secondly, it is also found that the reconstruction quality is influenced by the chosen explanatory
variables: The quality of reconstructing $X_1$ from $Y_{1,1}$ is better than the quality of reconstructing $X_1$ from
$X_2$ by (see Fig. 6b and 6c). Then, if more than one explanatory variable in the same layer are considered,
the reconstruction of $X_1$ from $X_2$ can be greatly improved (see Figs. 6b and 6c). For example, when all
of $X_2$, $X_{17}$ and $X_{18}$ are regarded as the explanatory variables, the nRMSE of reconstructed $X_1$ is reduced
from 0.13 to 0.08 (Table 3). For both of RC and LSTM machine learning, the multivariable





reconstruction reaches lower error than those from unit-variable reconstruction.

### 3.2.2     Explanation of the reconstruction quality dependency: Role of coupling strength

From Table 1 to Table 3, it is obvious that the value of nRMSE varies with the Pearson correlation
or CCM index. Furthermore, the lower nRMSE often corresponds to the higher correlation or CCM
index. Compared to RC and LSTM, it is revealed that BP and LSTM* are much more sensitive to the
Pearson correlation; meanwhile, the RC is sensitive to CCM index. In the following, we demonstrate
that the above reconstruction quality dependencies are determined by the coupling strength of
variables.
The setting of Eq. (8) is as follows: the value of $h_1$ is set as 0, and the value of $\theta$ is decreased
from 0.7 to 0.3. When $\theta$ is equal to 0.7, the forcing from $X_1$ to $Y_{1,1}$ is weak. At that time, the Pearson
correlation between $X_1$ and $Y_{1,1}$ is only 0.48, and the performances of BP and LSTM* are not good.
When $\theta$ is equal to 0.3, the forcing is dramatically magnified. As the second panel of Fig. 7a shows,
this strong forcing makes $Y_{j,i}$ synchronized to $X_i$, and the linear correlation between $X_1$ and $Y_{1,1}$ is
greatly increased to 0.8. When the forcing strength is magnified, the performance of machine learning
is also enhanced (Fig. 7b): red dots (BP output) and the blue line (LSTM* output) are much closer to
the black line (original target series). This means, that the reconstruction quality of BP and LSTM* is
sensitive to this linear coupling variation, and it is greatly improved when the linear correlation is
increased.




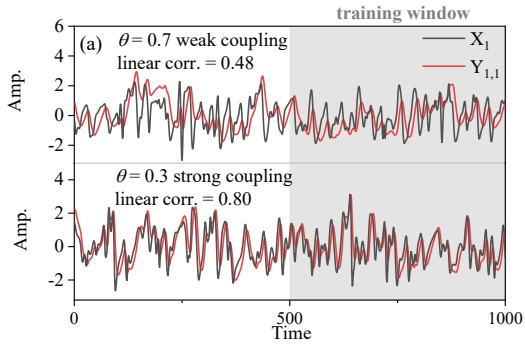


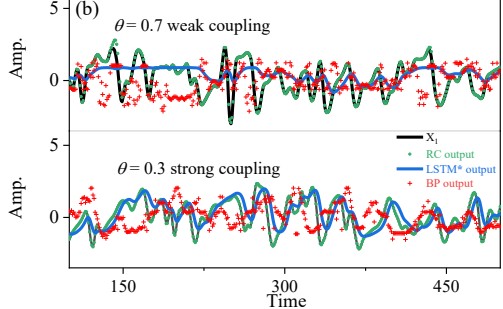

**Figure 7** Influence of strong coupling on series' linear correlation and machine learning performances. (a)
Comparison of the linear correlation when the coupling strength is different. The top panel corresponds to the weak
coupling strength, and the bottom panel corresponds to the strong coupling. The red lines present the input
explanatory variable and the black lines present the target series of machine learning. (b) Comparison of the machine
learning performances when the coupling strength is different. The top panel corresponds to the weak coupling
strength, and the bottom panel corresponds to the strong coupling. The black lines are the original series; the
reconstructed series by RC (green lines), LSTM*(blue lines) and BP (red dots) are shown respectively.

However, the RC is not so much restrained by the Pearson correlation. When $\theta$ is equal to 0.7

or 0.3, the values of CCM index are both higher than 0.9, that is to say, the nonlinear coupling strength
is not changed by $\theta$. Then, it can be found that the quality of reconstructed $X_1$ by RC is always good.
As Fig. 7b shows, the green dots (RC output) in Fig. 7b always overlap with the black line (original





target series). Actually, the reconstruction quality of the RC is determined more by the nonlinear
coupling strength. The values of CCM index are calculated between $X_1$ and $X_2$, $X_3$ …, $X_{18}$; meanwhile,
$X_1$ is reconstructed from $X_2$, $X_3$ …, $X_{18}$, respectively. Then, a significant correspondence exists
between the nRMSE and CCM index (Fig. 8), especially for the results of RC. This indicates that the
reconstruction quality is dependent on the coupling strength between the reconstructed variable and
different explanatory variables.

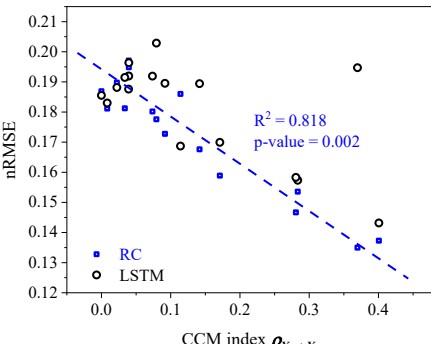


**Figure 8** Scatter plot of nRMSE values and CCM index values. The blue boxes are results of the RC machine
learning, and the black cycles are results of the LSTM machine learning. The blue dashed line is the fitted linear
trend of the blue cycles, and this dependency trend is significant because the p-value is much smaller than 0.05.

## 3.3    Application to real-world climate series: reconstructing SAT

The natural climate series are usually nonstationary, and are encoded with the information of

many physical processes in the earth system. In the following, we illustrate the utility of the above
methods and conclusions by investigating a real-world example mentioned in the introduction, by
showing that: whether the NHSAT time series can be reconstructed from the TSAT time series, and
whether the TSAT time series can be also reconstructed from the NHSAT time series.

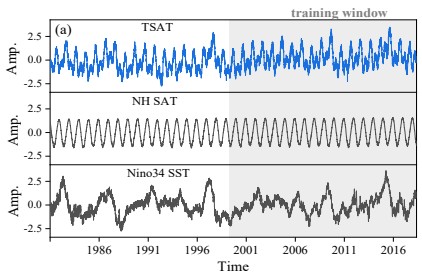


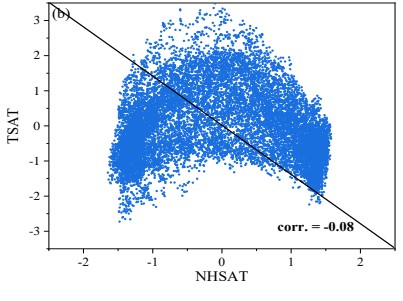


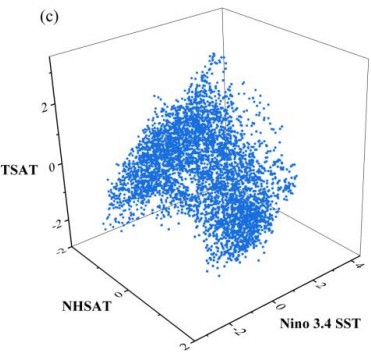

**Figure 9** (a) Daily time series of TSAT, NHSAT and Nino 3.4 index. (b) Scatter plot of normalized NHSAT and
normalized TSAT. (c) Three-dimensional scatter plot of normalized NHSAT, normalized TSAT and normalized Nino
3.4 SST.

The daily NHSAT and TSAT time series are shown in Fig. 9a. It is quite different for the

oscillation shapes of the NHSAT and TSAT series, and there is a weak linear correlation (0.08, see
Table 4) between them. In the scatter plot for the NHSAT and TSAT (Fig.9b), the marked nonlinear
structure is observed between NHSAT and TSAT. Such a weak linear correlation will make the linear





regression model fail to reconstruct one series from the other. Likewise, there is no explicit physical
expression that can transform TSAT and NHSAT to each other. Now we try to using machine learning
to reconstruct these climate series. CCM index of that NHSAT cross maps TSAT is $\rho_{N \to T} = 0.70$, and
CCM index of that TSAT cross maps NHSAT is $\rho_{T \to N} = 0.24$ (Table 4). CCM index indicates that
the reconstruction from NHSAT to TSAT will obtain a better quality than the opposite direction.
**Table 4** Details of temperature records' reconstruction

| Input (a) | Output (b) | *corr.* | $\rho_{a \to b}$ | Data length (training/testing) | Neural network | RMSE | nRMSE |
|---|---|---|---|---|---|---|---|
| NHSAT | TSAT | **0.08** | 0.70 | 8182/5454 | RC | **0.73** | **0.13** |
|  |  |  |  |  | LSTM | 1.14 | 0.20 |
|  |  |  |  |  | BP | 1.45 | 0.26 |
| TSAT | NHTSAT | **0.08** | **0.24** | 8182/5454 | RC | **0.97** | **0.21** |
|  |  |  |  |  | LSTM | 1.04 | 0.23 |
|  |  |  |  |  | BP | 1.23 | 0.37 |

By means of RC machine learning, TSAT can be described by the reconstructed time series
(Fig.10a). But the corresponding nRMSE is equal to 0.13, this is because some extremes of the TSAT
time series have not been described (Fig.10b). When using TSAT to reconstruct the time series of
NHSAT, the reconstructed time series cannot describe the original time series of NHSAT (Fig.10c),
and the corresponding nRMSE is equal to 0.21. Besides, we also use the LSTM and BP to reconstruct
these natural climate series, the performances of these two neural networks are worse than RC (Table
4). For BP, this might be due to its inability to deal with nonlinear coupling. As for LSTM, the unstable
variance might influence its performance because it is heavily impacted by the unstable variance.



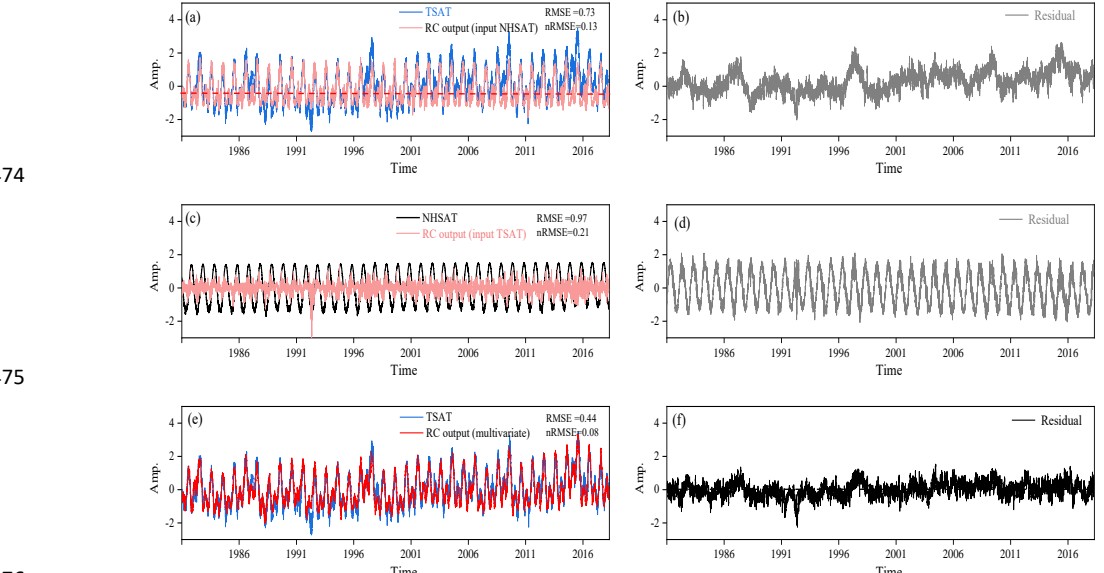

**Figure 10** (a) Reconstructed TSAT time series (red) when NHSAT is the explanatory variable; (b) Residual series

given by the original TSAT series and the reconstructed TSAT series of (a). (c) Reconstructed NHSAT time series

(red) when TSAT is the explanatory variable. (d) Residual series given by the original NHAST series and the

reconstructed NHSAT series of (c). (e) Reconstructed TSTA time series (red) when NHSAT and Nino3.4 index are

the explanatory variables. (f) Residual series given by the original TSAT series and the reconstructed TSAT series of

(e).

We can also improve the reconstruction quality of TSAT. Considering that the tropics climate

system do not only interact with the Northern Hemisphere climate system, we can use the information

of other subsystems to improve the reconstruction. Looking at the time series of Nino 3.4 index (Fig.

9), some of its extremes occur in the same time regions as the extremes of TSAT. Moreover, when

Nino3.4 is included into the scatter plot (Fig. 9c), a nonlinear attractor structure is revealed. We

combine NHSAT with Nino 3.4 index to reconstruct the time series of TSAT, and the reconstruction

is by means of the RC machine learning. The reconstructed TSAT (Fig. 10e) is much closer to the



original TSAT series, and the corresponding nRMSE has been improved to 0.08.

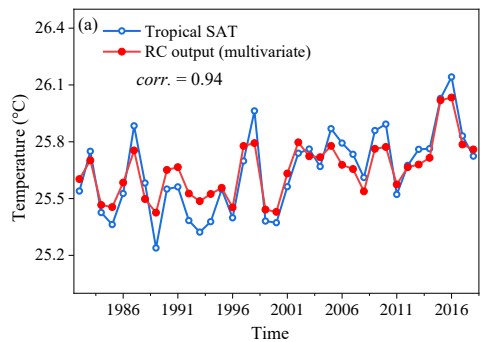

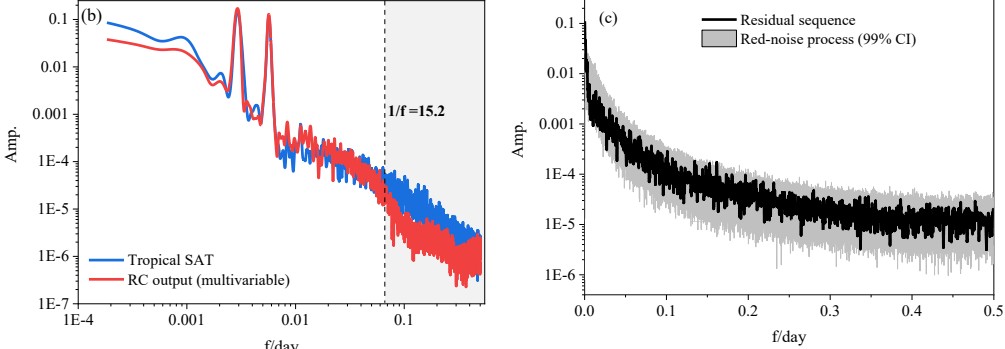

**Figure 11** (a) Comparison between the annual mean values of reconstructed TSAT (blue) and the annual mean values

of original TSAT (red). (b) Comparison between the power spectrum of reconstructed TSAT (blue) and the power

spectrum of original TSAT (red). (c) Red-noise test for residual series, the gray shaded area is the 99% CI of red-

noise process.

Finally, we further compare the original TSAT and the reconstructed TSAT: (i) the annual

variations of TSAT and the reconstructed TSAT are close to each other (Fig. 11a). (ii) The power
spectrum of TSAT and the reconstructed TSAT are compared in Fig. 11b, and it can be seen that the
main deviation is in the frequency bands corresponding to around 0-15 days. We guess the reason





might be the local weather processes are not input into this RC reconstruction. This conjecture can be
further confirmed by red-noise test with response time 15 days for the residual series (here the red-
noise test is the same as the method used in Roe, 2009). The gray shadow in Fig. 11c gives the 99%
confidence interval of red-noise process, and the black line is the residual series. All data points of the
residual series lie within the confidence intervals, and this means, the residual is possibly induced by
local weather processes that is not input into the RC.
**4    Conclusions and discussions**

In this study, three kinds of machine learning frameworks is used to reconstruct the time series

of toy models and real-world climate systems. One series can be reconstructed from the other series
by machine learning when they are governed by the common coupling relation. For the linear system,
variables are coupled by the linear mechanism, and a strong Pearson correlation benefits to machine
learning with bi-directional reconstruction. For a nonlinear system, the time series often have a weak
Pearson correlation, but the machine learning can still well reconstruct the time series when two
variables share the common information through their interactions; moreover, the reconstruction
quality is direction-dependent and variable-dependent, which is determined by the coupling strength
and causality between the dynamical variables.

Considering the reconstruction quality dependency, selecting the suitable explanatory variables

is crucial for obtaining a good reconstruction quality. Hence, we propose using CCM index to select
explanatory variables. Especially for the time series of nonlinear systems, when the Pearson
correlation is weak, CCM index might be strong enough, and then the corresponding variable can be
selected as an explanatory variable. It is well known that atmospheric or oceanic motions are





nonlinearly coupled over most of scales, and therefore, in the natural climate series, there would be
similar nonlinear correlations to the Lorenz 63 and Lorenz 96 system (the linear correlation is weak
but CCM indices are of high values). However, if only Pearson correlation is used to select the
explanatory variable, some useful nonlinearly correlated variables might be left out.

Besides, it is very important to select the correct direction for reconstructing one variable from

other variable within a nonlinear system. For example, there are three Eigen directions in the Lorenz
63 system, and only one of them has a positive Lyapunov exponent (Kantz and Schreiber, 2005;
Strogatz, 2018). It is known that the direction of reconstruction must correspond to the one with
positive Lyapunov exponent, so that the reconstruction could succeed (Lu et al., 2017 and 2018;
Strogatz, 2018). Such phenomenon might also occur in some natural nonlinear systems. But the
limited data length and red noise are common issues in natural time series, which makes the correct
Lyapunov exponent estimating difficult (Kantz and Schreiber, 2005). In this study, it has been shown
that the correct direction for reconstruction can be selected by CCM index without estimating the
correct Lyapunov exponent.

In section 3.3, the TSAT series is reconstructed from the NHSAT series by means of RC, where

CCM index plays a key role in guiding the reconstruction variable selection. Moreover, the result also
suggests that: (i) the coupling mechanism between the tropics and Northern Hemisphere is nonlinear
since their reconstructions are direction-dependent. (ii) It is noted that CCM index that TSAT cross
maps NHSAT is $\rho_{N \to T} = 0.70$, and CCM index that TSAT cross maps NHSAT is $\rho_{T \to N} = 0.24$.
According to the causality theory (Sugihara et al., 2012; Tsonis, 2018; Zhang et al., 2019), the
asymmetric CCM index indicates that the main energy transport direction might be mainly from the
tropics to the Northern Hemisphere. (iii) If there are enough historical measured series of the Northern





hemisphere and ENSO, the unmeasured climate series (such as SAT) in the tropics might be well
reconstructed.

Finally, it is worth noting once more that there are still more potential applications for machine

learning in the climate studies. For instance, a series $b(t)$ is unmeasured during some periods for the
measuring instrument failure, but there are other kinds of variables without missing observations.
Moreover, CCM can be applied to select the suitable variables coupled with $b(t)$, and then the RC can
be employed to reconstruct the unmeasured part of $b(t)$ (following Fig. 1). This is very useful to some
important climate studies, such as paleoclimate reconstruction (Brown, 1994; Donner 2012; Emile-
Geay and Tingley, 2016), interpolation for the missing points in measurements (Hofstra et al., 2008),
and the parameterization schemes (Wilks, 2005; Vissio and Lucarini, 2018). Our study in this article
is only a beginning for reconstructing climate series by machine learning, and more detailed
investigations will be reported soon.



*Code and data availability.* The code for CCM causality coefficient and the relevant time series are
available upon request to the authors. The data will be made available on zenodo.org once the
manuscript is accepted for discussion.
*Author contribution.* Yu Huang and Zuntao Fu designed this study. All of the authors contributed to
the preparation and writing of the manuscript.
*Competing interests.* The authors declare no competing interest.
*Acknowledgement.* We acknowledge the supports from National Natural Science Foundation of China
through Grants (No. 41675049 and No. 41475048).



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
