# Peer review of "Reconstructing coupled time series in climate systems using three kinds of machine learning methods"

_Earth System Dynamics, 2019_

## Short Comment (SC1) · 9 Jan 2020

In this paper, the authors studied the variable reconstruction problem with several machine learning methods, and test with simulations on several artificial climate models (Lorenz 63 and Lorenz 96) as well as real-world climate data. The authors innovatively use the convergent cross mapping (CCM) to estimate the nonlinear coupling relation between different variables and explain the reason why the variable reconstruction has direction dependence.

This paper is in general well written with sufficient simulations that support its conclusions. However, two main issues need to be addressed.

In Sec. 2.2, the authors introduce the reservoir computing method (Lu et al., 2017) for

the variable reconstruction problem. However, I find this introduction very confusing. It seems that different constructions of reservoir computers for different tasks (for reservoir observer or for predicting future of time seriers) are introduced as different layers for a single reservoir. (lines 144-150). It is also confusing why one would need the so-called prediction reservoir as a layer for this reservoir observer task. (lines 175-178) Does this closed-loop reservoir really being used in the simulation in this paper? If so, why is it necessary? A reservoir observer does not need to feedback its own output to its input, as it is simply trying to estimate variable b(t) based on the measured a(t), rather than predicting the future of both a(t) and b(t). The authors in Sec. 3.2.1-3.2.2 discuss the nonlinear coupling relation, which is essentially the nonlinear observability in the control theory, as being pointed out in (Lu et al., 2017). This direction dependence can be explained by the nonlinear observability. For example, in the Lorenz 63 model, due to the symmetry of that ODE system, both (x(t), y(t), z(t)) and (-x(t),-y(t), z(t)) are solutions on the same chaotic attractor. Thus, one can not construct any nonlinear state-observer that estimates the value of x or y given the time series of variable z. However, a state observer can estimate z(t) given either x(t) or y(t). It was also shown that $x^2(t)$ and $y^2(t)$ can be estimated given z(t) as it is nonlinearly observable. The authors employ CCM to quantify the "nonlinear coupling relation" and show that it is better than a linear coupling relation. It is the reviewer's opinion that a brief discussion of the relation between the CCM and the nonlinear observability should be given. Is CCM essentially the same as nonlinear-observability? If not, what is the difference?
* * *

---

## Referee Comment (RC1) · Anonymous Referee #1 · 13 Jan 2020

This manuscript investigates the potentialities of reconstructing time series using machine learning (ML) techniques. This approach is applied on a set of simple systems, and then applied to the interaction between the Tropical surface temperature and the Northern extra-tropical surface temperature. Different configurations of the machine learning approaches are explored, the reservoir computing, the long short-term memory, but also a simplified version of the latter and back-propagation. The authors use the correlation (for linear systems) and the convergent cross mapping (for nonlinear systems), CCM, as tools to evaluate the ability of the machine learning approaches to reproduce the original time series. Although I find the idea of putting in parallel the CCM with the ability of reconstructing time series based on ML very interesting, the description of the tools and the results is confusing, the presentation is quite poor and

many details on the approaches used are missing.

My first main point is the confusion present in the notation of input/output and the notion of directional dependence. Let me clarify my point by considering Table 2 in which the results for the Lorenz 3-variable system are displayed. The first column indicates the input of the ML approach (also indicated a(t)), the second the output of the ML (also indicated b(t)), while the fourth represents the CCM dependence. The later, as defined at lines 291-297, has high values if b(t) influence a(t). So according to that table if b(t) is influencing a(t) I should get good results of fitting from a(t) to b(t). I am really confused with this claim. I have the same problem with the other tables, and in particular with Table 4 which is even more confusing when related with the discussion in the text. In the table it is indicated that TSAT influences strongly NHSAT but then the ML modelling is done from NHSAT to TSAT. This is what is claimed at lines 463-464, while in the conclusion it is said (line 542) that the TSAT is mainly influencing the NHSAT. I hope this is just a matter of confused notation but I am not sure and I strongly recommend the authors to revisit carefully their notations and interpretation carefully.

A second important concern is the way the ML is used. In Figure 2 there are three parts but it seems to me that the ML system is composed of the two first ones, the third one being the application of the optimized system to new input data. So It should be worth to split both and also to clarify the details of the Machine Learning underlying structure, number of nodes, number of layers (if any)... Details on the different ML systems used are necessary. A detailed description is also missing for the CCM method.

These two main problems prevent me to recommend publication of this manuscript at this stage although the main question addressed is very interesting (CCM vs ML). A considerable effort of clarification and rewriting is necessary.

More specific points:

Line 54: What does mean "wile physics of systems is suggested for consideration"? Please rephrase.

[Figure]

Lines 57-58. You probably meant that: sensitivity to initial conditions is a property of the underlying system giving rise to the climate time series. Chaos theory is a framework in which this type of dynamics can be described. Please rephrase.

Line 67. What is nonlinear correlation? I think that this is not an appropriate terminology. Please revisit your manuscript with that in mind.

Line 72. You speak about "trajectories". Maybe this is more "relationships".

Line 87. "hided"?

Line 111. "learnt" should probably be "reconstructed"

Line 115. "learnt" is probably "estimated" or "inferred".

Figure 1. Why putting the training after the testing? It does not look natural (and also confusing).

Lines 175-178. Quite confusing. Please clarify the way prediction is done. I think that the presentation of the ML approach should be completely revisited.

Line 191. Why using this measure and why 0.1 is a good threshold? These should be detailed.

Line 212. Runge-Kutta integral? What does it mean? Maybe "integrator"?

Section 2.4.2. Please give more details on the way average is done, and whether the seasonality is removed and how? This also open the question on how the parameters of the ML are changing as a function of the season. There is not enough details on how the datasets are handled.

Lines 295-296. Sugihara (1994). This reference does not exist in the reference list. What is "empirical dynamics model? Much more information is needed on the way it is used. Embedding dimension and so on.

Line 302. What is "unstable local correlation". What is this?

Table 2. As already mentioned in my main comment, very confusing. Please modify.

Figure 6. Some typos in titles. Also where is panel (d)? Is it ( c )?

Table 3 and Fig 6. Why not using a multivariate CCM to compare with the ML fitting with multiple predictors?

Lines 536-543. Really confusing. What is influencing what? TSAT or NHSAT?

I have also noted many typographical errors, and the manuscript will benefit for a careful reading by the authors and by an English native speaker to rephrase some sentences.

---

## Author Comment (AC2) · 21 Jan 2020

We thank Dr. Zhixin Lu for his comments and suggestions. We are willing to revise the method description and discuss the association between "nonlinear observability" and "CCM" in our revised manuscript. In the supplement, we also would like to make response to the two issues suggested by Dr. Zhixin Lu.

Please also note the supplement to this comment:
https://www.earth-syst-dynam-discuss.net/esd-2019-63/esd-2019-63-AC2-supplement.pdf

———————————

2019.

---

## Short Comment (SC3) · 5 Feb 2020

Thanks for your responses. Based on the responses, the issues I raised in my previous comments should be resolved. Thus, I look forward to reading the revised manuscript.
* * *

---

## Author Comment (AC3) · 5 Feb 2020

**Reply to the comments of Anonymous Referee #1:**

**The comments of Anonymous Referee #1:**

1. This manuscript investigates the potentialities of reconstructing time series using machine learning (ML) techniques. This approach is applied on a set of simple systems, and then applied to the interaction between the Tropical surface temperature and the Northern extra-tropical surface temperature. Different configurations of the machine learning approaches are explored, the reservoir computing, the long short-term memory, but also a simplified version of the latter and back-propagation. The authors use the correlation (for linear systems) and the convergent cross mapping (for nonlinear systems), CCM, as tools to evaluate the ability of the machine learning approaches to reproduce the original time series.

   **Although I find the idea of putting in parallel the CCM with the ability of reconstructing time series based on ML very interesting, the description of the tools and the results is confusing, the presentation is quite poor and many details on the approaches used are missing.**

**Response:** Thanks for your comments and suggestions! We will carefully improve the description and details of the methods, including the machine learning framework and the CCM theory. **And then, the results and conclusions in the paper are correct. The confusion of Anonymous Referee #1 is the relationship between reconstruction direction and the CCM dependence, and this confusion is mainly induced by the lack of description of the CCM theory.** We will carefully introduce the CCM theory in the revised manuscript, so that the results part can be better understood. Meanwhile, we will also carefully improve the manuscript according to your specific comments and suggestions.

In addition, we also would like to summarize the contributions of this work with the following plain language:

**i)** Investigating how to better apply machine learning to the reconstruction of climate time series (under different coupling dynamics of climate systems), which might be very useful for some important climate problems such as **paleoclimate reconstruction**, **interpolation for the missing points in measurements** and **parameterization schemes**. For instance, for the records of proxy data (tree ring or ice core), we might obtain the data from the historical and current period. For the records of climatic variable like air temperature, we might only obtain the data from the current period. At that time, the conclusions of this paper will be useful to reconstruct the historical data of climatic variable.

**ii)** We proposed to use nonlinear causality coefficient to select explanatory variable, which is demonstrated more effective than the Pearson correlation.

**iii)** Revealing that the reconstruction quality is direction-dependent for two nonlinearly coupled variables: for example, the tropical average surface temperature can be well reconstructed from the average Northern Hemispheric surface temperature, but the average Northern Hemispheric surface temperature cannot be reconstructed from the tropical average surface temperature. Then we explain the reasons and how to deal with such issues. This might be an important suggestion for the future application of data-driven approach to geoscience.

2. My first main point is the confusion present in the notation of input/output and the notion of directional dependence. Let me clarify my point by considering Table 2 in which the results for the Lorenz 3-variable system are displayed. The first column indicates the input of the ML approach (also indicated a(t)), the second the output of the ML (also indicated b(t)), while the fourth represents the CCM dependence. The later, as defined at lines 291-297, has high values if b(t) influence a(t). So according to that table if b(t) is influencing a(t) I should get good results of fitting from a(t) to b(t). I am really confused with this claim.

**Response:** Thanks for your comments and suggestions. The results of Table 2 is correct: the Lorenz-X can be used to reconstruct the Lorenz-Z, but the Lorenz-Z cannot be used to reconstruct the Lorenz-X, which can be also seen in the previous literature of *Lu et al. 2017*[1]. In the paper of *Lu et al. 2017*[1], they used the "nonlinear observability" of the controlled system theory to explain such phenomenon. However, the "nonlinear observability" introduced in *Lu et al. 2017*[1] is only usable in the system with known mathematical equation, here we employ the CCM coefficient which does not rely on any known equation.

  **According to the literature** [2-6], **the claim about the relationship of the CCM dependence and reconstruction direction, is correct and accurate:** if $b$ influence $a$ but $a$ does not influence $b$, the information of $b$ can be shared with $a$ (through the information transfer from $b$ to $a$), but $a$ 's information cannot be shared with $b$ (there exists no information transfer from $a$ to $b$). Hence, the records of $a$ will be encoded with the information of $b$, and the time series of $b$ can be recovered from the records of $a$.

  As the above mentioned, the information transfer induced by causal influence, is the reason of that if $b$ influence $a$ and then $a$ can reconstruct $b$. Further, according to *Sugihara et al. 2012* [4-6], for the CCM index ($\rho_{a\rightarrow b}$), its computation is using a phase-space model [6] to estimate the values of $b$ from $a$'s records. And then the magnitude of $\rho_{a \to b}$ represents: when using $a$'s records to recover the values of $b$, how well the quality is. Hence, the magnitude of $\rho_{a \to b}$ also represents how much information of $b$ is encoded in $a$'s records.

**Sugihara et al. 2012 [4-6] ever suggested that the reconstruction direction is opposite to the causal dependence direction**: when $\rho_{a \to b}$ is high, this means that $b$ causes $a$, and we can get good results of reconstruction from $a$ to $b$.

In the previous manuscript, the above description about the CCM theory is not fully presented, so that it might take confusion to the understanding the results of Tables 2 and 4. But the results about Tables 2 and 4 are really correct. **We will carefully improve the description of the CCM theory [4-6], and add the necessary description of the CCM computational algorithm,** so that the results of the CCM and reconstruction quality will be better understood.

[1] Lu Z, Pathak J, Hunt B, Girvan M, Brockett R, Ott E. Reservoir observers: Model-free inference of unmeasured variables in chaotic systems. Chaos 27(4), 041102 , 2017.

[2] Takens, F.: Detecting strange attractors in turbulence. Dynamical Systems and Turbulence, Lecture Notes in Mathematics, 898, 366–381 (Springer Berlin Heidelberg), 1981.

[3] Hlaváčková-Schindler, K., Paluš, M., Vejmelka, M., Bhattacharya, J. Causality detection based on information-theoretic approaches in time series analysis. Physics Reports, 441(1), 1-46, 2007.

[4] Sugihara, G., May, R., Ye, H., Hsieh, C. H., Deyle, E., Fogarty, M., Munch, S.: Detecting causality in complex ecosystems. Science, 338(6106), 496-500, 2012.

[5] Vannitsem, S., Ekelmans, P. Causal dependences between the coupled ocean–atmosphere dynamics over the tropical Pacific, the North Pacific and the North Atlantic. Earth Syst. Dyn., 9(3), 1063-1083, 2018.

[6] Tsonis, A. A., Deyle, E. R., Ye, H., Sugihara, G.: Convergent cross mapping: theory and an example. In Advances in Nonlinear Geosciences (pp. 587-600), Springer, Cham., 2018.

Additionally, we will modify the sentences in lines 291-297 of the previous manuscript, as the following screenshot shows:

by CCM index $\rho$. For two time series $a(t)$ and $b(t)$, there are two CCM indices (Tsonis et al., 2018):

$\rho_{a \to b}$ and $\rho_{b \to a}$ : (i) If variable $b$ does influence variable $a$, the information of $b(t)$ will be encoded in

$a(t)$, and thus we will acquire a high value of cross-mapping skill $\rho_{a \to b}$. As Sugihara et al. 2012

revealed, the stronger the magnitude of $\rho_{a \to b}$ is, the more information of $b$ is encoded in $a$; (ii)

additionally, if variable $a$ also does influence variable $b$, the information of $a(t)$ will be encoded in

$b(t)$, and thus we will acquire a high value of cross-mapping skill $\rho_{b \to a}$. The more detailed algorithm and explanation for the CCM is shown in the Appendix.

3. I have the same problem with the other tables, and in particular with Table 4 which is even more confusing when related with the discussion in the text. In the table it is indicated that TSAT influences strongly NHSAT but then the ML modeling is done from NHSAT to TSAT. This is what is claimed at lines 463-464, while in the conclusion it is said (line 542) that the TSAT is mainly influencing the NHSAT. I hope this is just a matter of confused notation but I am not sure and I strongly recommend the authors to revisit carefully their notations and interpretation carefully.

**Response:** Thanks for your comments and suggestions. We have inspected the results and conclusions, and the results and conclusions about Table 4 are correct. **Sugihara et al. 2012 [1] ever suggested that the reconstruction direction is opposite to the causal dependence direction.** The confusion about the relationship between reconstruction direction and the CCM dependence, is induced by the lack of description of the CCM theory in the previous manuscript.

Firstly, we can comprehend the CCM index according to the literature [1-4]: if $b$ does influence $a$ ($a$ and $b$ are two arbitrary variables), and then the information of $b$ can be shared with $a$ (through the information transfer from $b$ to $a$). Hence, the records of $a$ will be encoded with the information of $b$, and the time series of $b$ can be recovered from the records of $a$. At that time, the CCM coefficient $\rho_{a \to b}$ denotes: when using $a$'s records to recover the values of $b$, how well the quality is. Likewise, the magnitude of $\rho_{a \to b}$ represents how much information of $b$ is encoded in the records of $a$.

Then, in our results about using NHSAT to reconstruct TSAT, the CCM index that NHSAT cross maps TSAT is of high value (0.7). This suggests that the NHSAT's records are able to recover the values of TSAT, which stems from that the information of TSAT is encoded in NHSAT. But the CCM index that TSAT cross maps NHSAT is of high value (0.24). According to the CCM theory, we know that the influence from NHSAT to TSAT, is not strong as the influence from TSAT to NHSAT, which also consists with the real dynamical process revealed by previous literature [6].

Finally, the information transfer inferred from the CCM suggests that: when employing Reservoir Computing to reconstruct TSAT from the NHSAT's records, the reconstruction quality will be better than reconstruct NHSAT from the TSAT's records. And our results are really consisting with the suggestion of CCM.

We will carefully improve the description of the CCM theory [1, 4, 5], and add the necessary description of the CCM computational algorithm, so that the results of the CCM and reconstruction quality will be better understood.

[1] Sugihara, G., May, R., Ye, H., Hsieh, C. H., Deyle, E., Fogarty, M., Munch, S.: Detecting causality in complex ecosystems. Science, 338(6106), 496-500, 2012.

[2] Takens, F.: Detecting strange attractors in turbulence. Dynamical Systems and Turbulence, Lecture Notes in Mathematics, 898, 366–381 (Springer Berlin Heidelberg), 1981.

[3] Hlaváčková-Schindler, K., Paluš, M., Vejmelka, M., Bhattacharya, J. Causality detection based on information-theoretic approaches in time series analysis. Physics Reports, 441(1), 1-46, 2007.

[4] Vannitsem, S., Ekelmans, P. Causal dependences between the coupled ocean–atmosphere dynamics over the tropical Pacific, the North Pacific and the North Atlantic. Earth Syst. Dyn., 9(3), 1063-1083, 2018.

[5] Tsonis, A. A., Deyle, E. R., Ye, H., Sugihara, G.: Convergent cross mapping: theory and an example. In Advances in Nonlinear Geosciences (pp. 587-600), Springer, Cham., 2018.

[6] Vallis, G. K., Farneti, R.: Meridional energy transport in the coupled atmosphere–ocean system: Scaling and numerical experiments. Q. J. Roy. Meteor. Soc., 135(644), 1643-1660, 2009.

4. A second important concern is the way the ML is used. In Figure 2 there are three parts but it seems to me that the ML system is composed of the two first ones, the third one being the application of the optimized system to new input data. So **It should be worth to split both and also to clarify the details of the Machine Learning underlying structure, number of nodes, number of layers (if any)**… **Details on the different ML systems used are necessary. A detailed description is also missing for the CCM method**.

**Response:** Thanks for your comments and suggestions. By means of the first two components shown in Figure 1*, the $a(t)$ is trained and then $\psi[r^*(t)]$ is obtained. In this procedure, the value of $\psi[r^*(t)]$ is already very close to the value of $b(t)$. Then, if $\psi[r^*(t)]$ is feedback to function "$f$" and "$\psi$", this repetitive operation might make the value of $\psi[r^*(t)]$ more close to the value of $b(t)$. Actually we also found this repetitive operation no longer influenced the results. This is to say, that the third component shown in Figure 1* might be redundant in this reconstruction framework, and the first two components are enough.

The Reservoir Computer framework used in our work is developed in *Lu et al.* 2017 [1]. In *Lu et al.* 2017 [1], the Reservoir Computer framework only has the first two components shown in Figure 1*. We have tested the third component (a repetitive operation for the first two components) did not influence the results, and the first two components were enough. In the revised manuscript, we will carefully improve the diagram and the description of Reservoir computer according to the introduction in *Lu et al.* 2017 [1].

[Figure]

*Listening reservoir*  *Synchronization reservoir*  *Prediction reservoir*

Figure 1* The schematic of Reservoir computer in the previous manuscript (we will revised this figure in the revised manuscript).

[1] Lu Z, Pathak J, Hunt B, Girvan M, Brockett R, Ott E. Reservoir observers: Model-free inference of unmeasured variables in chaotic systems. Chaos 27(4), 041102 , 2017.

**Then, we will improve the detail description of Reservoir Computer, including the structure, number of nodes, number of layers, and so on.** As the following screenshot shows:

computer (RC), back propagation (BP), and long short-term memory (LSTM) neural networks. The newly developed RC (Du et al., 2017; Lu et al., 2017; Pathak et al., 2018) has two layers: listening reservoir and synchronization reservoir (see Fig. 2). If $a(t)$ and $b(t)$ denote two time series from an arbitrary system, and then the following steps can estimate $b(t)$ from $a(t)$:

[Figure]

$$r^*(t) = \tanh[M \cdot r(t) + W_{in} \cdot a(t) + E] \qquad \hat{b}(t) = W_{out} \cdot r^*(t)$$

**Figure 2** Schematic of the RC neural network: the two layers of the RC neural network are the listening reservoir, and the synchronization reservoir. A time series $a(t)$ is input into the RC neural network. After the training process, the time series of $b$ variable can be reconstructed by machine learning, denoted as $\hat{b}(t)$.

(i) $a(t)$ (a vector with length $L$) is input into the listening reservoir layer. There are four components in this neural network layer: the initial reservoir state $r(t)$ (a vector with dimension $N$, representing the $N$ neurons), the adjacent matrix "$M$" (size $N \times N$) representing connectivity of the $N$ neurons, the input-to-reservoir weight matrix "$W_{in}$" (size $N \times L$), and the unit matrix "$E$" (size $N \times N$) which is crucial for modulating the bias in the training process. The elements of "$M$" and "$W_{in}$" are randomly chosen from a uniform distribution in [−1, 1], and we set $N = 1000$ here. These components are associated with Eq. (1), and then an updated reservoir state $r^*(t)$ is output.

$r^*(t) = \tanh[M \cdot r(t) + W_{in} \cdot a(t) + E]$,                                         (1)

(ii) $r^*(t)$ then gets into the synchronization reservoir consisting of the reservoir-to-output matrix

 "$W_{out}$". As Eq. (2) shows, $r^*(t)$ will be trained as the estimated value $\hat{b}(t)$. The mathematical form of "$W_{out}$" is shown by Eq. (3), which is a trainable matrix that fits the relation between $r^*(t)$ and

$b(t)$ in the training process. "$\|\cdot\|$" denotes the $L_2$-norm of a vector ($L_2$ represents the least square method) and $\alpha$ is the ridge regression coefficient, whose values will be determined after the training.

$$\hat{b}(t) = W_{out} \cdot r^*(t),\qquad(2)$$

$$W_{out} = \arg\min_{W_{out}} \left\| W_{out} \cdot r^*(t) - Y(t+\tau) \right\| + \alpha \| W_{out} \|,\qquad(3)$$

After this reservoir neural network has been trained, we can use it to estimate $b(t)$, where the estimated value is noted as $\hat{b}(t)$.

For the details of LSTM and BP, since both of them have been widely used and well-known in many fields, and in recent years the Matlab language turns them into products for ease of use. Their underlying structures and usage guideline are open access in https://ww2.mathworks.cn/help/deeplearning. We will add the details of parameter setting in the revised manuscript.

**Additionally, we will add the CCM computational algorithm into the revised manuscript, as the following screenshot shows:**

**Appendix**

**The CCM theory**

Considering $a(t)$ and $b(t)$ as two observational time series, we begin with the cross mapping [1] from $a(t)$ to

$b(t)$ through the following steps:

i) Embedding $a(t)$ (with length $L$) into the phase space with the vector $M_a(t_i) = \{a_{t_i}, a_{t_i - \tau_0}, \ldots, a_{t_i - (m-1)\tau}\}$ ("$t_i$"

represents a historical moment in the observations), where embedding dimension ($m$) and time delay ($\tau$) can be determined through the false nearest neighbor algorithm (Hegger and Kantz, 1999).

ii) Estimating the weight parameter $w_i$ denoting the associated weight between two vectors "$M_a(t)$" and "$M_a(t_i)$"

("$t$" denotes the excepted time in this cross mapping), defined as:

$$w_i = \frac{u_i}{\sum_{i=1}^{m+1} u_i},\qquad(A1)$$

$$u_i = exp\left\{-\frac{d\,[M_a(t), M_a(t_i)]}{d\,[M_a(t), M_a(t_l)]}\right\},\tag{A2}$$

where $d\,[M_a(t), M_a(t_i)]$ denotes the Euler distance between vectors "$M_a(t)$" and "$M_a(t_i)$". The nearest neighbor to "$M_a(t)$" generally corresponds to the largest weight.

iii) Cross mapping the value of $b(t)$ by

$$\hat{b}(t) = \sum_{i=l}^{m+l} w_i b(t_i).\tag{A3}$$

$\hat{b}(t)$ denotes the estimated value of $b(t)$ with this phase-space cross mapping. Then, we will evaluate the cross mapping skill (Sugihara et al., 2012; Tsonis et al., 2018) as Eq. (A4) shows:

$$\rho_{a\to b} = corr.\,[b(t), \hat{b}(t)].\tag{A4}$$

The cross mapping skill from $b$ to $a$ is also measured according to the above steps, marked as $\rho_{b\to a}$. *Sugihara et*

*al.* and *Tsonis et al.* defined the causal inference from $\rho_{a\to b}$ and $\rho_{b\to a}$ like that: (i) if $\rho_{a\to b}$ is convergent when

$L$ is increased, and $\rho_{a\to b}$ is of high value, then $b$ is suggested to be a causation of $a$. (ii) Besides, if $\rho_{b\to a}$ is also convergent when $L$ is increased, and is of high value, then the causal relationship between $a$ and $b$ is bidirectional ($a$ and $b$ cause each other). In our study, all the values of CCM indices are measured when they are convergent with the data length.

According to the literature (Takens, 1981; Sugihara et al., 2012): if $b$ influence $a$ but $a$ does not influence $b$, the information of $b$ can be shared with $a$ (through the information transfer from $b$ to $a$), but the information of $a$

cannot be shared with b (there exists no information transfer from $a$ to $b$). Hence, the records of $a$ will be encoded with the information of $b$, and the time series of $b$ can be recovered from the records of $a$. For the CCM index ($\rho_{a\to b}$), its magnitude represents how much information of $b$ is encoded in the records of $a$. So that the high value of $\rho_{a\to b}$ means that $b$ causes $a$, and we can get good results of reconstruction from $a$ to $b$.

5. These two main problems prevent me to recommend publication of this manuscript at this stage although the main question addressed is very interesting (CCM vs ML). A considerable effort of clarification and rewriting is necessary.

**Response:** Thanks for your comments and suggestions! According to your above suggestions, we will carefully work on the more detailed clarification and rewriting for the machine learning method and the CCM theory, so that the relationship between CCM and machine learning can be better presented. And then, results and conclusions will be better understood.

More specific points:

6. Line 54: What does mean "wile physics of systems is suggested for consideration"? Please rephrase.

**Response:** Thank you! The excepted meaning is that: we should focus on whether the dynamical properties in the underlying system can be described, and how the dynamical properties will influence the performance of machine learning. We will revise these sentences as the following screenshot:

| | |
|---|---|
| 53 | Kratzert et al., 2019; Feng et al., 2019). Recently it is demonstrated the large potentials for machine |
| 54 | learning to simulate the temporal dynamics of complex systems (Pathak et al., 2017; Du et al., 2017; |
| 55 | Watson, 2019). Thereinto, it is suggested for consideration whether the dynamical properties in the |
| 56 | underlying system can be described. For example, chaos is a dynamical property of the underlying |
| 57 | system giving rise to the climatic time series (Lorenz, 1963; Patil et al., 2001), and then the results |
| 58 | of applying machine learning to Lorenz system and Rossler model show that their chaotic attractors |
| 59 | are able to be well described (Pathak et al., 2017; Lu et al., 2018; Carroll, 2018), which |
| 60 | demonstrates the usability of machine learning on climatic series. In the further study, we should |
| 61 | also focus on how the dynamical properties will influence the performance of machine learning. |

7. Lines 57-58. You probably meant that: sensitivity to initial conditions is a property of the underlying system giving rise to the climate time series. Chaos theory is a framework in which this type of dynamics can be described. Please rephrase.

**Response:** Thank you! We will carefully rephrase these sentences, as the following screenshot shows:

| | |
|---|---|
| 55 | Watson, 2019). Thereinto, it is suggested for consideration whether the dynamical properties in the |
| 56 | underlying system can be described. For example, chaos is a dynamical property of the underlying |
| 57 | system giving rise to the climatic time series (Lorenz, 1963; Patil et al., 2001), and then the results |
| 58 | of applying machine learning to Lorenz system and Rossler model show that their chaotic attractors |
| 59 | are able to be well described (Pathak et al., 2017; Lu et al., 2018; Carroll, 2018), which |
| 60 | demonstrates the usability of machine learning on climatic series. In the further study, we should |
| 61 | also focus on how the dynamical properties will influence the performance of machine learning. |

8. Line 67. What is nonlinear correlation? I think that this is not an appropriate terminology. Please revisit your manuscript with that in mind.

**Response:** Thank you! We will carefully rephrase the explanation of "nonlinear correlation" in the revised manuscript.

Here the excepted meaning of "nonlinear correlation" is that: for two variables from a common system, their time series might have dynamical relationship with each other. Sometimes the linear Pearson correlation of these two time series is weak or even equal to zero, but by means of some other statistical measurement their relationship can be quantified. At that time, such relationship whose linear correlation is potentially weak, is regarded as nonlinear correlation.

We will modify the sentences as the following screenshot:

| 69 | Sugihara et al., 2012; Emile-Geay and Tingley, 2016). However, previous studies (Sugihara et al., |
| 70 | 2012; Emile-Geay and Tingley, 2016) suggest that, even though the linear correlation of two |
| 71 | variables is potentially weak, they are actually related to each other and can be exploited by analysis. |
| 72 | For instance, the linear cross-correlations of sea surface temperature series observed in different |
| 73 | tropical areas are unstable and vary with time, which leads to an overall weak linear correlation, but |
| 74 | this non-linear correlation can result in better El Niño predictions (Ludescher et al., 2014; Conti et |
| 75 | al., 2017). The phase plots of the ENSO/PDO index and some proxy variables are not linear lines |
| 76 | but nonlinear relationships, which contributes greatly to reconstructing longer climate series |

9. Line 72. You speak about "trajectories". Maybe this is more "relationships".

**Response:** Thank you! We will revise this word in the manuscript, as the screenshot shows:

| 75 | al., 2017). The phase plots of the ENSO/PDO index and some proxy variables are not linear lines |
| 76 | but nonlinear relationships, which contributes greatly to reconstructing longer climate series |

10. Line 87. "hided"?

**Response:** Thank you! We will revise this word in the manuscript, as the screenshot shows:

reconstructed from the TSAT series. Such a contrasting result means that the machine learning approach is not the same as the traditional statistical methods. Accordingly, is there any dynamical property existed in the NHSAT-TSAT coupling influencing this result? Furthermore, one might wonder if the similar phenomenon will be universal for some other coupled climate series.

**11. Line 111. "learnt" should probably be "reconstructed".**

**Response:** Thank you! We will revise this word in the manuscript, as the screenshot shows:

$a_1(t), a_2(t),...,a_n(t)$ and output $b(t)$. If this inherent coupling relation can be reconstructed by machine learning in the training series, the reconstructed coupling relation should be reflected by machine learning in the testing series. Therefore, the workflow of our study can be summarized as follows (see Fig. 1):

(i) During the training period, $a_1(t), a_2(t),...,a_n(t)$ and $b(t)$ are input into the machine learning frameworks to learn the coupling or dynamic relation $b(t) = F[a_1(t), a_2(t),...,a_n(t)]$. The inferred coupling relation is denoted as $b(t) = \hat{F}[a_1(t), a_2(t),...,a_n(t)]$. Then it is tested whether this coupling relation can be reconstructed by machine learning.

(ii) The second step is accomplished with the testing series to apply the reconstructed coupling relation $\hat{F}$ together with only $a_1(t'), a_2(t'),...,a_n(t')$ to derive $b(t')$, denoted as $\hat{b}(t')$. $\hat{b}(t')$ is called "the reconstructed $b(t')$" since only $a_1(t'), a_2(t'),...,a_n(t')$ and the reconstructed coupling relation $\hat{F}$ have been taken into account.

(iii) The first objective of this study is to answer whether the coupling relation

$b(t) = F[a_1(t), a_2(t),...,a_n(t)]$ can be reconstructed by machine learning, i.e., whether the

**12. Line 115. "learnt" is probably "estimated" or "inferred".**

**Response:** Thank you! We will revise this word in the manuscript, as the screenshot shows:

(i) During the training period, $a_1(t), a_2(t),...,a_n(t)$ and $b(t)$ are input into the machine learning frameworks to learn the coupling or dynamic relation $b(t) = F[a_1(t), a_2(t),...,a_n(t)]$. The inferred coupling relation is denoted as $b(t) = \hat{F}[a_1(t), a_2(t),...,a_n(t)]$. Then it is tested whether this coupling

13. Figure 1. Why putting the training after the testing? It does not look natural (and also confusing).

**Response:** Thanks for your suggestions. Such arrangement is due to the consideration of reconstructing climate records. We are inspired by that it is often necessary to reconstruct the historical records for climate variables.

For instance, as Figure 2* shows, for the records of proxy data (tree ring or ice core, labeled as $a(t)$ in Figure 2*), we might obtain the data from the historical and current period. For the records of climatic variable like air temperature (labeled as $b(t)$ in Figure 2*), we might only obtain the data from the current period. At that time, the data-driven approach (such linear regression) is often applied to fit the relation between proxy data ($a(t)$) and air temperature ($b(t)$) through their current observational data, and then the historical proxy data and the fitted relationship can be used to reconstruct the historical records of air temperature.

[Figure]

Figure 2* The blue solid line denotes the observational records of climatic variable (labeled as $b(t)$) in current period. The blue dashed line denotes that the records of climatic variable are absence of observation in the past time. The red solid line denotes the proxy data (labeled as $a(t)$) in both of current period and past time.

The above reconstruction scheme is also very useful for some important climate problems such as **paleoclimate reconstruction** [1], **interpolation for the missing points in measurements** [2] and **parameterization schemes** [3]. Our study is motivated by investigating how to better apply machine learning to the reconstruction of climate time series (under different coupling dynamics of climate systems).

[1] Emile-Geay, J., Tingley, M.: Inferring climate variability from nonlinear proxies: application to paleo-ENSO studies. Clim. Past., 12(1), 31-50, 2016.
[2] Hofstra, N., Haylock, M., New, M., Jones, P., Frei, C.: Comparison of six methods for the interpolation of daily European climate data. J. Geophys. Res., 113(D21), 2008.

[3] Vissio, G., Lucarini, V.: A proof of concept for scale‑adaptive parameterizations: the case of the Lorenz 96 model. Q. J. Roy. Meteor. Soc., 144(710), 63-75, 2018.

14. Lines 175-178. Quite confusing. Please clarify the way prediction is done. I think that the presentation of the ML approach should be completely revisited.

**Response:** Thanks for your suggestions. We will thoroughly rewrite this part about the machine learning framework, and detail description of Reservoir Computer, including the structure, number of nodes, number of layers will be clearly presented.

The Reservoir Computer framework used in our work is developed in *Lu et al.* 2017 [1]. And we will refer the introduction in *Lu et al.* 2017 [1] to modify the description. Our modified version will be as the screen shot shows:

[1] Lu Z, Pathak J, Hunt B, Girvan M, Brockett R, Ott E. Reservoir observers: Model-free inference of unmeasured variables in chaotic systems. Chaos 27(4), 041102 (2017).

computer (RC), back propagation (BP), and long short-term memory (LSTM) neural networks. The newly developed RC (Du et al., 2017; Lu et al., 2017; Pathak et al., 2018) has two layers: listening reservoir and synchronization reservoir (see Fig. 2). If $a(t)$ and $b(t)$ denote two time series from an arbitrary system, and then the following steps can estimate $b(t)$ from $a(t)$:

[Figure]

$$r^*(t) = \tanh[M \cdot r(t) + W_{in} \cdot a(t) + E] \qquad \hat{b}(t) = W_{out} \cdot r^*(t)$$

**Figure 2** Schematic of the RC neural network: the two layers of the RC neural network are the listening reservoir, and the synchronization reservoir. A time series $a(t)$ is input into the RC neural network. After the training process, the time series of $b$ variable can be reconstructed by machine learning, denoted as $\hat{b}(t)$.

(i) $a(t)$ (a vector with length $L$) is input into the listening reservoir layer. There are four components in this neural network layer: the initial reservoir state $r(t)$ (a vector with dimension $N$, representing the $N$ neurons), the adjacent matrix "$M$" (size $N{\times}N$) representing connectivity of the $N$ neurons, the input-to-reservoir weight matrix "$W_{in}$" (size $N{\times}L$), and the unit matrix "$E$" (size $N{\times}N$) which is crucial for modulating the bias in the training process. The elements of "$M$" and "$W_{in}$" are randomly chosen from a uniform distribution in $[-1, 1]$, and we set $N = 1000$ here. These components are associated with Eq. (1), and then an updated reservoir state $r^*(t)$ is output.

$$r^*(t)=\tanh\left[M \cdot r(t) + W_{in} \cdot a(t) + E\right], \tag{1}$$

(ii) $r^*(t)$ then gets into the synchronization reservoir consisting of the reservoir-to-output matrix

"$W_{out}$". As Eq. (2) shows, $r^*(t)$ will be trained as the estimated value $\hat{b}(t)$. The mathematical form of "$W_{out}$" is shown by Eq. (3), which is a trainable matrix that fits the relation between $r^*(t)$ and

$b(t)$ in the training process. "$\|\cdot\|$" denotes the $L_2$-norm of a vector ($L_2$ represents the least square method) and $\alpha$ is the ridge regression coefficient, whose values will be determined after the training.

$$\hat{b}(t)=W_{out} \cdot r^*(t), \tag{2}$$

$$W_{out}=\arg\min_{W_{out}}\left\|W_{out} \cdot r^*(t)-Y(t + \tau)\right\| +\alpha\|W_{out}\|, \tag{3}$$

After this reservoir neural network has been trained, we can use it to estimate $b(t)$, where the estimated value is noted as $\hat{b}(t)$.

15. Line 191. Why using this measure and why 0.1 is a good threshold? These should be detailed.

**Response:** Thanks for your suggestions. Normalizing the RMSE is to compare the time series with different variability and unit [1, 2]. For instance, the time series of $x_1$ and $x_2$ in Figure 3* are both with zero mean and unit variance, but the extreme values of $x_2$ are much stranger than of $x_1$. It is revealed [1, 2] that such difference will interfere in the fair comparison of the RMSE. In order to avoid such interference induced by the extreme values, we are suggested to normalize the RMSE with the max distribution range of the original data [1, 2], as equation (5) shows.

$$RMSE = \sqrt{\frac{1}{k} \sum_t [b(t') - \hat{b}(t')]^2},$$ (4)

$$nRMSE = \frac{RMSE}{\max[b(t')] - \min[b(t')]}.$$ (5)

[Figure]

Figure 3* The standardized time series of $x_1$(blue) and $x_2$ (red) with zero mean and unit variance. The $x_1$ is a random time series with Gaussian probability distribution, and $x_2$ is a random time series with extreme probability distribution.

"nRMSE = 0.1" means that the RMSE occupies 10% of the max distribution range of the original data, and this is a tolerable level of the bias [1, 2]. In the figures of comparing reconstructed series with real series, we can observe that when the reconstructed series is close to the real series in curves, the corresponding nRMSE is less than 0.1.

[1] Hyndman, R. J., Koehler, A. B.: Another look at measures of forecast accuracy. Int. J. Forecasting., 22(4), 679-688, 2006.

[2] Pennekamp, F., Iles, A. C., Garland, J., Brennan, G., Brose, U., Gaedke, U., Novak, M.: The intrinsic predictability of ecological time series and its potential to guide forecasting. Ecol, Monogr., e01359, 2019.

We will carefully explain the meaning of nRMSE and its threshold in the revised manuscript, as the following screenshot shows:

To evaluate the quality of reconstruction by machine learning, the root mean squared error (RMSE) of residual series (Hyndman and Koehler, 2006) is adopted (Eq. (4)), which represents the difference between the original series $b(t')$ and the reconstructed series $\hat{b}(t')$. In order to fairly compare the errors of reconstructing different processes with different variability and units (Hyndman and Koehler, 2006; Pennekamp et al., 2018), we will normalize the RMSE as Eq. (5)

shows.

$$RMSE = \sqrt{\frac{1}{k} \sum_t [b(t') - \hat{b}(t')]^2},$$ (4)

$$nRMSE = \frac{RMSE}{\max[b(t')] - \min[b(t')]}.$$ (5)

16. Line 212. Runge-Kutta integral? What does it mean? Maybe "integrator"?

**Response:** Thanks for your suggestions. We will revise this word in the manuscript, as the screenshot shows:

when $\theta$ is much smaller than 1. The Runge-Kutta integrator of the fourth order and periodic boundary condition are adopted (that is: $X_0 = X_K$ and $X_{K+1} = X_1$; $Y_{k,\,0} = Y_{k-1,\,J}$ and $Y_{k,\,J+1} = Y_{k+1,\,1}$), and

17. Section 2.4.2. Please give more details on the way average is done, and whether the seasonality is removed and how?

This also open the question on how the parameters of the ML are changing as a function of the season.

There is not enough details on how the datasets are handled.

**Response:** Thanks for your suggestions. We will improve the details on the way average is done in the manuscript.

The seasonality was not removed, and this did not influence the parameters of the machine learning. The reasons are as the following shows:

**Firstly,** literature [1-4] has revealed that seasonal cycle of air temperature is time-varying (especially for the mid-latitude regions [1] and tropics [2]), and the existing methods are often hard to thoroughly remove such time-varying seasonal cycle [4]. So that removing seasonality might take some controversial and unknown bias for the results [5].

[1] Paluš, M., Novotná, D., Tichavský, P.: Shifts of seasons at the European mid‐latitudes: Natural fluctuations correlated with the North Atlantic Oscillation. Geophysical research letters, 32(12), 2005.

[2] Qian, C., Wu, Z., Fu, C., Wang, D.: On changing El Niño: A view from time-varying annual cycle, interannual variability, and mean state. Journal of Climate, 24(24), 6486-6500, 2011.

[3] Jajcay, N., Hlinka, J., Kravtsov, S., Tsonis, A. A., Paluš, M.: Time scales of the European surface air temperature variability: The role of the 7–8 year cycle. Geophysical Research Letters, 43(2), 902-909, 2016.

[4] Deng, Q., Nian, D., Fu, Z.: The impact of inter-annual variability of annual cycle on long-term persistence of surface air temperature in long historical records. Climate dynamics, 50(3-4), 1091-1100, 2018.

[5] Theiler, J., Eubank, S.: Don't bleach chaotic data. Chaos: An Interdisciplinary Journal of Nonlinear Science, 3(4), 771-782, 1993.

**Secondly,** if focusing on the application in reconstructing regional temperature [6-8], the annual variability will be the most important and commonly concerned. At that time, the seasonality is not necessary to be removed. And as the Figure 4* shows, the annual variability of reconstructed series is really close to the real series. If we remove the seasonality, it might take with some unknown bias [4-5].

[6] Van Engelen, A. F., Buisman, J., Jnsen, F.: A millennium of weather, winds and water in the low countries. In History and climate (pp. 101-124). Springer, Boston, MA, 2001.

[7] Moberg, A., Sonechkin, D. M., Holmgren, K., Datsenko, N. M., Karlen, W.: 2,000-year Northern Hemisphere temperature reconstruction. IGBP PAGES/World Data Center for Paleoclimatology Data Contribution Series, 19, 2005.

[8] Mann, M. E., Zhang, Z., Rutherford, S., Bradley, R. S., Hughes, M. K., Shindell, D., Ni, F.: Global signatures and dynamical origins of the Little Ice Age and Medieval Climate Anomaly. Science, 326(5957), 1256-1260, 2009.

**Thirdly,** when employing neural network approach, it is a common step to divide the data into training data and testing data. Then the training data is used to train the parameters of neural network. After the training process is accomplished, the parameters of neural network will be determined and fixed. And then, the trained neural network will be used in the testing data, and they will be not changed any more.

**Fourthly,** if dividing the time series into different seasons, and respectively reconstructing them in different seasons, the parameters of machine learning might be changing in different seasons. However, after dividing these daily time series into different seasons, the data length will be not long enough to accomplish the machine learning approach, which might take the large bias to the results. So, we did not divide the time series according to different seasons, and the seasonality will not influence the parameters of machine learning changing with the season.

[Figure]

Figure 4* Comparison between the annual mean values of reconstructed TSAT (red) and the annual mean values of original TSAT (blue).

18. Lines 295-296. Sugihara (1994). This reference does not exist in the reference list. What is "empirical dynamics model? Much more information is needed on the way it is used. Embedding dimension and so on.

**Response:** Thanks for your suggestions. We will revise this part in the manuscript, as the screenshot shows:

**Appendix**

**The CCM theory**

Considering $a(t)$ and $b(t)$ as two observational time series, we begin with the cross mapping [1] from $a(t)$ to $b(t)$ through the following steps:

i) Embedding $a(t)$ (with length $L$) into the phase space with the vector $M_a(t_i) = \{a_{t_i}, a_{t_i - \tau_0}, \ldots, a_{t_i - (m-1)\tau}\}$ ("$t_i$" represents a historical moment in the observations), where embedding dimension ($m$) and time delay ($\tau$) can be determined through the false nearest neighbor algorithm (Hegger and Kantz, 1999).

ii) Estimating the weight parameter $w_i$ denoting the associated weight between two vectors "$M_a(t)$" and "$M_a(t_i)$" ("$t$" denotes the excepted time in this cross mapping), defined as:

$$w_i = \frac{u_i}{\sum_{i=1}^{m+l} u_i}, \tag{A1}$$

$$u_i = exp\left\{-\frac{d\,[M_a(t), M_a(t_i)]}{d\,[M_a(t), M_a(t_l)]}\right\}, \tag{A2}$$

where $d\,[M_a(t), M_a(t_i)]$ denotes the Euler distance between vectors "$M_a(t)$" and "$M_a(t_i)$". The nearest neighbor to "$M_a(t)$" generally corresponds to the largest weight.

iii) Cross mapping the value of $b(t)$ by

$$\hat{b}(t) = \sum_{i=1}^{m+l} w_i b(t_i). \tag{A3}$$

$\hat{b}(t)$ denotes the estimated value of $b(t)$ with this phase-space cross mapping. Then, we will evaluate the cross mapping skill (Sugihara et al., 2012; Tsonis et al., 2018) as Eq. (A4) shows:

$$\rho_{a \to b} = corr.\,[b(t), \hat{b}(t)]. \tag{A4}$$

The cross mapping skill from $b$ to $a$ is also measured according to the above steps, marked as $\rho_{b \to a}$. *Sugihara et al.* and *Tsonis et al.* defined the causal inference from $\rho_{a \to b}$ and $\rho_{b \to a}$ like that: (i) if $\rho_{a \to b}$ is convergent when $L$ is increased, and $\rho_{a \to b}$ is of high value, then $b$ is suggested to be a causation of $a$. (ii) Besides, if $\rho_{b \to a}$ is also convergent when $L$ is increased, and is of high value, then the causal relationship between $a$ and $b$ is bidirectional ($a$ and $b$ cause each other). In our study, all the values of CCM indices are measured when they are convergent with the data length.

According to the literature (Takens, 1981; Sugihara et al., 2012): if $b$ influence $a$ but $a$ does not influence $b$, the information of $b$ can be shared with $a$ (through the information transfer from $b$ to $a$), but the information of $a$ cannot be shared with b (there exists no information transfer from $a$ to $b$). Hence, the records of $a$ will be encoded with the information of $b$, and the time series of $b$ can be recovered from the records of $a$. For the CCM index ($\rho_{a \to b}$), its magnitude represents how much information of $b$ is encoded in the records of $a$. So that the high value of $\rho_{a \to b}$ means that $b$ causes $a$, and we can get good results of reconstruction from $a$ to $b$.

19. Line 302. What is "unstable local correlation". What is this?

**Response:** Thank you! The expected meaning of "unstable local correlation" is that the local Pearson correlation between two variables is time-varying. As the Figure 5*(a) shows, the time series of $X$ and $Z$ are sometimes positively correlated but sometimes nonlinear correlated at different regimes. Hence, the overall Pearson correlation between $X$ and $Z$ is very weak. Such time-varying local Pearson correlation is suggested to be universal in nonlinear dynamical systems [1].

[1] Sugihara, G., May, R., Ye, H., Hsieh, C. H., Deyle, E., Fogarty, M., Munch, S.: Detecting causality in complex ecosystems. Science, 338(6106), 496-500, 2012.

We will modify the word in the revised manuscript for better understanding, as the following screenshot shows:

> 300      turn to the Lorenz 63 system (Lorenz, 1963). There is a very weak linear correlation between
>
> 301      variables $X$ and $Z$ (with a Pearson correlation coefficient of 0.002) in the Lorenz63 model (see Table
>
> 302      2), and such a weak linear correlation is induced by the time-varying local correlation between
>
> 303      variables $X$ and $Z$ (see Fig. 4a): For example, $X$ and $Z$ are negatively correlated in the interval of 0–
>
> 304      200, but positively correlated in 200–400. This alternation of negative and positive correlation
>
> 305      appears over the whole processes of $X$ and $Z$, which leads to an overall weak linear correlation. In

[Figure]

Figure 5* (a) The $X$ time series (black) and the $Z$ time series (blue) of the Lorenz 63 system. (b) Scatter plot of $X$ time series and $Z$ time series of the Lorenz 63 model (blue dots).

**20.** Table 2. As already mentioned in my main comment, very confusing. Please modify.

**Response:** Thanks for your suggestions. The results and conclusion of Table 2 is correct (see also *Lu et al. 2017*[1]), and this confusion is induced by the lack description of the CCM theory. After the CCM theory is well explained in the manuscript, the result can be better understood.

[1] Lu Z, Pathak J, Hunt B, Girvan M, Brockett R, Ott E. Reservoir observers: Model-free inference of unmeasured variables in chaotic systems. Chaos 27(4), 041102 (2017).

**21.** Figure 6. Some typos in titles. Also where is panel (d)? Is it (c)?

**Response:** Thank you! We will revise this typo in the manuscript, as the screenshot shows:

> 372      **Figure 6** (a) The $Y_{1,1}$ time series(black), $X_2$ time series (black) and $X_1$ time series(blue)of the Lorenz 96 model. (b)
>
> 373      By means of the RC machine learning, when using $Y_{1,1}$, $X_2$ and multivariate to be the explanatory variable
>
> 374      respectively, the corresponding reconstructed $X_1$ time series are showed respectively from the top panel to the
>
> 375      bottom panel (red lines), and the original $X$ time series are presented by the blue lines. (c) By means of the LSTM
>
> 376      machine learning, when using $Y_{1,1}$, $X_2$ and multivariate to be the explanatory variable respectively, the

**22.** Table 3 and Fig 6. Why not using a multivariate CCM to compare with the ML fitting with multiple predictors?

**Response:** Many thanks for your suggestions! The multi-variable CCM analysis might be useful and promising, but first of all we need to know which variable is able to become the explanatory variable. Similar to the multi-variable regression analysis, if we do not know the Pearson correlation between the target variable with every potential explanatory variable, the multi-variable regression will easily suffer from the overfitting problem.

Considering the potential **overfitting problem** and **common-driver problem** [1-2], the comparison between the multi-variable CCM and the multi-variable machine learning **absolutely deserves a further investigation**. This might occupy too many words and figures in the manuscript, so that the presentation of the main and original ideal might be influenced. In the future study, we will consider a thorough investigation for the comparison between the multi-variable CCM and the multi-variable machine learning.

[1] Runge, J., Heitzig, J., Petoukhov, V., Kurths, J.: Escaping the curse of dimensionality in estimating multivariate transfer entropy. Physical review letters, 108(25), 258701, 2012.

[2] Runge, J., Bathiany, S., Bollt, E., Camps-Valls, G., Coumou, D., Deyle, E., van Nes, E. H.: Inferring causation from time series in Earth system sciences. Nature communications, 10(1), 1-13, 2019.

23. Lines 536-543. Really confusing. What is influencing what? TSAT or NHSAT?

**Response:** Thanks for your suggestions. The excepted meaning is that TSAT influences NHSAT, which can be explained by that the energy is transferred from the tropical climate system to the Northern Hemispheric climate system [1].

[1] Vallis, G. K., Farneti, R.: Meridional energy transport in the coupled atmosphere–ocean system: Scaling and numerical experiments. Q. J. Roy. Meteor. Soc., 135(644), 1643-1660, 2009.

We will improve the description as the following shows:

is nonlinear since their reconstructions are direction-dependent. (ii) The CCM index that NHSAT

cross maps TSAT is $\rho_{N \to T} = 0.70$, which means that amounts of information of TSAT is encoded in

NHSAT so that the NHSAT can be used to recover the values of TSAT. And the CCM index that

TSAT cross maps NHSAT is $\rho_{T \to N} = 0.24$, which means that the less information of NHSAT is encoded in TSAT. According to the CCM theory (see Appendix), the asymmetric CCM index indicates that the main energy transport direction might be mainly from the tropics to the Northern

Hemisphere. (iii) If there are enough historical measured series of the Northern hemisphere and

24. I have also noted many typographical errors, and the manuscript will benefit for a careful reading by the authors and by an English native speaker to rephrase some sentences.

**Response:** Thanks for your suggestions. We will carefully inspect the manuscript, and later than we will also invite a colleague of our field speaking native English to improve some sentences.

---

## Referee Comment (RC2) · Anonymous Referee #2 · 7 Feb 2020

This manuscript investigates the feasibility of using Machine Learning (ML) algorithm for the reconstruction of a time series with the help of a coupled time series. The study also examines the ability of an ML algorithm to represent the coupling strength of a system. The reconstruction analysis investigates three ML algorithms: Back Propagation (BP), Long Short-Term Memory (LSTM), and Reservoir Computing (RC). The study also investigates the influence of type of coupling (linear or non-linear) on the performance of ML algorithm. This is achieved by using a simple linear system, a simple non-linear system (Lorenz-63), a high-dimensional non-linear system (Lorenz-96), and a real-world system (coupling between Tropical surface air temperature and Northern Hemisphere surface air temperature). The linearity is measured using Pearson's correlation coefficient while the non-linearity is measure using Convergent Cross Map-

ping Causality index (CCM). The influence of the direction of coupling and coupling strength, and the number of explanatory variables on the accuracy of reconstruction of different ML algorithms is also examined. The performance evaluation of ML algorithms found that RC is most suitable for the reconstruction of non-linearly coupled time series. The work is scientifically sound and I see a lot of value in this work. Especially in the future applications of ML algorithms for reconstruction of coupled time series and in understanding the influence of coupling mechanisms on the behavior of ML algorithm. However, the presentation of the work in its current form is very confusing and diverts the attention of the reader from the importance of the work. The manuscript has errors related to English too which need to be corrected. Please find my major suggestions on the manuscript below.

The abstract talks about the reconstruction of a time series of a coupled system from its other coupled counter-parts. However, the introduction is not representing it intuitively. I would suggest the authors to focus on the problem of reconstruction of a time series and build the importance of coupling mechanism, importance of linear and non-linear coupling around the time series reconstruction. The Methodology section does not seem to have a description of BP and LSTM in it, in as much detail as stated for RC. I would suggest the authors to incorporate the description of BP and LSTM too, as it will help the readers to better understand the behavior of the algorithms. The CCM method has been introduced in the Results section. It should be introduced in the Methodology section. In the discussion of CCM method, relate it with the direction of reconstruction as well (explanatory variable to reconstructed variable), otherwise it is a little confusing to relate the notation of with its notation when it is being applied and shown in the Results section (Line number 462-463). The same goes for the description of Pearson's correlation coefficient, its description should be shifted from the Results to the Methodology section. The flow of the Results section is hard to follow. The Results section just lists the author's observations, from the Figures and Tables, and does not provide any insights into those observations. For example, line number 329 - 330 states that BP and LSTM* are not sensitive to non-linear coupling,

but no explanation is given as to why this is so. The authors should provide more insight into the observed behavior of the ML algorithms mentioned in the Results section. The conclusion section should be shortened.

Although the work is interesting and has a lot of future scope, the above concerns prevents me from recommending this work for publication in its current form. I hope the authors would incorporate the suggestions and rewrite the manuscript.

Specific Points:

Lines 43-46: The climate problems mentioned here are actually applications of climate data.

Lines 52-54: Re-write this sentences to make it intuitive. For example, this line: "...while the physics of systems is suggested for consideration" feels like it refers to the study by Watson, 2019, where neural network based algorithm is used to augment a physics based model to improve its performance. However, this is not clear from the text.

Lines 63-64: The statement infers that, since linear correlation is an intrinsic assumption of traditional statistical methods, cross-correlation analysis should be carried out for investigating the performance of ML algorithms. This is not a valid reasoning, as the approach of ML algorithms and traditional statistical methods are very different.

Lines 83-87: This part should be there in the Results section. However, this line can be modified to be a hypothesis the authors are trying to check.

Line 105: Typographical error: it should be "Learning" not "Leaning".

Figure 1: The big black arrow used to represent (3), is confusing in the sense that the reconstructed time series from the testing stage is being compared with the time series from the training stage. Which is not the case.

Lines 182-183: Mention clearly why an analysis of LSTM* reconstructed time series is required.

Lines 201-203: The introduction of the parameters, p, d, and q is not proper and causes confusion. Rewrite the sentence.

Lines 205-206: x(t) and the Gaussian noise () time series are the two time series being used for the coupled analysis. This has to be mentioned clearly in the text. This comment goes for all the cases of coupled time series being used (non-linear, higher order non-linear, real world scenario).

Lines 236-237: The time series are being standardized (mean is zero and standard deviation is one) before being used in the reconstruction analysis. Explain why are they standardized.

Lines 275-277: Incorporate the plots for LSTM* in Figure 3c and 3d.

Lines 286-297: The information about convergent cross mapping (CCM) should be introduced in the methodology section in detail. Are there other methods for estimating non-linear correlation or causality between two time-series. If so, why CCM was specifically used.

Lines 390-392: Explain the decrease in LSTM nRMSE with an increase in . As, this behavior is contradictory to the LSTM's nRMSE behavior in the other cases.

Lines 407-408: Explain how did the authors arrive at this statement. RC and LSTM performed better than LSTM* and BP in the linearly coupled system. And BP and LSTM* were not part of the analysis of the high dimensional lorenz-96 analysis. However, this statement can be the conclusion of this section, which shows the sensitivity of RC and LSTM to different coupling strength.

Lines 416-420: Examine LSTM for its behavior with change in $\theta$, like the one done for the behavior of LSTM*. This will probably give more insight into the behavior of LSTM*.

Line 430: Why is RC not sensitive to Pearson's correlation.

Figure 8: It is missing the R2 and p-value of LSTM. The behavior of LSTM should also

be evaluated in the same manner.

Lines 472-473: What do you mean by unstable variance, elaborate.

---

## Author Comment (AC4) · 5 Mar 2020

**Reply to the comments of Anonymous Referee #2:**

**The comments of Anonymous Referee #2:**

**1.** This manuscript investigates the feasibility of using Machine Learning (ML) algorithm for the reconstruction of a time series with the help of a coupled time series. The study also examines the ability of an ML algorithm to represent the coupling strength of a system. The reconstruction analysis investigates three ML algorithms: Back Propagation (BP), Long Short-Term Memory (LSTM), and Reservoir Computing (RC). The study also investigates the influence of type of coupling (linear or non-linear) on the performance of ML algorithm. This is achieved by using a simple linear system, a simple non-linear system (Lorenz-63), a high-dimensional non-linear system (Lorenz-96), and a real-world system (coupling between Tropical surface air temperature and Northern Hemisphere surface air temperature). The linearity is measured using Pearson's correlation coefficient while the non-linearity is measure using Convergent Cross Map ping Causality index (CCM). The influence of the direction of coupling and coupling strength, and the number of explanatory variables on the accuracy of reconstruction of different ML algorithms is also examined. The performance evaluation of ML algorithms found that RC is most suitable for the reconstruction of non-linearly coupled time series. The work is scientifically sound and I see a lot of value in this work. Especially in the future applications of ML algorithms for reconstruction of coupled time series and in understanding the influence of coupling mechanisms on the behavior of ML algorithm. However, the presentation of the work in its current form is very confusing and diverts the attention of the reader from the importance of the work. The manuscript has errors related to English too which need to be corrected. Please find my major suggestions on the manuscript below.

**Response:** Thanks for your thoughtful comments and suggestions! The suggestions are very helpful for improving our manuscript, and we will carefully revise the manuscript according to these suggestions.

**2.** The abstract talks about the reconstruction of a time series of a coupled system from its other coupled counter-parts. However, the introduction is not representing it intuitively. I would suggest the authors to focus on the problem of reconstruction of a time series and build the importance of coupling mechanism, importance of linear and non-linear coupling around the time series reconstruction.

**Response:** Thank you! In the introduction, we will focus more on the importance of coupling mechanism to the time series reconstruction, and the importance of linear and non-linear correlations. Some of our modification in the introduction is shown by the following screenshot:

| 68 | for them. Here, the coupling relation between different variables needs to be paid attention to. |
| 69 | Because different climate variables are coupled with one another (Donner and Large, 2008), and the |
| 70 | coupling relation will often result in that their observational time series are statistically correlated |
| 71 | (Brown, 1994). This is a crucial property for the climate system, and often contributes to the |
| 72 | analysis on the climatic time series. It is known that linear correlation is the implicit assumption for |
| 73 | traditional statistical methods, and they often fail if linear correlation is weak (Brown, 1994; |
| 74 | Sugihara et al., 2012; Emile-Geay and Tingley, 2016). However, previous studies (Sugihara et al., |
| 75 | 2012; Emile-Geay and Tingley, 2016) also suggest that, although the linear correlation of two |
| 76 | variables is potentially absent, they might be nonlinearly coupled and can be exploited by analysis. |
| 77 | For instance, the linear cross-correlations of sea surface temperature series observed in different |
| 78 | tropical areas are unstable and vary with time, which leads to an overall weak linear correlation, but |
| 79 | this non-linear correlation is conductive to the better El Niño predictions (Ludescher et al., 2014; |

| 88 | In a recent study (Lu et al., 2017), a machine learning framework was used to reconstruct the |
| 89 | unmeasured time series in the Lorenz 63 model (Lorenz, 1963). They found that $Z$ can be well |
| 90 | reconstructed from $X$, but it failed to reconstruct $Z$ from $X$. Lu et al. (Lu et al., 2017) demonstrated |
| 91 | that the nonlinear coupling dynamic between $X$ and $Z$ was responsible to this asymmetry in the |
| 92 | reconstruction. This was explained by the nonlinear observability in control theory (Hermann and |
| 93 | Krener, 1977; Lu et al., 2017): For the Lorenz 63 equation, both $(X(t), Y(t), Z(t))$ and $(-X(t), -Y(t),$ |
| 94 | $Z(t))$ could be its solutions. Hence, when $Z(t)$ was acting as an observer, it cannot distinguish $X(t)$ |
| 95 | from $-X(t)$, and the information content of $X$ was incomplete, which determined that $X$ cannot be |
| 96 | reconstructed by machine learning. However, nonlinear observability is often analyzed for the |
| 97 | nonlinear system with known equation (Hermann and Krener, 1977; Schumann-Bischoff et al., 2016; |
| 98 | Lu et al., 2017). For the observational records from a complex system without explicit equation, the |
| 99 | nonlinear observability might be hard to be analyzed. Furthermore, does this asymmetry in the |
| 100 | reconstruction also exist in other climatic time series which are nonlinear coupled? This is still an |
| 101 | open question. |

**3.** The Methodology section does not seem to have a description of BP and LSTM in it, in as much detail as stated for RC. I would suggest the authors to incorporate the description of BP and LSTM too, as it will help the readers to better understand the behavior of the algorithms.

**Response:** Thank you! We will add more detailed description of BP and LSTM into the revised manuscript.

But the algorithms of BP are much more complicated than that of RC, and there are too many equations (about 15 mathematical equations) for their algorithms so that the article will be not concise. We will carefully introduce the key steps for BP, and the relevant references will be cited for the steps.

**Especially, we will highlight the crucial differences in algorithms among RC, BP and LSTM, and this might be very helpful for understanding the application results of them.**

Our modification for the neural network algorithms are shown by the following screenshot:

**2.2.1 Reservoir computer**

[revised manuscript text omitted]

**4.** The CCM method has been introduced in the Results section. It should be introduced in the Methodology section. In the discussion of CCM method, relate it with the direction of reconstruction as well (explanatory variable to reconstructed variable)

**Response:** Thank you! We will add the description of the CCM algorithm into the method part of the revised manuscript, and also relate it with the direction of reconstruction. Our modification is shown by the following screenshot:

**2.4.2 Convergent cross mapping**

To measure the nonlinear coupling relation between two observational variables, we choose the convergent cross mapping method that has been demonstrated to be useful for many complex systems (Sugihara et al., 2012; Tsonis et al., 2018; Zhang et al. 2019). Considering $a(t)$ and $b(t)$ as two observational time series, we begin with the cross mapping (Sugihara et al., 2012) from $a(t)$ to $b(t)$ through the following steps:

i) Embedding $a(t)$ (with length $L$) into the phase space with the vector $M_a(t_i) = \{a_{t_i}, a_{t_i - \tau_0}, ..., a_{t_i - (m-1)\tau}\}$ ("$t_i$" represents a historical moment in the observations), where embedding dimension ($m$) and time delay ($\tau$) can be determined through the false nearest neighbor algorithm (Hegger and Kantz, 1999).

ii) Estimating the weight parameter $w_i$ denoting the associated weight between two vectors "$M_a(t)$" and "$M_a(t_i)$" ("$t$" denotes the excepted time in this cross mapping), defined as:

$$w_i = \frac{u_i}{\sum_{i=1}^{m+1} u_i}, \tag{7}$$

$$u_i = exp\left\{-\frac{d\,[M_a(t), M_a(t_i)]}{d\,[M_a(t), M_a(t_l)]}\right\}, \tag{8}$$

where $d\,[M_a(t), M_a(t_i)]$ denotes the Euler distance between vectors "$M_a(t)$" and "$M_a(t_i)$". The nearest neighbor to "$M_a(t)$" generally corresponds to the largest weight.

iii) Cross mapping the value of $b(t)$ by

$$\hat{b}(t) = \sum_{i=1}^{m+1} w_i b(t_i). \tag{9}$$

$\hat{b}(t)$ denotes the estimated value of $b(t)$ with this phase-space cross mapping. Then, we will evaluate the cross mapping skill (Sugihara et al., 2012; Tsonis et al., 2018) as the follows:

$$\rho_{a \to b} = corr.\,[b(t), \hat{b}(t)] \tag{10}$$

The cross mapping skill from $b$ to $a$ is also measured according to the above steps, marked as $\rho_{b \to a}$. *Sugihara et al.* and *Tsonis et al.* defined the causal inference from $\rho_{a \to b}$ and $\rho_{b \to a}$ like that: (i) if $\rho_{a \to b}$ is convergent when $L$ is increased, and $\rho_{a \to b}$ is of high value, then $b$ is suggested to be a causation of $a$. (ii) Besides, if $\rho_{b \to a}$ is also convergent when $L$ is increased, and is of high value, then the causal relationship between $a$ and $b$ is bidirectional ($a$ and $b$ cause each other). In our study, all the values of CCM indices are measured when they are convergent with the data length.

According to the literature (Takens, 1981; Sugihara et al., 2012), the CCM index is related to the ability of using one variable to reconstruct another variable: if $b$ influence $a$ but $a$ does not influence $b$, the information content of $b$ can be encoded in $a$ (through the information transfer from

$b$ to $a$), but the information content of $a$ is not encoded in $b$ (there exists no information transfer from $a$ to $b$. Hence, the time series of $b$ can be reconstructed from the records of $a$. For the CCM

index ($\rho_{a \to b}$), its magnitude represents how much information content of $b$ is encoded in the records of $a$. So that the high value of $\rho_{a \to b}$ means that $b$ causes $a$, and we can get good results of reconstruction from $a$ to $b$. In this paper, we can test the association between the CCM index and the reconstruction performance of machine learning.

5. Otherwise it is a little confusing to relate the notation of with its notation when it is being applied and shown in the Results section (Line number 462-463).

**Response:** Thank you! We will modify this narration, and improve such narration thoroughly in the revised manuscript. Our modification is shown by the following screenshot:

machine learning to reconstruct these climate series. The CCM index of that NHSAT cross maps

TSAT is 0.70, and the CCM index of that TSAT cross maps NHSAT is 0.24 (Table 4). The CCM

index means, that the information content of TSAT is well encoded in the records of NHSAT, and the information transfer might be mainly from TSAT to NHSAT, which is consistent with previous studies (Farneti and Vallis, 2013). Further, the CCM analysis indicates that the reconstruction from

NHSAT to TSAT might obtain a better quality than the opposite direction.

6. The same goes for the description of Pearson's correlation coefficient, its description should be shifted from the Results to the Methodology section.

**Response:** Thank you! We will move the description of Pearson's correlation to the method in the revised manuscript. Our modification is shown by the following screenshot:

**2.4.1 Linear correlation**

As the introduction mentioned, the linear Pearson correlation is a commonly-used method to quantify the linear relationship between two observational variables. The Pearson correlation between two series $a(t)$ and $b(t)$, is defined as

$$corr. = \frac{mean[(a - \bar{a}) \cdot (b - \bar{b})]}{std(a) \cdot std(b)}. \tag{6}$$

The symbols "*mean*" and "*std*" denote the average and standard deviation for series $a(t)$ and $b(t)$, respectively.

7. The flow of the Results section is hard to follow. **The Results section just lists the author's observations, from the Figures and Tables, and does not provide any insights into those observations. For example, line number 329 - 330 states that BP and LSTM\* are not sensitive to non-linear coupling, but no explanation is given as to why this is so.** The authors should provide more insight into the observed behavior of the ML algorithms mentioned in the Results section.

**Response:** Thank you! We will provide more insight into the observed behavior of the ML algorithms mentioned in the Results section. For the analysis on other results, we will also pay more attention to this.

For the results of that BP and LSTM\* are not sensitive to non-linear coupling, their algorithms might be responsible to this. **When analyzing their algorithm, we can find that the BP neural network cannot track the temporal evolution, because its neuron states are independent to the temporal variation of time series. For LSTM\*, it cannot include the information of previous time. Previous studies have revealed that the temporal evolution and memory are crucial properties for the nonlinear time series** [1, 2]**,** which should be considered when modeling nonlinear dynamics. But the algorithms of RC and LSTM have made improvements on these issues (we have added these contents into the method part of the revised manuscript).

[1] Kantz, H., Schreiber, T.: Nonlinear time series analysis (Vol. 7). Cambridge university press, 2004.

[2] Franzke C. L., Osprey, S. M., Davini, P., Watkins, N. W.: A dynamical systems explanation of the Hurst effect and atmospheric low-frequency variability. Sci. Rep., 5, 9068, 2015.

Our modification is shown by the following screenshot:

| | |
|---|---|
| 407 | As mentioned in the method, the BP neural network cannot track the temporal evolution, since |
| 408 | its neuron states are independent to the temporal variation of time series. For LSTM*, it cannot |
| 409 | include the information of previous time. These might be responsible to that BP and LSTM* fail in |
| 410 | dealing with nonlinear system. Previous studies have revealed that the temporal evolution and |
| 411 | memory are crucial properties for the nonlinear time series (Kantz and Schreiber, 2003; Franzke et |
| 412 | al. 2015), which should be considered when modeling nonlinear dynamics. |

8. The conclusion section should be shortened.

**Response:** Thanks for your suggestion. We will shorten the length of the conclusion, and move part of the discussion into the results part. Our modification for the conclusion is shown by the following screenshot:

| | |
|---|---|
| 591 | **5  Conclusions and discussions** |
| 592 | In this study, three kinds of machine learning frameworks are used to reconstruct the time |
| 593 | series of toy models and real-world climate systems. One series can be reconstructed from the other |
| 594 | series by machine learning when they are governed by the common coupling relation. For the linear |
| 595 | system, variables are coupled by the linear mechanism, and a strong Pearson correlation benefits to |
| 596 | machine learning with bi-directional reconstruction. For a nonlinear system, the time series often |
| 597 | have a weak Pearson correlation, but the machine learning can still well reconstruct the time series |
| 598 | when two variables share the common information through their interactions; moreover, the |
| 599 | reconstruction quality is direction-dependent and variable-dependent, which is determined by the |
| 600 | coupling strength and causality between the dynamical variables. |
| 601 | Considering the reconstruction quality dependency, selecting the suitable explanatory variables |
| 602 | is crucial for obtaining a good reconstruction quality. But the results show that machine learning |
| 603 | performance cannot be only explained by linear correlation. Hence, we propose using the CCM |
| 604 | index to select explanatory variables. Especially for the time series of nonlinear systems, when the |
| 605 | Pearson correlation is weak, the CCM index might be strong enough, and then the corresponding |
| 606 | variable can be selected as an explanatory variable. It is well known that atmospheric or oceanic |

variable can be selected as an explanatory variable. It is well known that atmospheric or oceanic motions are nonlinearly coupled over most of scales, and therefore, in the natural climate series, there would be similar nonlinear coupling relation to the Lorenz 63 and the Lorenz 96 systems (the linear correlation is weak but CCM indices are of high values). However, if only Pearson correlation is used to select the explanatory variable, then some useful nonlinearly correlated variables might be left out.

Finally, it is worth noting once more that there are still more potential applications for machine learning in the climate studies. For instance, a series $b(t)$ is unmeasured during some periods for the measuring instrument failure, but there are other kinds of variables without missing observations.

Moreover, CCM can be applied to select the suitable variables coupled with $b(t)$, and then the RC

can be employed to reconstruct the unmeasured part of $b(t)$ (following Fig. 1). This is very useful to some climate studies, such as paleoclimate reconstruction (Brown, 1994; Donner 2012; Emile-Geay and Tingley, 2016), interpolation for the missing points in measurements (Hofstra et al., 2008), and the parameterization schemes (Wilks, 2005; Vissio and Lucarini, 2018). Our study in this article is only a beginning for reconstructing climate series by machine learning, and more detailed investigations will be reported soon.

9. Although the work is interesting and has a lot of future scope, the above concerns prevents me from recommending this work for publication in its current form. I hope the authors would incorporate the suggestions and rewrite the manuscript.

**Response:** Thanks for your comments and suggestions! We will carefully improve the detail descriptions, and recognize most of parts according to your suggestions.

Specific Points:

10. Lines 43-46: The climate problems mentioned here are actually applications of climate data.

**Response:** Thank you! We will modify this narration. Our modification is shown by the following screenshot:

The application of climatic time series is important for climate research, such as paleoclimate reconstruction (Brown, 1994; Emile-Geay and Tingley, 2016), interpolation for the missing points in measurements (Hofstra et al., 2008), parameterization schemes (Wilks, 2005; Vissio and Lucarini,

2018), and seasonal climate prediction (Comeau et al., 2017; Wang et al., 2017). Neural

11. Lines 52-54: Re-write this sentences to make it intuitive. For example, this line: "...while the physics of systems is suggested for consideration" feels like it refers to the study by Watson, 2019, where neural network based algorithm is used to augment a physics based model to improve its performance. However, this is not clear from the text.

**Response:** Thank you! We will modify this narration. Our modification is shown by the following screenshot:

| | |
|---|---|
| 53 | Kratzert et al., 2019; Feng et al., 2019). Recently it is also demonstrated for the large potentials of |
| 54 | machine learning to simulate the temporal dynamics of complex systems (Pathak et al., 2017; Du et |
| 55 | al., 2017; Watson, 2019). Furthermore, some studies (Pathak et al., 2017; Lu et al., 2018) also |
| 56 | suggest to test whether the dynamical properties of the underlying system can be described by |
| 57 | machine learning, so that the machine learning application can be better understood. For example, |
| 58 | chaos is the key property of the underlying system giving rise to the climatic time series (Lorenz, |
| 59 | 1963; Patil et al., 2001), and then the results of applying machine learning to Lorenz system and |
| 60 | Rossler model show that their chaotic attractors are able to be well described (Pathak et al., 2017; Lu |
| 61 | et al., 2018; Carroll, 2018), which demonstrates the usability of machine learning on climatic series. |
| 62 | Further, we should also focus on how the dynamical properties of the system will influence the |
| 63 | performance of machine learning. |

12. Lines 63-64: The statement infers that, since linear correlation is an intrinsic assumption of traditional statistical methods, cross-correlation analysis should be carried out for investigating the performance of ML algorithms. This is not a valid reasoning, as the approach of ML algorithms and traditional statistical methods are very different.

**Response:** Thank you! We will modify this narration. Our modification is shown by the following screenshot:

Applying machine learning to climatic series attracts much attention, but it is still unclear what can be learnt by machine learning during the training process, and what is the key factor determining the performance of machine learning applied to climatic time series. This is crucial for investigating why machine learning performs not well with some datasets, and how to improve the performance for them. Here, the coupling relation between different variables needs to be paid attention to. Because different climate variables are coupled with one another (Donner and Large, 2008), and the coupling relation will often result in that their observational time series are statistically correlated (Brown, 1994). This is a crucial property for the climate system, and often contributes to the analysis on the climatic time series. It is known that linear correlation is the implicit assumption for traditional statistical methods, and they often fail if linear correlation is weak (Brown, 1994; Sugihara et al., 2012; Emile-Geay and Tingley, 2016). However, previous studies (Sugihara et al., 2012; Emile-Geay and Tingley, 2016) also suggest that, although the linear correlation of two variables is potentially absent, they might be nonlinearly coupled and can be exploited by analysis. For instance, the linear cross-correlations of sea surface temperature series observed in different tropical areas are unstable and vary with time, which leads to an overall weak linear correlation, but this non-linear correlation is conductive to the better El Niño predictions (Ludescher et al., 2014; Conti et al., 2017). The linear correlations between ENSO/PDO index and some proxy variables are weak but their nonlinear coupling relations can be detected, which contributes greatly to reconstructing longer paleoclimate time series (Mukhin et al., 2018). These studies indicate that nonlinear coupling relations can contribute to better analysis, reconstruction, and prediction (Hsieh et al., 2006; Donner, 2012; Schurer et al., 2013; Badin et al., 2014; Drótos et al., 2015; Van Nes et al., 2015; Comeau et al., 2017; Vannitsem and Ekelmans, 2018). Accordingly, when applying machine learning to climatic series, is it necessary to give attention to the linear or nonlinear relationships induced by the physical couplings? This is worth to be addressed.

13. Lines 83-87: This part should be there in the Results section. However, this line can be modified to be a hypothesis the authors are trying to check.

**Response:** Thank you! We will modify this narration. This part has been modified to be a hypothesis in the introduction. Our modification is shown by the following screenshot:

> 109       Moreover, we will also discuss a real-world example from climate system. It is known that
>
> 110       there exists coupling in the atmospheric motions between the tropics and the Northern Hemisphere,
>
> 111       which is through the transfer of atmospheric energy (Farneti and Vallis, 2013). Due to the
>
> 112       underlying complicated processes, it is difficult to use a formula to cover this coupling between the
>
> 113       tropical average surface air temperature (TSAT) series and the Northern Hemispheric surface air
>
> 114       temperature (NHSAT) series. We will employ machine learning to investigate whether the NHSAT
>
> 115       time series can be reconstructed from the TSAT time series, and whether the TSAT time series can
>
> 116       be also reconstructed from the NHSAT time series. Accordingly, the conclusions from our model
>
> 117       simulations can be further tested and generalized.

14. Line 105: Typographical error: it should be "Learning" not "Leaning".

**Response:** Thank you! We will modify this typographical error. We will also inspect the manuscript to avoid the any typographical error. Our modification is shown by the following screenshot:

> 124    **2.1   Learning coupling relations and reconstructing coupled time series**

15. Figure 1: The big black arrow used to represent (3), is confusing in the sense that the reconstructed time series from the testing stage is being compared with the time series from the training stage, which is not the case.

**Response:** Thank you! We will modify this figure. Our modification is shown by the following screenshot:

[Figure]

**Figure 1** Diagram illustration for reconstructing time series by machine learning. (1) The available part of the dataset $\{a_1(t), \ldots, a_n(t), b(t)\}$ is used to train the neural network ($a_1(t), \ldots, a_n(t)$ and $b(t)$ are the time series of the variables $a_1, \ldots, a_n, b$ ). So that the inherent coupling relation $F$ among these variables can be learnt by the neural network, and the learnt coupling relation is noted as $\hat{F}$. (2) $b(t')$ is unknown, but the dataset $\{a_1(t'),$

$a_2(t'), \ldots, a_n(t')\}$ is available which is input into the trained neural network, and the unknown series $b(t')$ can be reconstructed, denoted as $\hat{b}(t')$. (3) If $\hat{b}(t') \approx b(t')$, then $\hat{F} \approx F$ can be derived, and it indicates that the machine learning framework have learnt the intrinsic coupling relation.

16. Lines 182-183: Mention clearly why an analysis of LSTM* reconstructed time series is required.

**Response:** Thank you! We will modify this narration.

The crucial improvement of LSTM on the traditional recurrent neural network, is that LSTM has the **forget gate** which controls the information of the previous time to flow into the neural network. This also make the neural state of LSTM has ability to track the temporal evolution, which is also the crucial difference between LSTM and BP neural networks.

Here, we also test the LSTM neural network **without the forget gate, and call it LSTM\***. This means that the information of the previous time cannot flow into the LSTM* neural network, which does not have the memory for the past information. **We will compare the performance of LSTM with that of LSTM\*, so that the role of the neural network memory for the previous information can be demonstrated.**

Our modification is shown by the following screenshot:

| | |
|---|---|
| 217 | The crucial improvement of LSTM on the traditional recurrent neural network, is that LSTM |
| 218 | has the forget gate which controls the information of the previous time to flow into the neural |
| 219 | network. This will make the neural state of LSTM has ability to track the temporal evolution of time |
| 220 | series (Chattopadhyay et al., 2019; Kratzert et al., 2019), which is also the crucial difference |
| 221 | between the LSTM and the BP neural networks. |
| 222 | Here, we also test the LSTM neural network without the forget gate, and call it LSTM$^*$. This |
| 223 | means that the information of the previous time cannot flow into the LSTM$^*$ neural network, which |
| 224 | does not have the memory for the past information. We will compare the performance of LSTM |
| 225 | with that of LSTM$^*$, so that the role of the neural network memory for the previous information can |
| 226 | be demonstrated. |

17. Lines 201-203: The introduction of the parameters, p, d, and q is not proper and causes confusion. Rewrite the sentence.

**Response:** Thank you! We will modify this narration. Our modification is shown by the following screenshot:

**A linearly coupled model:** The autoregressive fractionally integrated moving average (ARFIMA) model (Granger and Joyeux, 1980) maps a Gaussian white noise $\varepsilon(t)$ into a correlated sequence $x(t)$ (Eq. (11)), which could simulate the linear dynamics of oceanic-atmospheric coupled system (Hasselmann, 1976; Franzke, 2012; Massah and Kantz, 2016; Cox et al., 2018).

$\varepsilon(t) \xrightarrow{ARFIMA(p,d,q)} x(t)$                   (11)

In this model, $d$ is a fractional differencing parameter, and $p$ and $q$ are the orders of the autoregressive and moving average components. Here, the parameters are set as: $p = 3$, $d = 0.2$ and $q$

$= 3$. Hence $x(t)$ is a time series composited with three components: the third-order autoregressive process whose coefficients are 0.6, 0.2 and 0.1, the fractional differencing process whose Hurst exponent is 0.7, and the third-order moving average process whose coefficients are 0.3, 0.2 and 0.1

(Granger and Joyeux, 1980). These two time series $\varepsilon(t)$ and $x(t)$ will be used for the reconstruction analysis.

18. Lines 205-206: x(t) and the Gaussian noise () time series are the two time series being used for the coupled analysis. This has to be mentioned clearly in the text. This comment goes for all the cases of coupled time series being used (non-linear, higher order non-linear, real world scenario).

**Response:** Thank you! We will mention this information for all the used data in the revised manuscript. Our modification is shown by the following screenshot:

**A linearly coupled model:** The autoregressive fractionally integrated moving average (ARFIMA) model (Granger and Joyeux, 1980) maps a Gaussian white noise $\varepsilon(t)$ into a correlated sequence $x(t)$ (Eq. (11)), which could simulate the linear dynamics of oceanic-atmospheric coupled system (Hasselmann, 1976; Franzke, 2012; Massah and Kantz, 2016; Cox et al., 2018).

$\varepsilon(t) \xrightarrow{ARFIMA(p,d,q)} x(t)$                  (11)

In this model, $d$ is a fractional differencing parameter, and $p$ and $q$ are the orders of the autoregressive and moving average components. Here, the parameters are set as: $p = 3$, $d = 0.2$ and $q$

$= 3$. Hence $x(t)$ is a time series composited with three components: the third-order autoregressive process whose coefficients are 0.6, 0.2 and 0.1, the fractional differencing process whose Hurst exponent is 0.7, and the third-order moving average process whose coefficients are 0.3, 0.2 and 0.1

(Granger and Joyeux, 1980). These two time series $\varepsilon(t)$ and $x(t)$ will be used for the reconstruction analysis.

**A nonlinearly coupled model:** The Lorenz 63 (in the following referred as to Lorenz63)

chaotic system (Lorenz, 1963) depicts the nonlinear coupling relation in a low-dimensional chaotic system. The system reads

$$\frac{dx}{dt} = -\sigma(x - y)$$
$$\frac{dy}{dt} = \mu x - xz - y \tag{12}$$
$$\frac{dz}{dt} = xy - Bz$$

When the parameters are fixed at $(\sigma, \mu, B) = (10, 28, 8/3)$, the state in the system is chaotic. We employed the Runge-Kutta integrator of the fourth order to acquire the series output from Lorenz63.

The time steps were 0.01. The time series $X(t)$ and $Z(t)$ will be used for the reconstruction analysis.

**A high-dimensional model:** The two-layer Lorenz 96 (in the following referred as to

Lorenz96) model (Lorenz, 1996) is a high-dimensional chaotic system, which is generally employed to mimic mid-latitude atmospheric dynamics (Chorin and Lu, 2015; Hu and Franzke, 2017; Vissio and Lucarini, 2018; Chen and Kalnay, 2019; Watson, 2019). It reads

$$\frac{dX_k}{dt} = X_{k-1}(X_{k+1} - X_{k-2}) - X_k + F - \frac{h_1}{J}\sum_{j=1}^{J} Y_{j,k}$$
$$\frac{dY_{k,j}}{dt} = \frac{1}{\theta}[Y_{k,j+1}(Y_{k,j-1} - Y_{k,j+2}) - Y_{k,j} + h_2 X_k]. \tag{13}$$

In the first layer of the Lorenz 96 there are 18 variables marked as $X_k$ ($k$ is a integer ranging from 1

to 18), and each $X_k$ is coupled with $Y_{k,j}$ ($Y_{k,j}$ is from the second layer). The parameters are set as fellows: $J = 20$, $h_1 = 1$, $h_2 = 1$, and $F=10$. The scale parameter $\theta$ controls the scale separation of the two layers. When $\theta > 1$, processes in the second layer will be slower than processes in the first layer because the increment of $Y_{k,j}$ is decreased by the term of $\theta$. The time scale of $Y_{k,j}$ can be also close to that of $X_k$ by modulating the value of $\theta$; especially, the coupling strength will be amplified when $\theta$ is much smaller than 1. The Runge-Kutta integrator of the fourth order and periodic boundary condition are adopted (that is: $X_0 = X_K$ and $X_{K+1} = X_1$; $Y_{k,0} = Y_{k-1,J}$ and $Y_{k,J+1} = Y_{k+1,1}$), and the integral time unit was taken as 0.05. The time series $X_1(t)$ and $Y_{1,1}(t)$ will be used for the reconstruction analysis.

**3.2   Real-world climatic time series**

TSAT, NHSAT and the Nino3.4 index are chosen to represent real-world climatic time series, and they will be used for the reconstruction analysis. The original data is from National Centers for

Environmental Prediction (https://www.esrl.noaa.gov/psd/data/gridded/data.ncep.reanalysis2.html)

and KNMI Climate Explorer (http://climexp.knmi.nl). The series of TSAT and NHSAT are from the

19. Lines 236-237: The time series are being standardized (mean is zero and standard deviation is one) before being used in the reconstruction analysis. Explain why are they standardized.

**Response:** Thank you! We will explain for this processing of standardization.

For the time series that come from different processes, they might have different variability and units. In order to avoid the disturbance given by such different variability and units, we select to standardize all the time series with uniform mean value and variance.

Our modification is shown by the following screenshot:

| | |
|---|---|
| 228 | To evaluate the quality of reconstruction by machine learning, the root mean squared error |
| 229 | (RMSE) of residual series (Hyndman and Koehler, 2006) is adopted (Eq. (4)), which represents the |
| 230 | difference between the real series $b(t')$ and the reconstructed series $\hat{b}(t')$. In order to fairly |
| 231 | compare the errors of reconstructing different processes with different variability and units |
| 232 | (Hyndman and Koehler, 2006; Pennekamp et al., 2018), we will normalize the RMSE as Eq. (5) |
| 233 | shows. |

$$RMSE = \sqrt{\frac{1}{k}\sum_t [b(t') - \hat{b}(t')]^2}, \qquad (4)$$

$$nRMSE = \frac{RMSE}{\max[b(t')] - \min[b(t')]}. \qquad (5)$$

| | |
|---|---|
| 324 | **Training and testing datasets:** Before analysis, all the used time series are standardized to |
| 325 | take zero mean and unit variance so that different cases can be fairly compared (Hyndman and |
| 326 | Koehler, 2006). We divide the total series into two parts: 60% of the time series training the neural |
| 327 | network and 40% being the testing series. Specific data lengths of the training series and testing |
| 328 | series will be also listed in the results section. |

20. Lines 275-277: Incorporate the plots for LSTM* in Figure 3c and 3d.

**Response:** Thank you! We will add the results of LSTM$^*$ into the corresponding figures. Our modification is shown by the following screenshot:

[Figure]

**Figure 4** (a) The $x(t)$ time series (blue) and the $\varepsilon(t)$ time series (black) of the ARFIMA(3,0.2,3) model. White lines depict the large-scale trends of these time series acquired by 50-step smoothing average. (b) Comparison of the power spectrum of $x(t)$ (blue) with the power spectrum of $\varepsilon(t)$ (black). (c) Comparison of the reconstructed time series of $x(t)$ by RC, LSTM, LSTM$^*$ and BP respectively (red dots), and the original $x(t)$ time series are presented by the blue lines. (d) Comparison of the reconstructed time series of $\varepsilon(t)$ by RC, LSTM, LSTM$^*$ and BP

respectively (red dots), and the original $\varepsilon(t)$ time series are presented by the black lines. Only partial segments of the reconstructed series are shown.

21. Lines 286-297: The information about convergent cross mapping (CCM) should be introduced in the methodology section in detail. Are there other methods for estimating non-linear correlation or causality between two time-series. If so, why CCM was specifically used.

**Response:** Thank you! We will move the detailed description of CCM to the method part.

Apart from CCM, the Granger method [1] and transfer entropy [2] can be also used to measure the causality. However, it has been demonstrated that the Granger causality cannot measure the causality or coupling in nonlinear systems [3]. Transfer entropy can be an alternative choice to measure the nonlinear coupling. But the index value of transfer entropy often ranges from 0 to 3 [4], while the CCM index always ranges from 0 to 1, so that it is often hard to judge if transfer entropy is strong or weak. In previous studies [5], the CCM index has been successfully used to measure the nonlinear coupling strength and causality in many kinds of complex systems. However, it is worth to make comparisons for CCM, transfer entropy and machine learning performance in the future study.

[1] Granger C. W.: Investigating causal relations by econometric models and cross-spectral methods. Econometrica 37, 424-438, 1969.

[2] Schreiber T.: Measuring information transfer. Phys Rev Lett 85(2), 461, 2000.

[3] Malevergne Y., Sornette D.: Extreme financial risks: From dependence to risk management. Springer Science & Business Media, 2006.

[4] Paluš, M.: Multiscale atmospheric dynamics: cross-frequency phase-amplitude coupling in the air temperature. Phys Rev Lett, 112(7), 078702, 2014.

[5] Tsonis A. A., Deyle E. R., Ye H., Sugihara G.: Convergent cross mapping: theory and an example. In Advances in Nonlinear Geosciences (pp. 587-600). Springer, Cham, 2018.

Our modification is shown by the following screenshot:

**2.4.2 Convergent cross mapping**

To measure the nonlinear coupling relation between two observational variables, we choose the convergent cross mapping method that has been demonstrated to be useful for many complex systems (Sugihara et al., 2012; Tsonis et al., 2018; Zhang et al. 2019). Considering $a(t)$ and $b(t)$ as two observational time series, we begin with the cross mapping (Sugihara et al., 2012) from $a(t)$ to

$b(t)$ through the following steps:

i) Embedding $a(t)$ (with length $L$) into the phase space with the vector

$M_a(t_i) = \{a_{t_i}, a_{t_i - \tau_0}, \ldots, a_{t_i - (m-1)\tau}\}$ ("$t_i$" represents a historical moment in the observations), where embedding dimension ($m$) and time delay ($\tau$) can be determined through the false nearest neighbor algorithm (Hegger and Kantz, 1999).

ii) Estimating the weight parameter $w_i$ denoting the associated weight between two vectors "$M_a(t)$"

and "$M_a(t_i)$" ("$t$" denotes the excepted time in this cross mapping), defined as:

$$w_i = \frac{u_i}{\sum_{i=1}^{m+1} u_i},$$ (7)

$$u_i = exp\{-\frac{d[M_a(t), M_a(t_i)]}{d[M_a(t), M_a(t_1)]}\},$$ (8)

where $d[M_a(t), M_a(t_i)]$ denotes the Euler distance between vectors "$M_a(t)$" and "$M_a(t_i)$". The nearest neighbor to "$M_a(t)$" generally corresponds to the largest weight.

iii) Cross mapping the value of $b(t)$ by

$$\hat{b}(t) = \sum_{i=1}^{m+1} w_i b(t_i).$$ (9)

$\hat{b}(t)$ denotes the estimated value of $b(t)$ with this phase-space cross mapping. Then, we will evaluate the cross mapping skill (Sugihara et al., 2012; Tsonis et al., 2018) as the follows:

$$\rho_{a \to b} = corr. [b(t), \hat{b}(t)]$$ (10)

The cross mapping skill from $b$ to $a$ is also measured according to the above steps, marked as $\rho_{b \to a}$.

*Sugihara et al.* and *Tsonis et al.* defined the causal inference from $\rho_{a \to b}$ and $\rho_{b \to a}$ like that: (i) if

$\rho_{a \to b}$ is convergent when $L$ is increased, and $\rho_{a \to b}$ is of high value, then $b$ is suggested to be a causation of $a$. (ii) Besides, if $\rho_{b \to a}$ is also convergent when $L$ is increased, and is of high value, then the causal relationship between $a$ and $b$ is bidirectional ($a$ and $b$ cause each other). In our study, all the values of CCM indices are measured when they are convergent with the data length.

According to the literature (Takens, 1981; Sugihara et al., 2012), the CCM index is related to the ability of using one variable to reconstruct another variable: if $b$ influence $a$ but $a$ does not influence $b$, the information content of $b$ can be encoded in $a$ (through the information transfer from

$b$ to $a$), but the information content of $a$ is not encoded in $b$ (there exists no information transfer from $a$ to $b$). Hence, the time series of $b$ can be reconstructed from the records of $a$. For the CCM

index ($\rho_{a \to b}$), its magnitude represents how much information content of $b$ is encoded in the records of $a$. So that the high value of $\rho_{a \to b}$ means that $b$ causes $a$, and we can get good results of reconstruction from $a$ to $b$. In this paper, we can test the association between the CCM index and the reconstruction performance of machine learning.

22. Lines 390-392: Explain the decrease in LSTM nRMSE with an increase in CCM. As, this behavior is contradictory to the LSTM's nRMSE behavior in the other cases.

**Response:** Thank you! We will supplement the explanation for this.

For all cases of RC results, when the CCM index is increasing, the nRMSE will be decreasing. Likewise, for most cases of LSTM results, when the CCM index is increasing, the nRMSE will be decreasing.

But in this case for LSTM, the relation between CCM and nRMSE is not like the normal cases. The reason might be that the used time series ($X_1$ and $X_2$ of Lorenz 96 system) have the time-varying local mean values (i. e. in the previous time period, the local mean value of time series is 0, and then in the next time period, the local mean value of time series is 0.5), and this influences the performance of LSTM.

We found that the time-varying mean values in time series tend to impact the performance of LSTM. For example, in a time series, at the previous time period, the local mean value of time series is 0, and then at the next time period, the local mean value of time series is 0.5. In this case, LSTM tends to perform badly, and the nRMSE might be increased. **The reason might be that the LSTM algorithm always requires incorporating the time-series values at previous time points (the memory for past time points), and then the varied local mean value of time series will easily influence the results of LSTM.**

However, we have not been able to ensure that this is the only reason. More investigations are needed in the further study. Our modification is shown by the following screenshot:

| 467 | The reconstruction between $X_1$ and $X_2$ in the same layer of Lorenz 96 is also shown. There is an |
|---|---|
| 468 | asymmetric causal relation ($\rho_{X_2 \to X_1} = 0.37$ and $\rho_{X_1 \to X_2} = 0.25$) between $X_1$ and $X_2$, and their linear |
| 469 | correlation is very weak (see Table 3). The RC gives better result of reconstructing $X_1$ from $X_2$ |
| 470 | (nRMSE=0.13) than reconstructing $X_2$ from $X_1$ (nRMSE=0.17). LSTM also has different results for |
| 471 | $X_1$ and $X_2$ (Table 3), where the quality of reconstructing from $X_1$ to $X_2$ (nRMSE=0.16) is better than |
| 472 | reconstructing from $X_2$ to $X_1$ (nRMSE=0.20). The reconstruction quality of LSTM is worse than the |
| 473 | RC, and the reconstruction results by LSTM are not consistent with the coupling strengths. This |
| 474 | might indicate that LSTM will perform worse in some cases than RC, the reason for this needs to be |
| 475 | further investigated in future study. |

24. Lines 407-408: Explain how did the authors arrive at this statement. RC and LSTM performed better than LSTM* and BP in the linearly coupled system. And BP and LSTM* were not part of the analysis of the high dimensional lorenz-96 analysis. However, this statement can be the conclusion of this section, which shows the sensitivity of RC and LSTM to different coupling strength.

**Response:** Thank you! We will modify this narration. In our previous manuscript, the expected meaning of this statement was not a conclusion, but was used to open the topic of this subsection. Our modification for this part is shown by the following screenshot:

**4.2.2 The association between reconstruction quality and coupling strength**

Now, we further investigate when the dynamical coupling strength is altered, how the
reconstruction quality of different neural networks is influenced.

The setting of Eq. (13) is as follows: the value of $h_I$ is set as 0, and the value of $\theta$ is decreased
from 0.7 to 0.3. When $\theta$ is equal to 0.7, the forcing from $X_I$ to $Y_{I,I}$ is weak. At that time, the
Pearson correlation between $X_I$ and $Y_{I,I}$ is only 0.48, and the performances of BP and LSTM* are
not good. When $\theta$ is equal to 0.3, the forcing is dramatically magnified. As the second panel of Fig.
8a shows, this strong forcing makes $Y_{j,i}$ synchronized to $X_i$, and the linear correlation between $X_I$ and
$Y_{I,I}$ is greatly increased to 0.8. When the forcing strength is magnified, the performance of machine
learning is also enhanced (Fig. 8b): the reconstructed series from BP and the reconstructed series
from LSTM* are much closer to the real target series. This means, that the reconstruction quality of
BP and LSTM* is sensitive to the linear correlation, and it is greatly improved when the linear
correlation is increased.

For RC and LSTM, their results are different from BP and LSTM*. When $\theta$ is equal to 0.7 and
0.3, the values of CCM index are 0.91 and 0.98 respectively. Then, it can be found that the quality
of reconstructed $X_I$ by RC is always good. As Fig. 8b shows, although the Pearson correlation has
been changed a lot, the reconstructed series of RC always overlap with the real target series. In this
experiment, the results of LSTM are almost the same as that of RC (not shown here). It is known
that the linear Pearson correlation cannot explain the true dynamical relation in a nonlinear coupled
system (Sugihara et al., 2012). As the method mentioned, the RC and LSTM can track the temporal
evolution and memory of the time series, and then they might rely on the nonlinear dynamics rather
than the Pearson correlation.

25. Lines 416-420: Examine LSTM for its behavior with change in θ, like the one done for the behavior of LSTM*. This will probably give more insight into the behavior of LSTM*.

**Response:** Thank you! In this case of reconstructing $X_1$ from $Y_{1,1}$ (Lorenz 96 system), all the results of LSTM and RC are almost overlapped with each other. We will supplement the results of LSTM in the revised manuscript.

Our modification for this part is shown by the following screenshot:

| 506 | For RC and LSTM, their results are different from BP and LSTM$^*$. When $\theta$ is equal to 0.7 and |
|---|---|
| 507 | 0.3, the values of CCM index are 0.91 and 0.98 respectively. Then, it can be found that the quality |
| 508 | of reconstructed $X_1$ by RC is always good. As Fig. 8b shows, although the Pearson correlation has |
| 509 | been changed a lot, the reconstructed series of RC always overlap with the real target series. In this |
| 510 | experiment, the results of LSTM are almost the same as that of RC (not shown here). It is known |
| 511 | that the linear Pearson correlation cannot explain the true dynamical relation in a nonlinear coupled |
| 512 | system (Sugihara et al., 2012). As the method mentioned, the RC and LSTM can track the temporal |
| 513 | evolution and memory of the time series, and then they might rely on the nonlinear dynamics rather |
| 514 | than the Pearson correlation. |

26. Line 430: Why is RC not sensitive to Pearson's correlation.

**Response:** Thank you! Here the RC was applied to the nonlinear Lorenz 96 system. It is known that the linear Pearson correlation cannot explain the true dynamical relation in a nonlinear coupled system [1-2]. As the method mentioned, the RC and LSTM can track the temporal evolution and memory of the time series, and then they might rely on the nonlinear dynamics rather than the Pearson correlation.

[1] Malevergne Y., Sornette D.: Extreme financial risks: From dependence to risk management. Springer Science & Business Media, 2006.

[2] Sugihara, G., May, R., Ye, H., Hsieh, C. H., Deyle, E., Fogarty, M., Munch, S.: Detecting causality in complex ecosystems. Science, 338(6106), 496-500, 2012.

We will add some words to explain such phenomenon, which is shown by the following screenshot:

| 506 | For RC and LSTM, their results are different from BP and LSTM*. When $\theta$ is equal to 0.7 and |
|---|---|
| 507 | 0.3, the values of CCM index are 0.91 and 0.98 respectively. Then, it can be found that the quality |
| 508 | of reconstructed $X_1$ by RC is always good. As Fig. 8b shows, although the Pearson correlation has |
| 509 | been changed a lot, the reconstructed series of RC always overlap with the real target series. In this |
| 510 | experiment, the results of LSTM are almost the same as that of RC (not shown here). It is known |
| 511 | that the linear Pearson correlation cannot explain the true dynamical relation in a nonlinear coupled |
| 512 | system (Sugihara et al., 2012). As the method mentioned, the RC and LSTM can track the temporal |
| 513 | evolution and memory of the time series, and then they might rely on the nonlinear dynamics rather |
| 514 | than the Pearson correlation. |
| 515 | Considering the CCM index can be used to estimate the true coupling relation in a nonlinear |
| 516 | system (Sugihara et al. 2012; Tsonis et al. 2018), now we employing the CCM index to reveal the |
| 517 | association between the performances of RC/ LSTM and coupling strength. The values of CCM |
| 518 | index are calculated between $X_1$ and $X_2$, $X_3$ …, $X_{18}$; meanwhile, $X_1$ is reconstructed from $X_2$, $X_3$ …, |
| 519 | $X_{18}$, respectively. Then, a significant correspondence exists between the nRMSE and CCM index |
| 520 | (Fig. 9), especially for the results of RC. This indicates that the reconstruction quality is dependent |
| 521 | on the coupling strength between the reconstructed variable and different explanatory variables. |

27. Figure 8: It is missing the R2 and p-value of LSTM. The behavior of LSTM should also be evaluated in the same manner.

**Response:** Thank you! We will add the results of LSTM into this figure. Our modification is shown by the following screenshot:

[Figure]

**Figure 9** Scatter plot of nRMSE values and CCM index values. The blue boxes are results of the RC machine learning, and the black cycles are results of the LSTM machine learning. The blue and grey dashed lines are the fitted linear trends of the blue boxes and black cycles respectively, and these two dependency trends are both significant because their p-values are both smaller than 0.05.

28. Lines 472-473: What do you mean by unstable variance, elaborate.

**Response:** Thank you! We will supplement the explanation for this.

For the real-world time series (such as the time series in figure R1), the local mean value and the local variance of the time series, are often time-varying. For example, in a time series, at the previous time period, the local mean value of time series is 0, and then at the next time period, the local mean value of time series is 0.5; at the previous time period, the local variance of time series is 1, and then at the next time period, the local variance of time series is 1.5.

[Figure]

Figure R1: Daily time series of the Tropical surface air temperature, the Northern Hemispheric surface aire temperature, and the Nino 3.4 index.

We found that the time-varying local mean value and local variance in time series tend to impact the performance of LSTM. In this case, LSTM tends to perform badly, and the nRMSE might be increased.

**The reason might be that the LSTM algorithm always requires incorporating the time-series values in previous time points (the memory for past time points), and then the varied local mean value of time series will easily influence the results of LSTM. Likewise, the varied local mean value of time series will also influence the results of LSTM.**

However, we have not been able to ensure that this is the only reason. More investigations are needed in the future study. Our modification in this part is shown by the following screenshot:

| | |
|---|---|
| 550 | By means of RC machine learning, TSAT can be described by the reconstructed time series |
| 551 | (Fig. 11a). But the corresponding nRMSE is equal to 0.13, this is because some extremes of the |
| 552 | TSAT time series have not been described (Fig. 11b). When using TSAT to reconstruct the time |
| 553 | series of NHSAT, the reconstructed time series cannot describe the original time series of NHSAT |
| 554 | (Fig. 11c), and the corresponding nRMSE is equal to 0.21. Besides, we also use the LSTM and BP |
| 555 | to reconstruct these natural climate series, the performances of these two neural networks are worse |
| 556 | than RC (Table 4). For BP, this might be due to its inability to deal with nonlinear coupling (As |
| 557 | mentioned in method, the BP neurons cannot track the temporal evolution of time series). As for that |
| 558 | LSTM performs worse than RC in this real-world case, the reason needs to be further investigated in |
| 559 | future study. |

---

## Author Response (AR1)

**Reply to the Editor's comments**

**Editor's comments to the Author:**

The reviewers are in agreement with the scientific soundness of the present study; however, all the reviewers have raised serious concerns such as improper presentation of hypothesis, methodology and results section, which makes it difficult to follow. Authors may revise the manuscript, taking into consideration these comments. In addition, after going through the manuscript myself, I have a few more comments.

**Reply:** Many thanks for your comments and suggestions. The three reviewers provided very detailed suggestions for us to improve the presentation, and all of their comments and suggestions were incorporated when we revised the manuscript. We have thoroughly modified our manuscript. Also, our colleagues helped us to check and improve presentation, and we have added our thanks for their help into the acknowledgement.

We also modified the manuscript according to your comments. In the following, we would like to reply to your comments in details.

(i) The title of the manuscript highlights the usage of machine learning algorithms, in general, for the reconstruction of time series. While, I agree that not all ML algorithms can be considered/compared in one study, authors may dilute the claim made, since the present study focuses on only three ML algorithms. Also, the selection of these three algorithms may be justified, while revising the manuscript – why possibly these three among the vast variety of M algorithms available?

**Reply:** Thank you! We would like to revise the title of this manuscript, so that the topic can be more specific. There are many variants of machine learning methods, and in our work we only investigate three commonly-used methods of them. Our modification is as the following screenshot shows:

**Reconstructing coupled time series in climate systems using three kinds of**

**machine learning methods**

Yu Huang[1], Lichao Yang[1], Zuntao Fu[1]*

For the selection of these three machine learning methods, we were inspired by several recent studies on climatic time series, and their results suggested that these three methods are more applicable to sequential data like climate time series. We have modified this in the revised manuscript, as the following screenshot shows:

questions can be addressed. There are several variants of machine learning methods (Reichstein et al., 2019), and recent studies (Lu et al., 2017; Reichstein et al., 2019; Chattopadhyay et al., 2019)

suggest that three of them are more applicable to sequential data like time series: reservoir computer (RC), back propagation based artificial neural network (BP), and long short-term memory (LSTM)

neural network. Here we adopt these three methods to carry out our study, and provide a (ii)   It is interesting to read about the applicability of CCM, in determining the independent and reconstructed variables. Authors may explain a bit more about the statistics behind that. Is there a suggested cutoff value of CCM, for the reconstruction in any direction to be considered or neglected?

**Reply:** Thank you! The explanation for CCM is helpful and necessary for our manuscript. We have added detailed explanations in the method, and we also explain the meaning of CCM for every application example in the result section. Our modification is as following screenshots show:

| 273 | According to literature (Sugihara et al., 2012; Ye et al., 2015), the CCM index is related to the |
| --- | --- |
| 274 | ability of using one variable to reconstruct another variable: if $b$ influence $a$ but $a$ does not influence |
| 275 | $b$, the information content of $b$ can be encoded in $a$ (through the information transfer from $b$ to $a$), |
| 276 | but the information content of $a$ is not encoded in $b$ (there exists no information transfer from $a$ to $b$). |
| 277 | Therefore, the time series of $b$ can be reconstructed from the records of $a$. For the CCM index |
| 278 | ($\rho_{a \to b}$), its magnitude represents how much information content of $b$ is encoded in the records of $a$. |
| 279 | Therefore, the high magnitude of $\rho_{a \to b}$ means that $b$ causes $a$, and we can get good results of |
| 280 | reconstruction from $a$ to $b$. In this paper, we will test the association between the CCM index and the |
| 281 | reconstruction performance of machine learning. |

| 391 | In a nonlinear coupled system, it is known that the coupling strength between two variables |
| --- | --- |
| 392 | cannot be estimated by the linear Pearson correlation (Brown, 1994; Sugihara et al., 2012). Here, we |
| 393 | use CCM to estimate the coupling strength between $X$ and $Z$, and then it shows a high magnitude of |
| 394 | the CCM index: $\rho_{X \to Z} = 0.91$. According to the CCM theory (see Method), such a high magnitude |
| 395 | of the CCM index indicates that the information content of $Z$ is encoded in the time series of $X$. |
| 396 | Therefore, we conjecture that: when inputting $X$ to the neural network, not only the information |
| 397 | content of $X$, but also the information content of $Z$ can be learned by the neural network. And then it |
| 398 | is possible to reconstruct $Z$ from the trained neural network. We will test it in the following. |

The daily NHSAT and TSAT time series are shown in Fig. 10a. It is quite different for the oscillation shapes of the NHSAT and TSAT series, and there is a weak linear correlation (0.08, see

Table 4) between them. In the scatter plot for the NHSAT and TSAT (Fig. 10b), the marked nonlinear structure is observed between NHSAT and TSAT. Such a weak linear correlation will make the linear regression model fail to reconstruct one series from the other. Likewise, there is no explicit physical expression that can transform TSAT and NHSAT to each other. Now we try to use machine learning to reconstruct these climate series. The CCM index of that NHSAT cross maps

TSAT is 0.70, and the CCM index of that TSAT cross maps NHSAT is 0.24 (Table 4). The CCM

index means, that the information content of TSAT is well encoded in the records of NHSAT, and the information transfer might be mainly from TSAT to NHSAT, which is consistent with previous studies (Farneti and Vallis, 2013). Further, the CCM analysis indicates that the reconstruction from

NHSAT to TSAT might obtain a better quality than the opposite direction.

For the reconstruction in any direction to be considered or neglected, we could reasonably define a suggested cutoff value of CCM. Previous studies often suggest that the CCM index higher than 0.5 may be a strong enough magnitude. Also, when the CCM index is higher than 0.5, it is observed that the nRMSE is often smaller 0.1, where the reconstructed series has been very close to the real series in the presented results. Therefore, the CCM index that is higher than 0.5 may be considered for selecting explanatory variables. Our modification is as following screenshot shows:

Considering the reconstruction quality dependency, selecting the suitable explanatory variables is crucial for obtaining a good reconstruction quality. But the results show that machine learning performance cannot be only explained by linear correlation. Hence, we propose using the CCM

index to select explanatory variables. Especially for the time series of nonlinear systems, when the

CCM index is strong enough, the corresponding variable can be selected as an explanatory variable.

When the CCM index is higher than 0.5 in this study, the nRMSE is often smaller 0.1, where the reconstructed series is very close to the real series in the presented results. Therefore, the CCM

index that is higher than 0.5 may be considered for selecting explanatory variables. It is well known (iii) BP is known for its ability to capture nonlinear relationships? Give some insights on why it possibly failed while dealing with the 2nd case?

**Reply:** Thank you! In our results, the performance of BP does not totally failed in nonlinear system. For instance, in the results of reconstructing Lorenz-Z time series (as the following figure shows), BP can capture most of the temporal variation of the real time series. But the performance of BP is not as well as RC and LSTM. We are willing to analyze the reason in the revised manuscript.

[Figure]

In the revised manuscript, we added the algorithm descriptions for the three machine learning methods, and this is helpful to understand the different performances of them. The crucial difference is as follows: unlike RC and LSTM, all the neuron states of the BP neural network are independent on the temporal variation of time series, while the neurons of RC or LSTM can track temporal evolution. This difference was ever reported in the previous literature (i.e. Chattopadhyay et al., 2019; Reichstein et al., 2019). Moreover, the temporal evolution is crucial for modeling the nonlinear dynamics (i.e. Kantz and Schreiber, 2003; Franzke et al. 2015). And this might be responsible for the failed performance of BP in nonlinear dynamics. Our modification is as following screenshot shows:

As mentioned in section 2.2, a BP neural network does not track the temporal evolution, since its neuron states are independent to the temporal variation of time series. For LSTM*, it cannot include the information of previous time. Previous studies have revealed that the temporal evolution and memory are crucial properties for the nonlinear time series (Kantz and Schreiber, 2003; Franzke et al. 2015), which could not be neglected when modeling nonlinear dynamics. These might be responsible to that BP and LSTM* fail in dealing with this nonlinear Lorenz 63 system.

Investigations for the application of BP in other different nonlinear relationships needs to be further addressed in the future.

[1] Chattopadhyay A., Hassanzadeh P., Palem K., Subramanian D.: Data-driven prediction of a multi-scale Lorenz 96 chaotic system using a hierarchy of deep learning methods: reservoir computing, ANN, and RNN-LSTM. arXiv preprint arXiv:1906.08829, 2019.

[2] Reichstein, M., Camps-Valls, G., Stevens, B., Jung, M., Denzler, J., Carvalhais, N.: Deep learning and process understanding for data-driven Earth system science. Nature, 566(7743), 195, 2019.

[3] Kantz, H., Schreiber, T.: Nonlinear time series analysis (Vol. 7). Cambridge university press, 2004.

[4] Franzke C. L., Osprey, S. M., Davini, P., Watkins, N. W.: A dynamical systems explanation of the Hurst effect and atmospheric low-frequency variability. Sci. Rep., 5, 9068, 2015

(iv) Finally, it turned out that RC is more sensitive to CCM index, while LSTM is not. What could be the possible reason behind this? Does it indicate that all these conclusions depend on the type of ML used?

**Reply:** Thank you! Both RC and LSTM are sensitive to the CCM index. For instance, the following figure demonstrates the association between the CCM index and reconstruction quality (nRMSE) of RC and LSTM. For both results of RC and LSTM, there exists a significant correspondence between the nRMSE and the CCM index.

[Figure]

**Figure 8** Scatter plot of nRMSE values and CCM index values. The blue boxes are results of the RC machine learning, and the black cycles are results of the LSTM machine learning. The blue and grey dashed lines are the fitted linear trends of the blue boxes and black cycles respectively, and these two dependency trends are both significant because their p-values are both smaller than 0.05.

Such phenomenon can be partially explained by the CCM theory (we provided it in the method section). For two variables which are dynamically coupled (called $X$ and $Y$ here), the CCM index can estimate how much information content of $Y$ is coded in the time series of $X$. Therefore, when inputting $X$ to the neural network, not only the information content of $X$, but also the information content of $Y$ can be learned by the neural network. And then it is possible to reconstruct $Y$ from the trained neural network. The more information content of $Y$ is encoded in $X$, the magnitude of the corresponding CCM index will be stronger, and the machine learning performance will be better. This might be the reason for the association between the RC/LSTM performance and the CCM index, and this is a reason based on information theory.

The technical architectures of different types of ML also influence their own performances, and this is not from the property of the data. The association between the RC/LSTM performance and the CCM index, presents the influence from the data and dynamical systems. **As for other machine learning methods, it is unknown whether their performances are also sensitive to the CCM index, and this needs a further investigation in the**

**future.** We have modified the narration in the revised manuscript, as the following screenshot shows:

$X_{18}$; meanwhile, $X_1$ is reconstructed from $X_2$, $X_3$ …, $X_{18}$, respectively. We find a significant correspondence exists between the nRMSE and the CCM index (Fig. 8), for both results of RC and

LSTM. Here we only use a simple LSTM architecture, and there are many other variants of this architecture where the abnormal point of LSTM in Fig. 8 might be reduced. The result of Fig. 8

reveals the robust association between the CCM index and reconstruction quality in the machine learning frameworks of RC and LSTM. For other machine learning methods, such association deserves further investigation.

Please respond to all the comments and revise your manuscript.

**Reply:** Thank you! We have revised our manuscript according to all the comments by you and three reviewers.

**Reply to the comments of Anonymous Referee #1**

**The comments of Anonymous Referee #1:**

1. This manuscript investigates the potentialities of reconstructing time series using machine learning (ML) techniques. This approach is applied on a set of simple systems, and then applied to the interaction between the Tropical surface temperature and the Northern extra-tropical surface temperature. Different configurations of the machine learning approaches are explored, the reservoir computing, the long short-term memory, but also a simplified version of the latter and back-propagation. The authors use the correlation (for linear systems) and the convergent cross mapping (for nonlinear systems), CCM, as tools to evaluate the ability of the machine learning approaches to reproduce the original time series.

   **Although I find the idea of putting in parallel the CCM with the ability of reconstructing time series based on ML very interesting, the description of the tools and the results is confusing, the presentation is quite poor and many details on the approaches used are missing.**

**Response:** Many thanks for your comments and suggestions! T**he results and conclusions in the paper are correct. The confusion of Anonymous Referee #1 is the relationship between reconstruction direction and the CCM dependence, and this confusion is mainly induced by the lack of description of the CCM theory.**

   We have thoroughly improved the manuscript by incorporating all of your comments and suggestions. Please see our revised manuscript. In the following, we would like to reply to your comments.

2. My first main point is the confusion present in the notation of input/output and the notion of directional dependence. Let me clarify my point by considering Table 2 in which the results for the Lorenz 3-variable system are displayed. The first column indicates the input of the ML approach (also indicated a(t)), the second the output of the ML (also indicated b(t)), while the fourth represents the CCM dependence. The later, as defined at lines 291-297, has high values if b(t) influence a(t). So according to that table if b(t) is influencing a(t) I should get good results of fitting from a(t) to b(t). I am really confused with this claim.

**Response:** Thank you! **The results of Table 2 are correct:** the Lorenz-X can be used to reconstruct the Lorenz-Z, but the Lorenz-Z cannot be used to reconstruct the Lorenz-X, which can be also seen in the previous literature of *Lu et al. 2017*[1]. In the paper of *Lu et al. 2017*[1], they used the "nonlinear observability" of the controlled system theory to explain such phenomenon. However, the "nonlinear observability" introduced in *Lu et al. 2017*[1] is only usable in the system with known mathematical equation, here we employ the CCM coefficient which does not rely on any known equation.

**According to the literature** [2-6]**, the claim about the relationship of the CCM dependence and reconstruction direction, is correct and accurate:** if $b$ influence $a$ but $a$ does not influence $b$, the information of $b$ can be shared with $a$ (through the information transfer from $b$ to $a$), but $a$'s information cannot be shared with $b$ (there exists no information transfer from $a$ to $b$). Hence, the records of $a$ will be encoded with the information of $b$, and the time series of $b$ can be recovered from the records of $a$.

[1] Lu Z, Pathak J, Hunt B, Girvan M, Brockett R, Ott E. Reservoir observers: Model-free inference of unmeasured variables in chaotic systems. Chaos 27(4), 041102 , 2017.

[2] Takens, F.: Detecting strange attractors in turbulence. Dynamical Systems and Turbulence, Lecture Notes in Mathematics, 898, 366–381 (Springer Berlin Heidelberg), 1981.

[3] Hlaváčková-Schindler, K., Paluš, M., Vejmelka, M., Bhattacharya, J. Causality detection based on information-theoretic approaches in time series analysis. Physics Reports, 441(1), 1-46, 2007.

[4] Sugihara, G., May, R., Ye, H., Hsieh, C. H., Deyle, E., Fogarty, M., Munch, S.: Detecting causality in complex ecosystems. Science, 338(6106), 496-500, 2012.

[5] Vannitsem, S., Ekelmans, P. Causal dependences between the coupled ocean–atmosphere dynamics over the tropical Pacific, the North Pacific and the North Atlantic. Earth Syst. Dyn., 9(3), 1063-1083, 2018.

[6] Tsonis, A. A., Deyle, E. R., Ye, H., Sugihara, G.: Convergent cross mapping: theory and an example. In Advances in Nonlinear Geosciences (pp. 587-600), Springer, Cham., 2018.

We have modified the manuscript, and then the association between of the CCM and reconstruction quality will be better understood. As the following screenshot shows:

| 273 | According to literature (Sugihara et al., 2012; Ye et al., 2015), the CCM index is related to the |
| --- | --- |
| 274 | ability of using one variable to reconstruct another variable: if $b$ influence $a$ but $a$ does not influence |
| 275 | $b$, the information content of $b$ can be encoded in $a$ (through the information transfer from $b$ to $a$), |
| 276 | but the information content of $a$ is not encoded in $b$ (there exists no information transfer from $a$ to $b$). |
| 277 | Therefore, the time series of $b$ can be reconstructed from the records of $a$. For the CCM index |
| 278 | ($\rho_{a \to b}$), its magnitude represents how much information content of $b$ is encoded in the records of $a$. |
| 279 | Therefore, the high magnitude of $\rho_{a \to b}$ means that $b$ causes $a$, and we can get good results of |
| 280 | reconstruction from $a$ to $b$. In this paper, we will test the association between the CCM index and the |
| 281 | reconstruction performance of machine learning. |

In a nonlinear coupled system, it is known that the coupling strength between two variables
cannot be estimated by the linear Pearson correlation (Brown, 1994; Sugihara et al., 2012). Here, we
use CCM to estimate the coupling strength between $X$ and $Z$, and then it shows a high magnitude of
the CCM index: $\rho_{X \to Z} = 0.91$. According to the CCM theory (see Method), such a high magnitude
of the CCM index indicates that the information content of $Z$ is encoded in the time series of $X$.
Therefore, we conjecture that: when inputting $X$ to the neural network, not only the information
content of $X$, but also the information content of $Z$ can be learned by the neural network. And then it
is possible to reconstruct $Z$ from the trained neural network. We will test it in the following.

3. I have the same problem with the other tables, and in particular with Table 4 which is even more confusing when related with the discussion in the text. In the table it is indicated that TSAT influences strongly NHSAT but then the ML modeling is done from NHSAT to TSAT. This is what is claimed at lines 463-464, while in the conclusion it is said (line 542) that the TSAT is mainly influencing the NHSAT. I hope this is just a matter of confused notation but I am not sure and I strongly recommend the authors to revisit carefully their notations and interpretation carefully.

**Response:** Thank you! We thoroughly improved the notations and interpretations in the manuscript, as the following screenshot shows:

The daily NHSAT and TSAT time series are shown in Fig. 10a. It is quite different for the
oscillation shapes of the NHSAT and TSAT series, and there is a weak linear correlation (0.08, see
Table 4) between them. In the scatter plot for the NHSAT and TSAT (Fig. 10b), the marked
nonlinear structure is observed between NHSAT and TSAT. Such a weak linear correlation will
make the linear regression model fail to reconstruct one series from the other. Likewise, there is no
explicit physical expression that can transform TSAT and NHSAT to each other. Now we try to use
machine learning to reconstruct these climate series. The CCM index of that NHSAT cross maps
TSAT is 0.70, and the CCM index of that TSAT cross maps NHSAT is 0.24 (Table 4). The CCM
index means that the information content of TSAT is well encoded in the records of NHSAT, and
the information transfer might be mainly from TSAT to NHSAT, which is consistent with previous
studies (Farneti and Vallis, 2013). Further, the CCM analysis indicates that the reconstruction from
NHSAT to TSAT might obtain a better quality than the opposite direction.

We have inspected the results and conclusions, and the results and conclusions about Table 4 are correct. **Sugihara et al. 2012 [1] ever suggested that the reconstruction direction is opposite to the causal**

**dependence direction.** The confusion about the relationship between reconstruction direction and the CCM dependence, is induced by the lack of description of the CCM theory in the previous manuscript.

Firstly, as the literature shows [1-4]: if *b* does influence *a* (*a* and *b* are two arbitrary variables), and then the information of *b* can be shared with *a* (through the information transfer from *b* to *a*). Therefore, the records of *a* will be encoded with the information of *b*, and the time series of *b* can be recovered from the records of *a*. At that time, the CCM coefficient $\rho_{a \to b}$ denotes: when using *a*'s records to recover the values of *b*, how well the quality is. Likewise, the magnitude of $\rho_{a \to b}$ represents how much information of *b* is encoded in the records of *a*.

Then, in our results about using NHSAT to reconstruct TSAT, the CCM index that NHSAT cross maps TSAT is of high value (0.7). This suggests that the NHSAT's records are able to recover the values of TSAT, which stems from that the information of TSAT is encoded in NHSAT. But the CCM index that TSAT cross maps NHSAT is of high value (0.24). According to the CCM theory, we know that the influence from NHSAT to TSAT, is not strong as the influence from TSAT to NHSAT, which also consists with the real dynamical process revealed by previous research [6].

Finally, the information transfer inferred from the CCM suggests that: when employing Reservoir Computing to reconstruct TSAT from the NHSAT's records, the reconstruction quality will be better than reconstruct NHSAT from the TSAT's records. And our results are really consisting with the indication of CCM.

[1] Sugihara, G., May, R., Ye, H., Hsieh, C. H., Deyle, E., Fogarty, M., Munch, S.: Detecting causality in complex ecosystems. Science, 338(6106), 496-500, 2012.

[2] Takens, F.: Detecting strange attractors in turbulence. Dynamical Systems and Turbulence, Lecture Notes in Mathematics, 898, 366–381 (Springer Berlin Heidelberg), 1981.

[3] Hlaváčková-Schindler, K., Paluš, M., Vejmelka, M., Bhattacharya, J. Causality detection based on information-theoretic approaches in time series analysis. Physics Reports, 441(1), 1-46, 2007.

[4] Vannitsem, S., Ekelmans, P. Causal dependences between the coupled ocean–atmosphere dynamics over the tropical Pacific, the North Pacific and the North Atlantic. Earth Syst. Dyn., 9(3), 1063-1083, 2018.

[5] Tsonis, A. A., Deyle, E. R., Ye, H., Sugihara, G.: Convergent cross mapping: theory and an example. In Advances in Nonlinear Geosciences (pp. 587-600), Springer, Cham., 2018.

[6] Vallis, G. K., Farneti, R.: Meridional energy transport in the coupled atmosphere–ocean system: Scaling and numerical experiments. Q. J. Roy. Meteor. Soc., 135(644), 1643-1660, 2009.

4.   A second important concern is the way the ML is used. In Figure 2 there are three parts but it seems to me that the ML system is composed of the two first ones, the third one being the application of the optimized system to new input data. So **It should be worth to split both and also to clarify the details of the Machine Learning underlying structure, number of nodes, number of layers (if any)… Details on the different ML systems used are necessary. A detailed description is also missing for the CCM method**.

**Response:** Thanks for your comments and suggestions.

The Reservoir Computer framework used in our work is developed in *Lu et al.* 2017 [1]. In *Lu et al.* 2017 [1], the Reservoir Computer framework only has the first two components shown in Figure 1*. We have tested the third component (a repetitive operation for the first two components) did not influence the results, and the first two components were enough. In the revised manuscript, we will carefully improve the diagram and the description of Reservoir computer according to the introduction in *Lu et al.* 2017 [1].

[Figure]

Figure 1* The schematic of Reservoir computer in the previous manuscript (we will revised this figure in the revised manuscript).

[1] Lu Z, Pathak J, Hunt B, Girvan M, Brockett R, Ott E. Reservoir observers: Model-free inference of unmeasured variables in chaotic systems. Chaos 27(4), 041102 , 2017.

**We improved the detail descriptions for all used machine learning methods, and the CCM method, as the following screenshot shows:**

[revised manuscript text omitted]

5. These two main problems prevent me to recommend publication of this manuscript at this stage although the main question addressed is very interesting (CCM vs ML). A considerable effort of clarification and rewriting is necessary.

**Response:** Thank you! According to your above suggestions, we carefully worked on the more detailed clarification and rewriting for the machine learning method and the CCM theory, so that the relationship between CCM and machine learning could be better presented. And then, results and conclusions will be better understood.

More specific points:

6. Line 54: What does mean "wile physics of systems is suggested for consideration"? Please rephrase.

**Response:** Thank you! The excepted meaning is that: we should focus on whether the dynamical properties in the underlying system can be described, and how the dynamical properties will influence the performance of machine learning. In the revised manuscript, we thoroughly rearranged the introduction part, so that it can be easier to follow the story. Please see the manuscript.

7. Lines 57-58. You probably meant that: sensitivity to initial conditions is a property of the underlying system giving rise to the climate time series. Chaos theory is a framework in which this type of dynamics can be described. Please rephrase.

**Response:** Thank you! We carefully rephrased these sentences, as the following screenshot shows:

| 43 | Neural network-based machine learning provides effective tools for studying climatic data |
|---|---|
| 44 | (Reichstein et al., 2019), which attracts great attention recently. The machine learning approach is |
| 45 | widely applied to downscaling and data mining analyses (Mattingly et al., 2016; Racah et al., 2017), |
| 46 | and it can be also used to predict the time series of climate variables, such as temperature, humidity, |
| 47 | runoff and air pollution (Zaytar and Amrani, 2016; Biancofiore et al., 2017; Kratzert et al., 2019; |
| 48 | Feng et al., 2019). Recently, it is demonstrated that a large potential application of machine learning |
| 49 | is to reconstruct the temporal dynamics of complex systems (Pathak et al., 2017; Du et al., 2017; |
| 50 | Watson, 2019). Studies (Pathak et al., 2017; Lu et al., 2018; Carroll, 2018) have shown that the |
| 51 | chaotic attractors in Lorenz system and Rossler system can be described by machine learning. Since |
| 52 | chaos is the key property of the underlying climate system giving rise to climatic time series (Lorenz, |
| 53 | 1963; Patil et al., 2001), these studies provide a theoretical explanation why the machine learning |
| 54 | can be well applied in reconstructing climate temporal dynamics. |

8. Line 67. What is nonlinear correlation? I think that this is not an appropriate terminology. Please revisit your manuscript with that in mind.

**Response:** Thank you! We carefully rephrased the explanation of "nonlinear correlation" in the revised manuscript.

Here the excepted meaning of "nonlinear correlation" is that: for two variables from a common system, their time series might have dynamical relationship with each other. Sometimes the linear Pearson correlation of these two time series is weak or even equal to zero, but their relationship can be quantified by means of some other statistical measurement. At that time, such relationship whose linear correlation is potentially weak, is regarded as nonlinear correlation.

We will modify the sentences as the following screenshot:

| 60 | variables. Because different climate variables are coupled with one another (Donner and Large, |
| 61 | 2008), and the coupled variables will share their information content with one another through the |
| 62 | information transfer (Takens, 1981; Schreiber, 2000; Sugihara et al., 2012). Furthermore, a coupling |
| 63 | often results in that the observational time series are statistically correlated (Brown, 1994). |
| 64 | Correlation is a crucial property for the climate system, and often influences the climatic time series |
| 65 | analysis. "Pearson Coefficient" is often used to detect the correlation, which only detects the linear |
| 66 | correlation. It is known that when the Pearson correlation coefficient is weak, most of traditional |
| 67 | regression methods will fail in dealing with the climatic data, such as fitting, reconstruction and |
| 68 | prediction (Brown, 1994; Sugihara et al., 2012; Emile-Geay and Tingley, 2016). However, a weak |
| 69 | linear correlation does not mean that there is no coupling relation between the variables. Previous |
| 70 | studies (Sugihara et al., 2012; Emile-Geay and Tingley, 2016) have suggested that, although the |
| 71 | linear correlation of two variables is potentially absent, they might be nonlinearly coupled and can |
| 72 | be exploited by analysis. For instance, the linear cross-correlations of sea surface temperature series |

9. Line 72. You speak about "trajectories". Maybe this is more "relationships".

**Response:** Thank you! We revised this narration, as the screenshot shows:

| 75 | (Ludescher et al., 2014; Conti et al., 2017). The linear correlations between ENSO/PDO index and |
| 76 | some proxy variables are weak but their nonlinear coupling relations can be detected, which |
| 77 | contributes greatly to reconstructing longer paleoclimate time series (Mukhin et al., 2018). These |
| 78 | studies indicate that nonlinear coupling relations would contribute to the better analysis, |

10. Line 87. "hided"?

**Response:** Thank you! We revise this word in the manuscript, as the screenshot shows:

| 112 | Finally, we will discuss a real-world example from climate system. It is known that there exist |
| 113 | atmospheric energy transportations between the tropics and the Northern Hemisphere, which results |
| 114 | in the coupling between the climate systems in these two regions (Farneti and Vallis, 2013). Due to |
| 115 | the underlying complicated processes, it is difficult to use a formula to cover this coupling between |
| 116 | the tropical average surface air temperature (TSAT) series and the Northern Hemispheric surface air |
| 117 | temperature (NHSAT) series. We employ machine learning methods to investigate whether the |
| 118 | NHSAT time series can be reconstructed from the TSAT time series, and whether the TSAT time |
| 119 | series can be also reconstructed from the NHSAT time series. Accordingly, the conclusions from our |
| 120 | model simulations can be further tested and generalized. |

**11. Line 111. "learnt" should probably be "reconstructed".**

**Response:** Thank you! We revised this word in the manuscript, as the screenshot shows:

i.e., there is the stable coupling or dynamic relation $b(t) = F[a_1(t), a_2(t),...,a_n(t)]$ among inputs

$a_1(t), a_2(t),...,a_n(t)$ and output $b(t)$. If this inherent coupling relation can be reconstructed by machine learning in the training series, the reconstructed coupling relation should be reflected by machine learning in the testing series. Therefore, the workflow of our study can be summarized as follows (see Fig. 1):

(i) During the training period, $a_1(t), a_2(t),...,a_n(t)$ and $b(t)$ are input into the machine learning frameworks to learn the coupling or dynamic relation $b(t) = F[a_1(t), a_2(t),...,a_n(t)]$. The inferred coupling relation is denoted as $b(t) = \hat{F}[a_1(t), a_2(t),...,a_n(t)]$. Then it is tested whether this coupling relation can be reconstructed by machine learning.

(ii) The second step is accomplished with the testing series to apply the reconstructed coupling relation $\hat{F}$ together with only $a_1(t'), a_2(t'),...,a_n(t')$ to derive $b(t')$, denoted as $\hat{b}(t')$. $\hat{b}(t')$ is called "the reconstructed $b(t')$" since only $a_1(t'), a_2(t'),...,a_n(t')$ and the reconstructed coupling relation $\hat{F}$ have been taken into account.

(iii) The first objective of this study is to answer whether the coupling relation

$b(t) = F[a_1(t), a_2(t),...,a_n(t)]$ can be reconstructed by machine learning, i.e., whether the reconstructed coupling relation $\hat{F}$ can well approximate the real coupling relation $F$. Since we do not intend to reach an explicit formula of the reconstructed coupling relation $\hat{F}$, we will answer this question indirectly by comparing the reconstructed series $\hat{b}(t')$ with the original series $b(t')$. If

$\hat{b}(t') \approx b(t')$, then it can be regarded as $\hat{F} \approx F$, and the machine learning can indeed learn the

**12. Line 115. "learnt" is probably "estimated" or "inferred".**

**Response:** Thank you! We will revise this word in the manuscript, as the screenshot shows:

(i) During the training period, $a_1(t), a_2(t),...,a_n(t)$ and $b(t)$ are input into the machine learning frameworks to learn the coupling or dynamic relation $b(t) = F[a_1(t), a_2(t),...,a_n(t)]$. The inferred coupling relation is denoted as $b(t) = \hat{F}[a_1(t), a_2(t),...,a_n(t)]$. Then it is tested whether this coupling relation can be reconstructed by machine learning.

**13. Figure 1. Why putting the training after the testing? It does not look natural (and also confusing).**

**Response:** Thanks for your suggestions. Such arrangement is due to the consideration of reconstructing climate records. We are inspired by that it is often necessary to reconstruct the historical records for climate variables.

For instance, as Figure 2* shows, for the records of proxy data (tree ring or ice core, labeled as $a(t)$ in Figure 2*), we might obtain the data from the historical and current period. For the records of climatic variable like air temperature (labeled as $b(t)$ in Figure 2*), we might only obtain the data from the current period. At that time, the data-driven approach (such linear regression) is often applied to fit the relation between proxy data ($a(t)$) and air temperature ($b(t)$) through their current observational data, and then the historical proxy data and the fitted relationship can be used to reconstruct the historical records of air temperature.

[Figure]

Figure 2* The blue solid line denotes the observational records of climatic variable (labeled as $b(t)$) in current period. The blue dashed line denotes that the records of climatic variable are absence of observation in the past time. The red solid line denotes the proxy data (labeled as $a(t)$) in both of current period and past time.

The above reconstruction scheme is also very useful for some important climate problems such as **paleoclimate reconstruction** [1], **interpolation for the missing points in measurements** [2] and **parameterization schemes** [3]. Our study is motivated by investigating how to better apply machine learning to the reconstruction of climate time series (under different coupling dynamics of climate systems).

[1] Emile-Geay, J., Tingley, M.: Inferring climate variability from nonlinear proxies: application to paleo-ENSO studies. Clim. Past., 12(1), 31-50, 2016.
[2] Hofstra, N., Haylock, M., New, M., Jones, P., Frei, C.: Comparison of six methods for the interpolation of daily European climate data. J. Geophys. Res., 113(D21), 2008.
[3] Vissio, G., Lucarini, V.: A proof of concept for scale‐adaptive parameterizations: the case of the Lorenz 96 model. Q. J. Roy. Meteor. Soc., 144(710), 63-75, 2018.

**14. Lines 175-178. Quite confusing. Please clarify the way prediction is done. I think that the presentation of the ML approach should be completely revisited.**

**Response:** Thank you! We thoroughly rewrited this part about the machine learning framework, and detail description of Reservoir Computer, including the structure, number of nodes, number of layers will be clearly presented.

The Reservoir Computer framework used in our work is developed in *Lu et al.* 2017 [1]. And we referred the introduction in *Lu et al.* 2017 [1] to modify the description. Our modified version is as the screen shot shows:

[1] Lu Z, Pathak J, Hunt B, Girvan M, Brockett R, Ott E. Reservoir observers: Model-free inference of unmeasured variables in chaotic systems. Chaos 27(4), 041102 (2017).

A newly developed neural network called RC (Du et al., 2017; Lu et al., 2017; Pathak et al.,

2018) has three layers: the input layer, the reservoir layer and the output layer (see Fig. 2). If

$a(t)$ and $b(t)$ denote two time series from a system, and then the following steps can estimate $b(t)$

from $a(t)$:

[Figure]

**Figure 2** Schematic of the RC neural network: the three layers are the input layer, the reservoir layer, and the output layer. The input layer consists of a matrix "$W_{in}$" (whose elements are randomly chosen from the interval

[-1, 1]). The reservoir layer consists of $N$ reservoir neurons whose connectivity is through the adjacent matrix "$M$", and $r(t)$ represents the activations of the $N$ neurons. The output layer consists of a matrix "$W_{out}$", whose elements are trainable in the training process. A time series $a(t)$ is input into the RC neural network. After the training process, the time series of $b$ variable can be reconstructed by machine learning, denoted as $\hat{b}(t)$.

(i) $a(t)$ (a vector with length $L$) is input into the input layer and reservoir layer. There are four components in this process: the initial reservoir state $r(t)$ (a vector with dimension $N$, representing the $N$ neurons), the adjacent matrix "$M$" (size $N \times N$) representing connectivity of the $N$ neurons, the input-to-reservoir weight matrix "$W_{in}$" (size $N \times L$), and the unit matrix "$E$" (size $N \times N$) which is crucial for modulating the bias in the training process (Lu et al., 2018). The elements of "$M$" and

"$W_{in}$" are randomly chosen from a uniform distribution in [−1, 1], and we set $N = 1000$ here (we

182     have tested that this yields the good performance). These components are employed by Eq. (1), and
183     then an updated reservoir state $r^*(t)$ is output.

$$r^*(t)=\tanh\left[M \cdot r(t) + W_{in} \cdot a(t) + E\right],\tag{1}$$

185     (ii) $r^*(t)$ then gets into the output layer that consists of the reservoir-to-output matrix "$W_{out}$". As
186     Eq. (2) shows, $r^*(t)$ will be trained as the estimated value $\hat{b}(t)$. The mathematical form of "$W_{out}$"
187     is shown by Eq. (3), which is a trainable matrix that fits the relation between $r^*(t)$ and $b(t)$ in the
188     training process. "$\|\cdot\|$" denotes the $L_2$-norm of a vector ($L_2$ represents the least square method) and
189     $\alpha$ is the ridge regression coefficient, whose values are determined after the training.

$$\hat{b}(t)=W_{out}\cdot r^*(t),\tag{2}$$

$$W_{out}=\arg\min_{W_{out}}\left\| W_{out}\cdot r^*(t)-Y(t+\tau)\right\| +\alpha\|W_{out}\|,\tag{3}$$

192     After this reservoir neural network has been trained, we can use it to estimate $b(t)$, where the
193     estimated value is noted as $\hat{b}(t)$.

15. Line 191. Why using this measure and why 0.1 is a good threshold? These should be detailed.

**Response:** Thank you! Normalizing the RMSE is to compare the time series with different variability and unit [1, 2]. For instance, the time series of $x_1$ and $x_2$ in Figure 3* are both with zero mean and unit variance, but the extreme values of $x_2$ are much stranger than of $x_1$. It is revealed [1, 2] that such difference will interfere in the fair comparison of the RMSE. In order to avoid such interference induced by the extreme values, we are suggested to normalize the RMSE with the max distribution range of the original data [1, 2], as equation (5) shows.

$$RMSE = \sqrt{\frac{1}{k}\sum_{t}[b(t')-\hat{b}(t')]^2},\tag{4}$$

$$nRMSE = \frac{RMSE}{\max[b(t')]-\min[b(t')]}.\tag{5}$$

[Figure]

Figure 3* The standardized time series of $x_1$(blue) and $x_2$ (red) with zero mean and unit variance. The $x_1$ is a random time series with Gaussian probability distribution, and $x_2$ is a random time series with extreme probability distribution.

"nRMSE = 0.1" means that the RMSE occupies 10% of the max distribution range of the original data, and this is a tolerable level of the bias [1, 2]. In the figures of comparing reconstructed series with real series, we can observe that when the reconstructed series is close to the real series in curves, the corresponding nRMSE is less than 0.1.

[1] Hyndman, R. J., Koehler, A. B.: Another look at measures of forecast accuracy. Int. J. Forecasting., 22(4), 679-688, 2006.
[2] Pennekamp, F., Iles, A. C., Garland, J., Brennan, G., Brose, U., Gaedke, U., Novak, M.: The intrinsic predictability of ecological time series and its potential to guide forecasting. Ecol, Monogr., e01359, 2019.

We will carefully explain the meaning of nRMSE and its threshold in the revised manuscript, as the following screenshot shows:

To evaluate the quality of reconstruction by machine learning, the root mean squared error (RMSE) of residual series (Hyndman and Koehler, 2006) is adopted (Eq. (4)), which represents the difference between the real series $b(t')$ and the reconstructed series $\hat{b}(t')$. In order to fairly compare the errors of reconstructing different processes with different variability and units (Hyndman and Koehler, 2006; Pennekamp et al., 2018; Huang and Fu, 2019), we normalize the

RMSE as Eq. (5) shows.

$$RMSE = \sqrt{\frac{1}{k}\sum_t [b(t')-\hat{b}(t')]^2},\qquad(4)$$

$$nRMSE = \frac{RMSE}{\max[b(t')]-\min[b(t')]}.\qquad(5)$$

16. Line 212. Runge-Kutta integral? What does it mean? Maybe "integrator"?

**Response:** Thanks for your suggestions. We will revise this word in the manuscript, as the screenshot shows:

$\theta$ is decreased, the coupling strength between $X_k$ and $Y_{k,\,j}$ will be enhanced. The fourth-order

Runge-Kutta integrator and periodic boundary condition are adopted (that is: $X_0 = X_K$ and $X_{K+1} = X_1$;

$Y_{k,\,0} = Y_{k-1,\,J}$ and $Y_{k,\,J+1} = Y_{k+1,\,1}$), and the integral time unit was taken as 0.05. The time series $X_1(t)$

and $Y_{1,\,1}(t)$ are used for the reconstruction analysis.

17. Section 2.4.2. Please give more details on the way average is done, and whether the seasonality is removed and how?

This also open the question on how the parameters of the ML are changing as a function of the season. There is not enough details on how the datasets are handled.

**Response:** Thank you! We improved the details on the way average is done in the manuscript.

The seasonality was not removed, and this did not influence the parameters of the machine learning. The reasons are as the following shows:

**Firstly,** literature [1-4] has revealed that seasonal cycle of air temperature is time-varying (especially for the mid-latitude regions [1] and tropics [2]), and the existing methods are often hard to thoroughly remove such time-varying seasonal cycle [4]. So that removing seasonality might take some controversial and unknown bias for the results [5].

[1] Paluš, M., Novotná, D., Tichavský, P.: Shifts of seasons at the European mid‐latitudes: Natural fluctuations correlated with the North Atlantic Oscillation. Geophysical research letters, 32(12), 2005.

[2] Qian, C., Wu, Z., Fu, C., Wang, D.: On changing El Niño: A view from time-varying annual cycle, interannual variability, and mean state. Journal of Climate, 24(24), 6486-6500, 2011.

[3] Jajcay, N., Hlinka, J., Kravtsov, S., Tsonis, A. A., Paluš, M.: Time scales of the European surface air temperature variability: The role of the 7–8 year cycle. Geophysical Research Letters, 43(2), 902-909, 2016.

[4] Deng, Q., Nian, D., Fu, Z.: The impact of inter-annual variability of annual cycle on long-term persistence of surface air temperature in long historical records. Climate dynamics, 50(3-4), 1091-1100, 2018.

[5] Theiler, J., Eubank, S.: Don't bleach chaotic data. Chaos: An Interdisciplinary Journal of Nonlinear Science, 3(4), 771-782, 1993.

**Secondly,** if focusing on the application in reconstructing regional temperature [6-8], the annual variability will be the most important and commonly concerned. At that time, the seasonality is not necessary to be removed. And as the Figure 4* shows, the annual variability of reconstructed series is really close to the real series. If we remove the seasonality, it might take with some unknown bias [4-5].

[6] Van Engelen, A. F., Buisman, J., Jnsen, F.: A millennium of weather, winds and water in the low countries. In History and climate (pp. 101-124). Springer, Boston, MA, 2001.

[7] Moberg, A., Sonechkin, D. M., Holmgren, K., Datsenko, N. M., Karlen, W.: 2,000-year Northern Hemisphere temperature reconstruction. IGBP PAGES/World Data Center for Paleoclimatology Data Contribution Series, 19, 2005.

[8] Mann, M. E., Zhang, Z., Rutherford, S., Bradley, R. S., Hughes, M. K., Shindell, D., Ni, F.: Global signatures and dynamical origins of the Little Ice Age and Medieval Climate Anomaly. Science, 326(5957), 1256-1260, 2009.

**Thirdly,** when employing neural network approach, it is a common step to divide the data into training data and testing data. Then the training data is used to train the parameters of neural network. After the training process is accomplished, the parameters of neural network will be determined and fixed. And then, the trained neural network will be used in the testing data, and they will be not changed any more.

**Fourthly,** if dividing the time series into different seasons, and respectively reconstructing them in different seasons, the parameters of machine learning might be changing in different seasons. However, after dividing these daily time series into different seasons, the data length will be not long enough to accomplish the machine learning approach, which might take the large bias to the results. So, we did not divide the time series according to different seasons, and the seasonality will not influence the parameters of machine learning changing with the season.

[Figure]

Figure 4* Comparison between the annual mean values of reconstructed TSAT (red) and the annual mean values of original TSAT (blue).

18. Lines 295-296. Sugihara (1994). This reference does not exist in the reference list. What is "empirical dynamics model? Much more information is needed on the way it is used. Embedding dimension and so on.

**Response:** Thank you! We rewrote this part in the manuscript, as the screenshot shows:

[Figure]

ii) Estimating the weight parameter $w_i$ which denotes the associated weight between two vectors

"$M_a(t)$" and "$M_a(t_i)$" ("$t$" denotes the excepted time in this cross mapping), defined as:

$w_i = \frac{u_i}{\sum_{i=1}^{m+1} u_i}$,                           (7)

$u_i = exp\{-\frac{d[M_a(t), M_a(t_i)]}{d[M_a(t), M_a(t_1)]}\}$,                  (8)

where $d[M_a(t), M_a(t_i)]$ denotes the Euler distance between vectors "$M_a(t)$" and "$M_a(t_i)$". The nearest neighbor to "$M_a(t)$" generally corresponds to the largest weight.

iii) Cross mapping the value of $b(t)$ by

$\hat{b}(t) = \sum_{i=1}^{m+1} w_i b(t_i)$.                          (9)

$\hat{b}(t)$ denotes the estimated value of $b(t)$ with this phase-space cross mapping. Then, we will evaluate the cross mapping skill (Sugihara et al., 2012; Tsonis et al., 2018) as the follows:

$\rho_{a \to b} = corr. [b(t), \hat{b}(t)]$                        (10)

The cross mapping skill from $b$ to $a$ is also measured according to the above steps, marked as $\rho_{b \to a}$:

Sugihara et al. and Tsonis et al. ever defined the causal inference according to $\rho_{a \to b}$ and $\rho_{b \to a}$ like that: (i) if $\rho_{a \to b}$ is convergent when $L$ is increased, and $\rho_{a \to b}$ is of high magnitude, then $b$ is suggested to be a causation of $a$. (ii) Besides, if $\rho_{b \to a}$ is also convergent when $L$ is increased, and is of high magnitude, then the causal relationship between $a$ and $b$ is bidirectional ($a$ and $b$ cause each other). In our study, all values of the CCM indices are measured when they are convergent with the data length (Tsonis et al. 2018).

**19. Line 302. What is "unstable local correlation". What is this?**

**Response:** Thank you! The expected meaning of "unstable local correlation" is that the local Pearson correlation between two variables is time-varying. As the Figure 5*(a) shows, the time series of $X$ and $Z$ are sometimes positively correlated but sometimes nonlinear correlated at different regimes. Hence, the overall Pearson correlation between $X$ and $Z$ is very weak. Such time-varying local Pearson correlation is suggested to be universal in nonlinear dynamical systems [1].

[1] Sugihara, G., May, R., Ye, H., Hsieh, C. H., Deyle, E., Fogarty, M., Munch, S.: Detecting causality in complex ecosystems. Science, 338(6106), 496-500, 2012.

     We modified the words in the revised manuscript for better understanding, as the following screenshot shows:

0.002) in the Lorenz63 model (Table 2), and such a weak linear correlation is resulted from the time-varying local correlation between variables $X$ and $Z$ (see Fig. 5a): For example, $X$ and $Z$ are negatively correlated in the time interval of 0-200, but positively correlated in 200-400. This alternation of negative and positive correlation appears over the whole temporal evolutions of $X$ and

$Z$, which leads to an overall weak linear correlation. In this case, we cannot use a feasible linear

[Figure]

Figure 5* (a) The $X$ time series (black) and the $Z$ time series (blue) of the Lorenz 63 system. (b) Scatter plot of $X$

time series and $Z$ time series of the Lorenz 63 model (blue dots).

20. Table 2. As already mentioned in my main comment, very confusing. Please modify.

**Response:** Thank you! The results and conclusion of Table 2 is correct (see also *Lu et al. 2017*[1]), and this confusion is induced by the lack description of the CCM theory. After the CCM theory is well explained in the manuscript, the result can be better understood.

[1] Lu Z, Pathak J, Hunt B, Girvan M, Brockett R, Ott E. Reservoir observers: Model-free inference of unmeasured
    variables in chaotic systems. Chaos 27(4), 041102 (2017).

21. Figure 6. Some typos in titles. Also where is panel (d)? Is it (c)?

**Response:** Thank you! We revised this typo in the manuscript, as the screenshot shows:

**Figure 7** (a) The $Y_{1,1}$ time series(black), $X_2$ time series (black) and $X_1$ time series(blue)of the Lorenz 96 model. (b)

By means of the RC machine learning, when using $Y_{1,1}$, $X_2$ and multivariate to be the explanatory variable respectively, the corresponding reconstructed $X_1$ time series are showed respectively from the top panel to the bottom panel (red lines), and the original $X$ time series are presented by the blue lines. (c) By means of the LSTM

machine learning, when using $Y_{1,1}$, $X_2$ and multivariate to be the explanatory variable respectively, the corresponding reconstructed $X_1$ time series are showed respectively from the top panel to the bottom panel (red lines), and the original $X$ time series are presented by the blue lines.

**22. Table 3 and Fig 6. Why not using a multivariate CCM to compare with the ML fitting with multiple predictors?**

**Response:** Many thanks for your suggestions! The multi-variable CCM analysis might be useful and promising, but first of all we need to know which variable is able to become the explanatory variable. Similar to the multi-variable regression analysis, if we do not know the Pearson correlation between the target variable with every potential explanatory variable, the multi-variable regression will easily suffer from the overfitting problem.

Considering the potential **overfitting problem** and **common-driver problem** [1-2], the comparison between the multi-variable CCM and the multi-variable machine learning **absolutely deserves a further investigation**. This might occupy too many words and figures in the manuscript, so that the presentation of the main and original ideal might be influenced. In the future study, we will consider a thorough investigation for the comparison between the multi-variable CCM and the multi-variable machine learning.

[1] Runge, J., Heitzig, J., Petoukhov, V., Kurths, J.: Escaping the curse of dimensionality in estimating multivariate transfer entropy. Physical review letters, 108(25), 258701, 2012.

[2] Runge, J., Bathiany, S., Bollt, E., Camps-Valls, G., Coumou, D., Deyle, E., van Nes, E. H.: Inferring causation from time series in Earth system sciences. Nature communications, 10(1), 1-13, 2019.

**23. Lines 536-543. Really confusing. What is influencing what? TSAT or NHSAT?**

**Response:** Thank you! The excepted meaning is that TSAT influences NHSAT, which can be explained by that the energy is transferred from the tropical climate system to the Northern Hemispheric climate system [1]. We revised the narration in the revised manuscript.

[1] Vallis, G. K., Farneti, R.: Meridional energy transport in the coupled atmosphere–ocean system: Scaling and numerical experiments. Q. J. Roy. Meteor. Soc., 135(644), 1643-1660, 2009.

24. I have also noted many typographical errors, and the manuscript will benefit for a careful reading by the authors and by an English native speaker to rephrase some sentences.

**Response:** Thank you! We carefully inspected the manuscript, and we also invited colleagues of our field speaking native English to improve some sentences.

**Reply to the comments of Anonymous Referee #2**

**The comments of Anonymous Referee #2:**

**25.** This manuscript investigates the feasibility of using Machine Learning (ML) algorithm for the reconstruction of a time series with the help of a coupled time series. The study also examines the ability of an ML algorithm to represent the coupling strength of a system. The reconstruction analysis investigates three ML algorithms: Back Propagation (BP), Long Short-Term Memory (LSTM), and Reservoir Computing (RC). The study also investigates the influence of type of coupling (linear or non-linear) on the performance of ML algorithm. This is achieved by using a simple linear system, a simple non-linear system (Lorenz-63), a high-dimensional non-linear system (Lorenz-96), and a real-world system (coupling between Tropical surface air temperature and Northern Hemisphere surface air temperature). The linearity is measured using Pearson's correlation coefficient while the non-linearity is measure using Convergent Cross Map ping Causality index (CCM). The influence of the direction of coupling and coupling strength, and the number of explanatory variables on the accuracy of reconstruction of different ML algorithms is also examined. The performance evaluation of ML algorithms found that RC is most suitable for the reconstruction of non-linearly coupled time series. The work is scientifically sound and I see a lot of value in this work. Especially in the future applications of ML algorithms for reconstruction of coupled time series and in understanding the influence of coupling mechanisms on the behavior of ML algorithm. However, the presentation of the work in its current form is very confusing and diverts the attention of the reader from the importance of the work. The manuscript has errors related to English too which need to be corrected. Please find my major suggestions on the manuscript below.

**Response:** Many thanks for your thoughtful comments and suggestions! The suggestions were very helpful for improving our manuscript, and we carefully revised the manuscript according to these suggestions. Please see the revised manuscript.

In the following, we would reply to your comments and suggestions.

**26.** The abstract talks about the reconstruction of a time series of a coupled system from its other coupled counter-parts. However, the introduction is not representing it intuitively. I would suggest the authors to focus on the problem of reconstruction of a time series and build the importance of coupling mechanism, importance of linear and non-linear coupling around the time series reconstruction.

**Response:** Thank you! We thoroughly rewrote the introduction in the revised manuscript, as shown in the following screenshot:

[revised manuscript text omitted]

**27.** The Methodology section does not seem to have a description of BP and LSTM in it, in as much detail as stated for RC. I would suggest the authors to incorporate the description of BP and LSTM too, as it will help the readers to better understand the behavior of the algorithms.

**Response:** Thank you! We added more detailed descriptiona of BP and LSTM into the revised manuscript.

But the algorithms of BP are much more complicated than that of RC, and there are too many equations (about 15 mathematical equations) for their algorithms so that the article will be not concise. We will carefully introduce the key steps for BP, and the relevant references will be cited for the steps.

**Especially, we will highlight the crucial differences in algorithms among RC, BP and LSTM, and this might be very helpful for understanding the application results of them.**

Our modification for the neural network algorithms are shown by the following screenshot:

**2.2.1 Reservoir computer**

A newly developed neural network called RC (Du et al., 2017; Lu et al., 2017; Pathak et al.,

2018) has three layers: the input layer, the reservoir layer and the output layer (see Fig. 2). If

$a(t)$ and $b(t)$ denote two time series from a system, and then the following steps can estimate $b(t)$

from $a(t)$:

[Figure]

**Figure 2** Schematic of the RC neural network: the three layers are the input layer, the reservoir layer, and the output layer. The input layer consists of a matrix "$W_{in}$" (whose elements are randomly chosen from the interval

[-1, 1]). The reservoir layer consists of $N$ reservoir neurons whose connectivity is through the adjacent matrix "$M$", and $r(t)$ represents the activations of the $N$ neurons. The output layer consists of a matrix "$W_{out}$", whose elements are trainable in the training process. A time series $a(t)$ is input into the RC neural network. After the training process, the time series of $b$ variable can be reconstructed by machine learning, denoted as $\hat{b}(t)$.

(i) $a(t)$ (a vector with length $L$) is input into the input layer and reservoir layer. There are four components in this process: the initial reservoir state $r(t)$ (a vector with dimension $N$, representing the $N$ neurons), the adjacent matrix "$M$" (size $N \times N$) representing connectivity of the $N$ neurons, the input-to-reservoir weight matrix "$W_{in}$" (size $N \times L$), and the unit matrix "$E$" (size $N \times N$) which is crucial for modulating the bias in the training process (Lu et al., 2018). The elements of "$M$" and

"$W_{in}$" are randomly chosen from a uniform distribution in $[-1, 1]$, and we set $N = 1000$ here (we have tested that this yields the good performance). These components are employed by Eq. (1), and then an updated reservoir state $r^{*}(t)$ is output.

$$r^{*}(t) = \tanh\left[M \cdot r(t) + W_{in} \cdot a(t) + E\right], \qquad (1)$$

(ii) $r^{*}(t)$ then gets into the output layer that consists of the reservoir-to-output matrix "$W_{out}$". As

Eq. (2) shows, $r^{*}(t)$ will be trained as the estimated value $\hat{b}(t)$. The mathematical form of "$W_{out}$"

is shown by Eq. (3), which is a trainable matrix that fits the relation between $r^{*}(t)$ and $b(t)$ in the training process. "$\|\cdot\|$" denotes the $L_2$-norm of a vector ($L_2$ represents the least square method) and

$\alpha$ is the ridge regression coefficient, whose values are determined after the training.

$$\hat{b}(t) = W_{out} \cdot r^{*}(t), \qquad (2)$$

$$W_{out} = \arg\min_{W_{out}} \left\| W_{out} \cdot r^{*}(t) - Y(t + \tau) \right\| + \alpha \| W_{out} \|, \qquad (3)$$

After this reservoir neural network has been trained, we can use it to estimate $b(t)$, where the estimated value is noted as $\hat{b}(t)$.

**2.2.2 Back propagation based artificial neural network**

Here, the used BP artificial neural network is a traditional neural computing framework which has been widely used in climate research (Chattopadhyay et al., 2019; Watson, 2019; Reichstein et al., 2019). There are six layers in the BP neural network: the input layer with 8 neurons; 4 hidden layers with 100 neurons each; the output layer with 8 neurons. In each layer, the connectivity weights of the neurons need to be computed during training process, where the back propagation optimization with the complicated gradient decent algorithm is used (Dueben and Bauer, 2018). A crucial difference between the BP and the RC neural networks is as follows: unlike RC, all neuron states of the BP neural network are independent on the temporal variation of time series (Chattopadhyay et al., 2019; Reichstein et al., 2019), while the neurons of RC can track temporal evolution (such as the neuron state $r(t)$ in Fig. 2) (Chattopadhyay et al., 2019). If $a(t)$ and $b(t)$ are two time series of a system, through the BP neural network, we can also reconstruct $b(t)$ from $a(t)$.

**2.2.3 Long short-term memory neural network**

The LSTM neural network is an improved recurrent neural network to deal with time series (Reichstein et al., 2019; Chattopadhyay et al., 2019). As Fig. 3 shows, LSTM has a series of components: a memory cell, input gate, output gate, and a forget gate in addition to the hidden state in traditional recurrent neural network. When a time series $a(t)$ is input to train this neural network, the information of $a(t)$ will flow through all these components, and then the parameters at different components will be computed for fitting the relation between $a(t)$ and $b(t)$. The govern equations for the LSTM architecture are shown in the Appendix. After the training is accomplished, $a(t)$ can be used to reconstruct $b(t)$ by this neural network.

[Figure]

**Figure 3** Schematic of the LSTM architecture. LSTM has a memory cell, input gate, output gate, and a forget gate to control the information of the previous time to flow into the neural network.

The crucial improvement of LSTM on the traditional recurrent neural network (Reichstein et al., 2019) is, that LSTM has the forget gate which controls the information of the previous time to flow

| 220 | into the neural network. This will make the neuron states of LSTM have ability to track the temporal |
| 221 | evolution of time series (Chattopadhyay et al., 2019; Kratzert et al., 2019; Reichstein et al., 2019), |
| 222 | which is also the crucial difference between the LSTM and the BP neural networks. |
| 223 | Here, we also test the LSTM neural network without the forget gate, and call it LSTM[*]. This |
| 224 | means that the information of the previous time cannot flow into the LSTM[*] neural network, which |
| 225 | does not have the memory for the past information. We will compare the performance of LSTM |
| 226 | with that of LSTM[*], so that the role of the neural network memory for the previous information can |
| 227 | be presented. |

**28.** The CCM method has been introduced in the Results section. It should be introduced in the Methodology section. In the discussion of CCM method, relate it with the direction of reconstruction as well (explanatory variable to reconstructed variable)

**Response:** Thank you! We added the description of the CCM algorithm into the method part of the revised manuscript, and also related it with the direction of reconstruction. Our modification is shown by the following screenshot:

| 245 | **2.4.2 Convergent cross mapping** |
| 246 | To measure the nonlinear coupling relation between two observational variables, we choose the |
| 247 | convergent cross mapping method that has been demonstrated to be useful for many complex |
| 248 | nonlinear systems (i.e. Sugihara et al., 2012; Tsonis et al., 2018; Zhang et al. 2019). Considering $a(t)$ |
| 249 | and $b(t)$ as two observational time series, we begin with the cross mapping (Sugihara et al., 2012) |
| 250 | from $a(t)$ to $b(t)$ through the following steps: |
| 251 | i) Embedding $a(t)$ (with length $L$) into the phase space with a vector |
| 252 | $M_a(t_i) = \{a_{t_i},\ a_{t_i - \tau_0},\ \ldots,\ a_{t_i - (m-1)\tau}\}$ ("$t_i$" represents a historical moment in the observations), where |
| 253 | embedding dimension ($m$) and time delay ($\tau$) can be determined through the false nearest neighbor |
| 254 | algorithm (Hegger and Kantz, 1999). |
| 255 | ii) Estimating the weight parameter $w_i$ which denotes the associated weight between two vectors |
| 256 | "$M_a(t)$" and "$M_a(t_i)$" ("$t$" denotes the excepted time in this cross mapping), defined as: |
| 257 | $w_i = \frac{u_i}{\sum_{i=1}^{m+1} u_i}$,     (7) |
| 258 | $u_i = exp\{-\frac{d\,[M_a(t), M_a(t_i)]}{d\,[M_a(t), M_a(t_1)]}\}$,     (8) |
| 259 | where $d\,[M_a(t), M_a(t_i)]$ denotes the Euler distance between vectors "$M_a(t)$" and "$M_a(t_i)$". The |

nearest neighbor to "$M_a(t)$" generally corresponds to the largest weight.

iii) Cross mapping the value of $b(t)$ by

$$\hat{b}(t) = \sum_{i=1}^{m+1} w_i b(t_i). \tag{9}$$

$\hat{b}(t)$ denotes the estimated value of $b(t)$ with this phase-space cross mapping. Then, we will evaluate the cross mapping skill (Sugihara et al., 2012; Tsonis et al., 2018) as the follows:

$$\rho_{a \to b} = corr. [b(t), \hat{b}(t)] \tag{10}$$

The cross mapping skill from $b$ to $a$ is also measured according to the above steps, marked as $\rho_{b \to a}$.

Sugihara et al. and Tsonis et al. ever defined the causal inference according to $\rho_{a \to b}$ and $\rho_{b \to a}$ like that: (i) if $\rho_{a \to b}$ is convergent when $L$ is increased, and $\rho_{a \to b}$ is of high magnitude, then $b$ is suggested to be a causation of $a$. (ii) Besides, if $\rho_{b \to a}$ is also convergent when $L$ is increased, and is of high magnitude, then the causal relationship between $a$ and $b$ is bidirectional ($a$ and $b$ cause each other). In our study, all values of the CCM indices are measured when they are convergent with the data length (Tsonis et al. 2018).

According to literature (Sugihara et al., 2012; Ye et al., 2015), the CCM index is related to the ability of using one variable to reconstruct another variable: if $b$ influence $a$ but $a$ does not influence

$b$, the information content of $b$ can be encoded in $a$ (through the information transfer from $b$ to $a$), but the information content of $a$ is not encoded in $b$ (there exists no information transfer from $a$ to $b$).

Therefore, the time series of $b$ can be reconstructed from the records of $a$. For the CCM index ($\rho_{a \to b}$), its magnitude represents how much information content of $b$ is encoded in the records of $a$.

Therefore, the high magnitude of $\rho_{a \to b}$ means that $b$ causes $a$, and we can get good results of reconstruction from $a$ to $b$. In this paper, we will test the association between the CCM index and the reconstruction performance of machine learning.

29. Otherwise it is a little confusing to relate the notation of with its notation when it is being applied and shown in the Results section (Line number 462-463).

**Response:** Thank you! We modifed this narration, and improved such narration thoroughly in the revised manuscript. Our modification is shown by the following screenshot:

machine learning to reconstruct these climate series. The CCM index of that NHSAT cross maps

TSAT is 0.70, and the CCM index of that TSAT cross maps NHSAT is 0.24 (Table 4). The CCM

index means that the information content of TSAT is well encoded in the records of NHSAT, and the information transfer might be mainly from TSAT to NHSAT, which is consistent with previous studies (Farneti and Vallis, 2013). Further, the CCM analysis indicates that the reconstruction from

NHSAT to TSAT might obtain a better quality than the opposite direction.

6. The same goes for the description of Pearson's correlation coefficient, its description should be shifted from the Results to the Methodology section.

**Response:** Thank you! We moved the description of Pearson's correlation to the method in the revised manuscript. Our modification is shown by the following screenshot:

**2.4 Coupling detection**

**2.4.1 Linear correlation**

As the introduction mentioned, the linear Pearson correlation is a commonly-used method to quantify the linear relationship between two observational variables. The Pearson correlation between two series $a(t)$ and $b(t)$, is defined as

$$corr. = \frac{mean[(a-\bar{a}) \cdot (b-\bar{b})]}{std(a) \cdot std(b)}. \qquad (6)$$

The symbols "*mean*" and "*std*" denote the average and standard deviation for series $a(t)$ and $b(t)$, respectively.

7. The flow of the Results section is hard to follow. **The Results section just lists the author's observations, from the Figures and Tables, and does not provide any insights into those observations. For example, line number 329 - 330 states that BP and LSTM\* are not sensitive to non-linear coupling, but no explanation is given as to why this is so.** The authors should provide more insight into the observed behavior of the ML algorithms mentioned in the Results section.

**Response:** Thank you! We provided more insights into the observed behavior of the ML algorithms mentioned in the Results section. For the analysis on other results, we paid more attention. Please see our revised manuscript.

For the results of that BP and LSTM\* are not sensitive to non-linear coupling, their algorithms might be responsible to this. **When analyzing their algorithm, we can find that the BP neural network cannot track the temporal evolution, because its neuron states are independent to the temporal variation of time series. For LSTM\*, it cannot include the information of previous time. Previous studies have revealed that the temporal evolution and memory are crucial properties for the nonlinear time series** [1, 2]**,** which should be considered when modeling nonlinear dynamics. But the algorithms of RC and LSTM have made improvements on these issues (we have added these contents into the method part of the revised manuscript).

[1] Kantz, H., Schreiber, T.: Nonlinear time series analysis (Vol. 7). Cambridge university press, 2004.

[2] Franzke C. L., Osprey, S. M., Davini, P., Watkins, N. W.: A dynamical systems explanation of the Hurst effect and atmospheric low-frequency variability. Sci. Rep., 5, 9068, 2015.

Our modification is shown by the following screenshot:

| 411 | As mentioned in section 2.2, a BP neural network does not track the temporal evolution, since |
| 412 | its neuron states are independent to the temporal variation of time series. For LSTM*, it   does not |
| 413 | include the information of previous time. Previous studies have revealed that the temporal evolution |
| 414 | and memory are very important properties for a nonlinear time series (Kantz and Schreiber, 2003; |
| 415 | Franzke et al. 2015), which could not be neglected when modeling nonlinear dynamics. These might |
| 416 | be responsible for that BP and LSTM* fail in dealing with this nonlinear Lorenz 63 system. |
| 417 | Investigations for the application of BP in other different nonlinear relationships needs to be further |
| 418 | addressed in the future. |

8.  The conclusion section should be shortened.

**Response:** Thank you! We shortened the length of the conclusion, and moved part of the discussion into the results part. Please see our revised manuscript.

9.  Although the work is interesting and has a lot of future scope, the above concerns prevents me from recommending this work for publication in its current form. I hope the authors would incorporate the suggestions and rewrite the manuscript.

**Response:** Many thanks for your comments and suggestions! We carefully improved the detail descriptions, and thoroughly rewrote the manuscript according to your suggestions.

Specific Points:

10. Lines 43-46: The climate problems mentioned here are actually applications of climate data.

**Response:** Thank you! We modified this narration. Our modification is shown by the following screenshot:

Neural network-based machine learning provides effective tools for studying climatic data (Reichstein et al., 2019), which attracts great attention recently. The machine learning approach is widely applied to downscaling and data mining analyses (Mattingly et al., 2016; Racah et al., 2017), and it can be also used to predict the time series of climate variables, such as temperature, humidity, runoff and air pollution (Zaytar and Amrani, 2016; Biancofiore et al., 2017; Kratzert et al., 2019;

Feng et al., 2019). Recently, it is demonstrated that a large potential application of machine learning is to reconstruct the temporal dynamics of complex systems (Pathak et al., 2017; Du et al., 2017;

11. Lines 52-54: Re-write this sentences to make it intuitive. For example, this line: "...while the physics of systems is suggested for consideration" feels like it refers to the study by Watson, 2019, where neural network based algorithm is used to augment a physics based model to improve its performance. However, this is not clear from the text.

**Response:** Thank you! We modified this narration. Our modification is shown by the following screenshot:

Neural network-based machine learning provides effective tools for studying climatic data (Reichstein et al., 2019), which attracts great attention recently. The machine learning approach is widely applied to downscaling and data mining analyses (Mattingly et al., 2016; Racah et al., 2017), and it can be also used to predict the time series of climate variables, such as temperature, humidity, runoff and air pollution (Zaytar and Amrani, 2016; Biancofiore et al., 2017; Kratzert et al., 2019;

Feng et al., 2019). Recently, it is demonstrated that a large potential application of machine learning is to reconstruct the temporal dynamics of complex systems (Pathak et al., 2017; Du et al., 2017;

Watson, 2019). Studies (Pathak et al., 2017; Lu et al., 2018; Carroll, 2018) have shown that the chaotic attractors in Lorenz system and Rossler system can be described by machine learning. Since chaos is the key property of the underlying climate system giving rise to climatic time series (Lorenz,

1963; Patil et al., 2001), these studies provide a theoretical explanation why the machine learning can be well applied in reconstructing climate temporal dynamics.

12. Lines 63-64: The statement infers that, since linear correlation is an intrinsic assumption of traditional statistical methods, cross-correlation analysis should be carried out for investigating the performance of ML algorithms. This is not a valid reasoning, as the approach of ML algorithms and traditional statistical methods are very different.

**Response:** Thank you! We will modify this narration. Our modification is shown by the following screenshot:

| 55 | Though applying machine learning to climatic series attracts much attention, it is still open |
|---|---|
| 56 | questions what can be learnt by machine learning during the training process, and what is the key |
| 57 | factor determining the performance of machine learning approach to climatic time series. This is |
| 58 | crucial for investigating why machine learning cannot perform well with some datasets, and how to |
| 59 | improve the performance for them. One possible key factor is the coupling between different |
| 60 | variables. Because different climate variables are coupled with one another (Donner and Large, |
| 61 | 2008), and the coupled variables will share their information content with one another through the |
| 62 | information transfer (Takens, 1981; Schreiber, 2000; Sugihara et al., 2012). Furthermore, a coupling |
| 63 | often results in that the observational time series are statistically correlated (Brown, 1994). |
| 64 | Correlation is a crucial property for the climate system, and often influences the climatic time series |
| 65 | analysis. "Pearson Coefficient" is often used to detect the correlation, which only detects the linear |
| 66 | correlation. It is known that when the Pearson correlation coefficient is weak, most of traditional |
| 67 | regression methods will fail in dealing with the climatic data, such as fitting, reconstruction and |
| 68 | prediction (Brown, 1994; Sugihara et al., 2012; Emile-Geay and Tingley, 2016). However, a weak |
| 69 | linear correlation does not mean that there is no coupling relation between the variables. Previous |
| 70 | studies (Sugihara et al., 2012; Emile-Geay and Tingley, 2016) have suggested that, although the |
| 71 | linear correlation of two variables is potentially absent, they might be nonlinearly coupled and can |
| 72 | be exploited by analysis. For instance, the linear cross-correlations of sea surface temperature series |
| 73 | observed in different tropical areas are unstable and vary with time, which leads to an overall weak |
| 74 | linear correlation, but this non-linear correlation is conductive to the better El Niño predictions |

13. Lines 83-87: This part should be there in the Results section. However, this line can be modified to be a hypothesis the authors are trying to check.

**Response:** Thank you! We modified this narration, as the following screenshot shows:

| 112 | Finally, we will discuss a real-world example from climate system. ==It is known that there exist== |
| 113 | ==atmospheric energy transportations between the tropics and the Northern Hemisphere, which results== |
| 114 | ==in the coupling between the climate systems in these two regions== (Farneti and Vallis, 2013). Due to |
| 115 | the underlying complicated processes, it is difficult to use a formula to cover this coupling between |
| 116 | the tropical average surface air temperature (TSAT) series and the Northern Hemispheric surface air |
| 117 | temperature (NHSAT) series. ==We employ machine learning methods to investigate whether the== |
| 118 | ==NHSAT time series can be reconstructed from the TSAT time series, and whether the TSAT time== |
| 119 | ==series can be also reconstructed from the NHSAT time series. Accordingly, the conclusions from our== |
| 120 | ==model simulations can be further tested and generalized.== |

14. Line 105: Typographical error: it should be "Learning" not "Leaning".

**Response:** Thank you! We will modify this typographical error. We will also inspect the manuscript to avoid the any typographical error. Our modification is shown by the following screenshot:

**2.1 ==Learning== coupling relations and reconstructing coupled time series**

15. Figure 1: The big black arrow used to represent (3), is confusing in the sense that the reconstructed time series from the testing stage is being compared with the time series from the training stage, which is not the case.

**Response:** Thank you! We modified this figure, as the following screenshot shows:

[Figure]

**Figure 1** Diagram illustration for reconstructing time series by machine learning. (1) The available part of the

16.

16. Lines 182-183: Mention clearly why an analysis of LSTM* reconstructed time series is required.

**Response:** Thank you! We will modify this narration.

The crucial improvement of LSTM on the traditional recurrent neural network, is that LSTM has the **forget gate** which controls the information of the previous time to flow into the neural network. This also make the neural state of LSTM has ability to track the temporal evolution, which is also the crucial difference between LSTM and BP neural networks.

Here, we also test the LSTM neural network **without the forget gate, and call it LSTM\***. This means that the information of the previous time cannot flow into the LSTM* neural network, which does not have the memory for the past information. **We will compare the performance of LSTM with that of LSTM\*, so that the role of the neural network memory for the previous information can be demonstrated.**

Our modification is shown by the following screenshot:

| 218 | The crucial improvement of LSTM on the traditional recurrent neural network (Reichstein et al., |
| 219 | 2019) is, that LSTM has the forget gate which controls the information of the previous time to flow |
| 220 | into the neural network. This will make the neuron states of LSTM have ability to track the temporal |
| 221 | evolution of time series (Chattopadhyay et al., 2019; Kratzert et al., 2019; Reichstein et al., 2019), |
| 222 | which is also the crucial difference between the LSTM and the BP neural networks. |
| 223 | Here, we also test the LSTM neural network without the forget gate, and call it LSTM*. This |
| 224 | means that the information of the previous time cannot flow into the LSTM* neural network, which |
| 225 | does not have the memory for the past information. We will compare the performance of LSTM |
| 226 | with that of LSTM*, so that the role of the neural network memory for the previous information can |
| 227 | be presented. |

17. Lines 201-203: The introduction of the parameters, p, d, and q is not proper and causes confusion. Rewrite the sentence.

**Response:** Thank you! We modified this narration, as the following screenshot shows:

284 **A linearly coupled model:** The autoregressive fractionally integrated moving average

285 (ARFIMA) model (Granger and Joyeux, 1980) maps a Gaussian white noise $\varepsilon(t)$ into a correlated

286 sequence $x(t)$ (Eq. (11)), which could simulate the linear dynamics of oceanic-atmospheric coupled

287 system (Hasselmann, 1976; Franzke, 2012; Massah and Kantz, 2016; Cox et al., 2018).

288 $\varepsilon(t) \xrightarrow{ARFIMA(p,d,q)} x(t)$  (11)

289 In this model, $d$ is a fractional differencing parameter, and $p$ and $q$ are the orders of the

290 autoregressive and moving average components. Here, the parameters are set as: $p = 3$, $d = 0.2$ and $q$

291 $= 3$. Hence $x(t)$ is a time series composited with three components: the third-order autoregressive

292 process whose coefficients are 0.6, 0.2 and 0.1, the fractional differencing process whose Hurst

293 exponent is 0.7, and the third-order moving average process whose coefficients are 0.3, 0.2 and 0.1

294 (Granger and Joyeux, 1980). These two time series $\varepsilon(t)$ and $x(t)$ are used for the reconstruction

295 analysis.

18. Lines 205-206: x(t) and the Gaussian noise () time series are the two time series being used for the coupled analysis. This has to be mentioned clearly in the text. This comment goes for all the cases of coupled time series being used (non-linear, higher order non-linear, real world scenario).

**Response:** Thank you! We mentioned this information for all the used data in the revised manuscript, as the following screenshot shows:

284 **A linearly coupled model:** The autoregressive fractionally integrated moving average

285 (ARFIMA) model (Granger and Joyeux, 1980) maps a Gaussian white noise $\varepsilon(t)$ into a correlated

286 sequence $x(t)$ (Eq. (11)), which could simulate the linear dynamics of oceanic-atmospheric coupled

287 system (Hasselmann, 1976; Franzke, 2012; Massah and Kantz, 2016; Cox et al., 2018).

288 $\varepsilon(t) \xrightarrow{ARFIMA(p,d,q)} x(t)$  (11)

289 In this model, $d$ is a fractional differencing parameter, and $p$ and $q$ are the orders of the

290 autoregressive and moving average components. Here, the parameters are set as: $p = 3$, $d = 0.2$ and $q$

291 $= 3$. Hence $x(t)$ is a time series composited with three components: the third-order autoregressive

292 process whose coefficients are 0.6, 0.2 and 0.1, the fractional differencing process whose Hurst

293 exponent is 0.7, and the third-order moving average process whose coefficients are 0.3, 0.2 and 0.1

294 (Granger and Joyeux, 1980). These two time series $\varepsilon(t)$ and $x(t)$ are used for the reconstruction

295 analysis.

**A nonlinearly coupled model:** The Lorenz 63 chaotic system (Lorenz, 1963) depicts the nonlinear coupling relation in a low-dimensional chaotic system. The system reads

$$\frac{dx}{dt} = -\sigma(x-y)$$

$$\frac{dy}{dt} = \mu x - xz - y \qquad\qquad (12)$$

$$\frac{dz}{dt} = xy - Bz$$

When the parameters are fixed at $(\sigma, \mu, B) = (10, 28, 8/3)$, the state in the system is chaotic. We employed the fourth-order Runge-Kutta integrator to acquire the series output from this Lorenz 63

system. The time steps were 0.01. The time series $X(t)$ and $Z(t)$ are used for the reconstruction analysis.

**A high-dimensional model:** The two-layer Lorenz 96 model (Lorenz, 1996) is a high-dimensional chaotic system, which is commonly used to mimic mid-latitude atmospheric dynamics (Chorin and Lu, 2015; Hu and Franzke, 2017; Vissio and Lucarini, 2018; Chen and

Kalnay, 2019; Watson, 2019). It reads

$$\frac{dX_k}{dt} = X_{k-1}(X_{k+1} - X_{k-2}) - X_k + F - \frac{h_1}{J}\sum_{j=1}^{J} Y_{j,k}$$

$$\frac{dY_{k,j}}{dt} = \frac{1}{\theta}[Y_{k,j+1}(Y_{k,j-1} - Y_{k,j+2}) - Y_{k,j} + h_2 X_k]. \qquad (13)$$

In the first layer of the Lorenz 96 system there are 18 variables marked as $X_k$ ($k$ is a integer ranging from 1 to 18), and each $X_k$ is coupled with $Y_{k,j}$ ($Y_{k,j}$ is from the second layer). The parameters are set as fellows: $J = 20$, $h_1 = 1$, $h_2 = 1$, and $F=10$. The parameter $\theta$ can alter the coupling strength: when

$\theta$ is decreased, the coupling strength between $X_k$ and $Y_{k,j}$ will be enhanced. The fourth-order

Runge-Kutta integrator and periodic boundary condition are adopted (that is: $X_0 = X_K$ and $X_{K+1} = X_1$;

$Y_{k,0} = Y_{k-1,J}$ and $Y_{k,J+1} = Y_{k+1,1}$), and the integral time unit was taken as 0.05. The time series $X_1(t)$

and $Y_{1,1}(t)$ are used for the reconstruction analysis.

19. Lines 236-237: The time series are being standardized (mean is zero and standard deviation is one) before

   being used in the reconstruction analysis. Explain why are they standardized.

**Response:** Thank you! We will explain for this processing of standardization.

     For the time series that come from different processes, they might have different variability and units. In order to avoid the disturbance given by such different variability and units, we select to standardize all the time series with uniform mean value and variance.

     Our modification is shown by the following screenshot:

To evaluate the quality of reconstruction by machine learning, the root mean squared error
(RMSE) of residual series (Hyndman and Koehler, 2006) is adopted (Eq. (4)), which represents the
difference between the real series $b(t')$ and the reconstructed series $\hat{b}(t')$. In order to fairly
compare the errors of reconstructing different processes with different variability and units
(Hyndman and Koehler, 2006; Pennekamp et al., 2018; Huang and Fu, 2019), we normalize the
RMSE as Eq. (5) shows.

$$RMSE = \sqrt{\frac{1}{k}\sum_t [b(t') - \hat{b}(t')]^2},$$  (4)

$$nRMSE = \frac{RMSE}{\max[b(t')] - \min[b(t')]}.$$  (5)

**Training and testing datasets:** Before analysis, all the used time series are standardized to
take zero mean and unit variance so that any possible impact of mean and variance on the statistical
analysis is avoided (Brown, 1994; Hyndman and Koehler, 2006; Chattopadhyay et al., 2019). We

20. Lines 275-277: Incorporate the plots for LSTM* in Figure 3c and 3d.

**Response:** Thank you! We will add the results of LSTM[*] into the corresponding figures. Our modification is
shown by the following screenshot:

[Figure]

**Figure 4** (a) The $x(t)$ time series (blue) and the $\varepsilon(t)$ time series (black) of the ARFIMA(3,0.2,3) model. White lines depict the large-scale trends of these time series acquired by 50-step smoothing average. (b) Comparison of the power spectrum of $x(t)$ (blue) with the power spectrum of $\varepsilon(t)$ (black). (c) Comparison of the reconstructed time series of $x(t)$ by RC, LSTM, LSTM[*] and BP respectively (red dots), and the original $x(t)$ time series are presented by the blue lines. (d) Comparison of the reconstructed time series of $\varepsilon(t)$ by RC, LSTM, LSTM[*] and BP

respectively (red dots), and the original $\varepsilon(t)$ time series are presented by the black lines. Only partial segments of

21. Lines 286-297: The information about convergent cross mapping (CCM) should be introduced in the methodology section in detail. Are there other methods for estimating non-linear correlation or causality between two time-series. If so, why CCM was specifically used.

**Response:** Thank you! We will move the detailed description of CCM to the method part.

Apart from CCM, the Granger method [1] and transfer entropy [2] can be also used to measure the causality. However, it has been demonstrated that the Granger causality cannot measure the causality or coupling in nonlinear systems [3]. Transfer entropy can be an alternative choice to measure the nonlinear coupling. But the index value of transfer entropy often ranges from 0 to 3 [4], while the CCM index always ranges from 0 to 1, so that it is often hard to judge if transfer entropy is strong or weak. In previous studies [5], the CCM index has been successfully used to measure the nonlinear coupling strength and causality in many kinds of complex systems. However, it is worth to make comparisons for CCM, transfer entropy and machine learning performance in the future study.

[1] Granger C. W.: Investigating causal relations by econometric models and cross-spectral methods. Econometrica 37, 424-438, 1969.
[2] Schreiber T.: Measuring information transfer. Phys Rev Lett 85(2), 461, 2000.
[3] Malevergne Y., Sornette D.: Extreme financial risks: From dependence to risk management. Springer Science & Business Media, 2006.
[4] Paluš, M.: Multiscale atmospheric dynamics: cross-frequency phase-amplitude coupling in the air temperature. Phys Rev Lett, 112(7), 078702, 2014.
[5] Tsonis A. A., Deyle E. R., Ye H., Sugihara G.: Convergent cross mapping: theory and an example. In Advances in Nonlinear Geosciences (pp. 587-600). Springer, Cham, 2018.

Our modification is shown by the following screenshot:

According to literature (Sugihara et al., 2012; Ye et al., 2015), the CCM index is related to the
ability of using one variable to reconstruct another variable: if $b$ influence $a$ but $a$ does not influence
$b$, the information content of $b$ can be encoded in $a$ (through the information transfer from $b$ to $a$),
but the information content of $a$ is not encoded in $b$ (there exists no information transfer from $a$ to $b$).
Therefore, the time series of $b$ can be reconstructed from the records of $a$. For the CCM index
($\rho_{a \to b}$), its magnitude represents how much information content of $b$ is encoded in the records of $a$.
Therefore, the high magnitude of $\rho_{a \to b}$ means that $b$ causes $a$, and we can get good results of
reconstruction from $a$ to $b$. In this paper, we will test the association between the CCM index and the
reconstruction performance of machine learning.

22. Lines 390-392: Explain the decrease in LSTM nRMSE with an increase in CCM. As, this behavior is contradictory to the LSTM's nRMSE behavior in the other cases.

**Response:** Thank you! We will supplement the explanation for this.

For all cases of RC results, when the CCM index is increasing, the nRMSE will be decreasing. Likewise, for most cases of LSTM results, when the CCM index is increasing, the nRMSE will be decreasing.

But in this case for LSTM, the relation between CCM and nRMSE is not like the normal cases. The reason might be that the used time series ($X_1$ and $X_2$ of Lorenz 96 system) have the time-varying local mean values (i. e. in the previous time period, the local mean value of time series is 0, and then in the next time period, the local mean value of time series is 0.5), and this influences the performance of LSTM.

We found that the time-varying mean values in time series tend to impact the performance of LSTM. For example, in a time series, at the previous time period, the local mean value of time series is 0, and then at the next time period, the local mean value of time series is 0.5. In this case, LSTM tends to perform badly, and the nRMSE might be increased. **The reason might be that the LSTM algorithm always requires incorporating the time-series values at previous time points (the memory for past time points), and then the varied local mean value of time series will easily influence the results of LSTM.**

However, we have not been able to ensure that this is the only reason. More investigations are needed in the further study. Our modification is shown by the following screenshot:

The reconstruction between $X_1$ and $X_2$ in the same layer of Lorenz 96 system is also shown.

There is an asymmetric causal relation ($\rho_{X_2 \to X_1} = 0.37$ and $\rho_{X_1 \to X_2} = 0.25$) between $X_1$ and $X_2$, and their linear correlation is very weak (see Table 3). The RC gives better result of reconstructing $X_1$

from $X_2$ (nRMSE=0.13) than reconstructing $X_2$ from $X_1$ (nRMSE=0.17). LSTM also has different results for $X_1$ and $X_2$ (Table 3), where the quality of reconstructing from $X_1$ to $X_2$ (nRMSE=0.16) is better than reconstructing from $X_2$ to $X_1$ (nRMSE=0.20). In this case, the reconstruction quality of

LSTM is worse than the RC, and the reconstruction results by LSTM are not consistent with the indication of the CCM index. A previous study (Chattopadhyay et al., 2019) also suggests that

LSTM performs worse than RC in some cases, and this might be related to only a simple variant of the LSTM architecture used. So in this high-dimensional system, the reconstruction quality is also influenced by the chosen explanatory variables: The quality of reconstructing $X_1$ from $Y_{1,1}$ is better than the quality of reconstructing $X_1$ from $X_2$ by RC and LSTM (see Fig. 7b and 7c).

23. Lines 407-408: Explain how did the authors arrive at this statement. RC and LSTM performed better than LSTM* and BP in the linearly coupled system. And BP and LSTM* were not part of the analysis of the high dimensional lorenz-96 analysis. However, this statement can be the conclusion of this section, which shows the sensitivity of RC and LSTM to different coupling strength.

**Response:** Thank you! We modified this narration. In our previous manuscript, the expected meaning of this statement was not a conclusion, but was used to open the topic of this subsection.

We thoroughly rewrote this section in the revised manuscript, please see our revised manuscript. Part of them is as the following screenshot shows:

| | |
|---|---|
| 509 | **4.2.3 Performance of BP and LSTM* in Lorenz 96 system** |
| 510 | Since that BP and LSTM* cannot track the temporal evolutions of a nonlinear time series, in |
| 511 | the above cases of nonlinear system, we did not obtain similar result to RC and LSTM (not shown |
| 512 | here). Here we present a simple experiment, to illustrate what might influence the performances of |
| 513 | BP and LSTM* in a nonlinear system. |
| 514 | The experiment is set as follows: in Eq. (13), the value of $h_1$ is set as 0, and the value of $\theta$ is |
| 515 | decreased from 0.7 to 0.3. When $\theta$ is equal to 0.7, the forcing from $X_1$ to $Y_{1,1}$ is weak. At that time, |
| 516 | the Pearson correlation between $X_1$ and $Y_{1,1}$ is only 0.48, and the performances of BP and LSTM* |
| 517 | are not good. When $\theta$ is equal to 0.3, the forcing is dramatically magnified. As the second panel of |
| 518 | Fig. 9a shows, this strong forcing makes $Y_{j,i}$ synchronized to $X_i$, and the Pearson correlation between |
| 519 | $X_1$ and $Y_{1,1}$ is greatly increased to 0.8. When the forcing strength is magnified, the performance of |

24. Lines 416-420: Examine LSTM for its behavior with change in θ, like the one done for the behavior of LSTM*. This will probably give more insight into the behavior of LSTM*.

**Response:** Thank you! In this case of reconstructing $X_1$ from $Y_{1,1}$ (Lorenz 96 system), all the results of LSTM and RC are almost overlapped with each other. We will supplement the results of LSTM in the revised manuscript.

Our modification for this part is shown by the following screenshot:

[Figure]

**Figure 9** Influence of strong nonlinear coupling on linear Pearson correlation and machine learning performances.

(a) Comparison of the linear correlation when the coupling strength is different. The top panel corresponds to the weak coupling strength, and the bottom panel corresponds to the strong coupling. The red lines present the input explanatory variable and the black lines present the target series of machine learning. (b) Comparison of the machine learning performances when the coupling strength is different. The top panel corresponds to the weak coupling strength, and the bottom panel corresponds to the strong coupling. The black lines are the original series; the reconstructed series by RC (green lines), LSTM*(blue lines) and BP (red dots) are shown respectively. In this case, the results of LSTM are overlapped with that of RC.

25. Line 430: Why is RC not sensitive to Pearson's correlation.

**Response:** Thank you! Here the RC was applied to the nonlinear Lorenz 96 system. It is known that the linear Pearson correlation cannot explain the true dynamical relation in a nonlinear coupled system [1-2]. As the method mentioned, the RC and LSTM can track the temporal evolution and memory of the time series, and then they might rely on the nonlinear dynamics rather than the Pearson correlation. We thoroughly rewrote this section in the revised manuscript, please see our revised manuscript.

[1] Malevergne Y., Sornette D.: Extreme financial risks: From dependence to risk management. Springer Science & Business Media, 2006.

[2] Sugihara, G., May, R., Ye, H., Hsieh, C. H., Deyle, E., Fogarty, M., Munch, S.: Detecting causality in complex ecosystems. Science, 338(6106), 496-500, 2012.

26. Figure 8: It is missing the R2 and p-value of LSTM. The behavior of LSTM should also be evaluated in the same manner.

**Response:** Thank you! We added the results of LSTM into this figure. Our modification is shown by the following screenshot:

[Figure]

**Figure 8** Scatter plot of nRMSE values and CCM index values. The blue boxes are results of the RC machine learning, and the black cycles are results of the LSTM machine learning. The blue and grey dashed lines are the fitted linear trends of the blue boxes and black cycles respectively, and these two dependency trends are both significant because their p-values are both smaller than 0.05.

27. Lines 472-473: What do you mean by unstable variance, elaborate.

**Response:** Thank you! We will supplement the explanation for this.

For the real-world time series (such as the time series in figure R1), the local mean value and the local variance of the time series, are often time-varying. For example, in a time series, at the previous time period, the local mean value of time series is 0, and then at the next time period, the local mean value of time series is 0.5; at the previous time period, the local variance of time series is 1, and then at the next time period, the local variance of time series is 1.5.

[Figure]

Figure R1: Daily time series of the Tropical surface air temperature, the Northern Hemispheric surface aire temperature, and the Nino 3.4 index.

We found that the time-varying local mean value and local variance in time series tend to impact the performance of LSTM. In this case, LSTM tends to perform badly, and the nRMSE might be increased.

**The reason might be that the LSTM algorithm always requires incorporating the time-series values in previous time points (the memory for past time points), and then the varied local mean value of time series will easily influence the results of LSTM. Likewise, the varied local mean value of time series will also influence the results of LSTM.**

However, we have not been able to ensure that this is the only reason. More investigations are needed in the future study. Our modification in this part is shown by the following screenshot:

| 561 | When using TSAT to reconstruct the time series of NHSAT, the reconstructed time series cannot |
| 562 | describe the original time series of NHSAT (Fig. 11c), and the corresponding nRMSE is equal to |
| 563 | 0.21. Besides, we also use LSTM and BP to reconstruct these natural climate series, the |
| 564 | performances of these two neural networks are worse than RC (Table 4). For BP, this might be due |
| 565 | to its inability to deal with nonlinear coupling (As mentioned in method, the BP neurons cannot |
| 566 | track the temporal evolution of a time series). LSTM performs worse than RC in this real-world case |
| 567 | might be induced by the used simple variant of LSTM architecture. |

**Reply to the comments of Dr. Zhixin Lu**

**The comments of Dr. Zhixin Lu:**

In this paper, the authors studied the variable reconstruction problem with several machine learning methods, and test with simulations on several artificial climate models (Lorenz 63 and Lorenz 96) as well as real-world climate data. The authors innovatively use the convergent cross mapping (CCM) to estimate the nonlinear coupling relation between different variables and explain the reason why the variable reconstruction has direction dependence.

This paper is in general well written with sufficient simulations that support its conclusions. However, two main issues need to be addressed.

**Response:** Many thanks for your comments and suggestions. We are willing to revise the method description and discuss the association between "nonlinear observability" and "CCM" in our revised manuscript.

Additionally, we also would like to make response to the two questions of Dr. Zhixin Lu in the following.

1. In Sec. 2.2, the authors introduce the reservoir computing method (Lu et al., 2017) for the variable reconstruction problem. However, I find this introduction very confusing. It seems that different constructions of reservoir computers for different tasks (for reservoir observer or for predicting future of time seriers) are introduced as different layers for a single reservoir. (lines 144-150). It is also confusing why one would need the so-called prediction reservoir as a layer for this reservoir observer task. (lines 175-178) Does this closed-loop reservoir really being used in the simulation in this paper? If so, why is it necessary? A reservoir observer does not need to feedback its own output to its input, as it is simply trying to estimate variable b(t) based on the measured a(t), rather than predicting the future of both a(t) and b(t).

**Response:** Thank you! By means of the first two components shown in Figure 1*, the $a(t)$ is trained and then $\psi[r^*(t)]$ is obtained. In this procedure, the value of $\psi[r^*(t)]$ is already very close to the value of $b(t)$.

Then, if $\psi[r^*(t)]$ is feedback to function "$f$" and "$\psi$", this repetitive operation might make the value of $\psi[r^*(t)]$ more close to the value of $b(t)$. Actually we also found this repetitive operation no longer influenced the results. This is to say, that the third component shown in Figure 1* might be redundant in this reconstruction framework, and the first two components are enough. In the revised manuscript, we will carefully modify the diagram and the introduction of Reservoir computer according to the introduction in *Lu et al.* 2017 [1].

[Figure]

Figure 1* The schematic of Reservoir computer in the previous manuscript (we will revised this figure in the revised manuscript).

[1] Lu Z, Pathak J, Hunt B, Girvan M, Brockett R, Ott E. Reservoir observers: Model-free inference of unmeasured variables in chaotic systems. Chaos 27(4), 041102 (2017).

2. The authors in Sec. 3.2.1-3.2.2 discuss the nonlinear coupling relation, which is essentially the nonlinear observability in the control theory, as being pointed out in (Lu et al., 2017). This direction dependence can be explained by the nonlinear observability. For example, in the Lorenz 63 model, due to the symmetry of that ODE system, both (x(t), y(t), z(t)) and (-x(t),-y(t), z(t)) are solutions on the same chaotic attractor. Thus, one can not construct any nonlinear state-observer that estimates the value of x or y given the time series of variable z. However, a state observer can estimate z(t) given either x(t) or y(t). It was also shown that $x^2(t)$ and $y^2(t)$ can be estimated given z(t) as it is nonlinearly observable. The authors employ CCM to quantify the "nonlinear coupling relation" and show that it is better than a linear coupling relation. It is the reviewer's opinion that a brief discussion of the relation between the CCM and the nonlinear observability should be given. Is CCM essentially the same as nonlinear-observability? If not, what is the difference?

**Response:** Thank you!

Referred to the literature [1-6], we found that the meanings of "nonlinear observability" and "CCM" are partially close to each other: "Nonlinear observability":

For two variables $x_0$ and $x_1$ , their time series follows that: $x_0(t) \in U$ and $x_1(t) \in \Sigma$. If they are from nonlinear systems, it is a general fact that $\Sigma$ restricted to U is not necessarily complete [1]. Hence, *Hermann and Krener* 1977 [1] demonstrated that $x_0(t)$ might be not able to totally recover the values of $x_1(t)$. Then, the asymmetry reconstruction between $x_0(t)$ and $x_1(t)$ is common for nonlinear systems, which is also called "estimability" and is discussed in the previous paper [2-3].

**"CCM":** The convergent cross mapping (CCM) coefficient is a kind of causality index [4]. Takens 1981 [5] proposed that: for two variables $x$ and $y$, if $x$ does influence $y$ in the dynamical system, the value of $x$ can be recovered from the records of *y*.

Further, Sugihara et al. 2012 [4] demonstrated this theorem of Takens determines the reconstruction between two variables: for two variables *x* and *y*, if *x* does influence *y* in the dynamical system (but *y* does not influence *x*), the information of *x* will be transferred into *y*, and so that the records of *y* will be able to recover the values of *x*. However, this information transfer between *x* and *y* is asymmetry, and then the reconstruction between *x* and *y* will be also asymmetry. Hence, the CCM index is proposed to measure such asymmetry information transfer between the observational variables [4, 6].

The "nonlinear observability" is often measured for the nonlinear system with known mathematical equation. For the observational records from real-world system without known mathematical equation, the "nonlinear observability" might be hard to be measured. However, the CCM coefficient can be used to measure asymmetry information transfer between the observational variables in different real systems [4, 6].

Additionally, we also used CCM to analyze the "nonlinear observability" in the Lorenz 63 system. As Figure 2* and table 1* show, when using $z(t)$ to reconstruct $x(t)$, the reconstructed series largely deviates from the real $x(t)$. However, when using using $z(t)$ to reconstruct $[x(t)]^2$ or $x(t)*y(t)$, the reconstruction errors are much smaller. As Table 1* shows, we measured the CCM coefficient for $z(t)$ and $x(t)$, $z(t)$ and $[x(t)]^2$, and $z(t)$ and $x(t)*y(t)$ respectively, they are equal to 0.03, 0.95, and 0.91 respectively. Such results of CCM coefficient are really close to the analysis of "nonlinear observability".

We will discuss such association between "nonlinear observability" and "CCM" in the revised manuscript.

[1] Hermann R, Krener A. Nonlinear controllability and observability. IEEE Transactions on automatic control, 22(5), 728-740 (1977).

[2] Lu Z, Pathak J, Hunt B, Girvan M, Brockett R, Ott E. Reservoir observers: Model-free inference of unmeasured variables in chaotic systems. Chaos 27(4), 041102 (2017).

[3] Schumann-Bischoff J, Luther S, Parlitz U. Estimability and dependency analysis of model parameters based on delay coordinates. Phys Rev E, 94(3), 032221 (2016).

[4] Takens, F.: Detecting strange attractors in turbulence. Dynamical Systems and Turbulence, Lecture Notes in Mathematics, 898, 366–381 (Springer Berlin Heidelberg) (1981).

[5] Sugihara, G, May R, Ye H, Hsieh CH, Deyle E, Fogarty M, Munch S. Detecting causality in complex ecosystems. Science, 338(6106), 496-500 (2012).

[6] Tsonis AA, Deyle ER, Ye H, Sugihara G. Convergent cross mapping: theory and an example. In Advances in Nonlinear Geosciences (pp. 587-600), Springer, Cham., (2018).

[Figure]

Figure 2* (a) The results of applying RC to reconstruct $x(t)$ from $z(t)$ (Lorenz 63 system). (b) The results of applying RC to reconstruct $[x(t)]^2$ from $z(t)$. (c) The results of applying RC to reconstruct $x(t)*y(t)$ from $z(t)$. The blue lines denote the real time series, and red lines represent the reconstructed series through the RC machine learning.

**Table 1\*** Details of Lorenz63 system reconstruction

| Input (*a*) | Output (*b*) | CCM index $\rho_{a \to b}$ | Data length (training/testing) | Neural network | RMSE |
|---|---|---|---|---|---|
| *Z(t)* | *X(t)* | 0.03 | 2400/1600 | RC | 1.13 |
| *Z(t)* | *X(t)*$^2$ | 0.95 | 2400/1600 | RC | 0.01 |
| *Z(t)* | *X(t)\*Y(t)* | 0.91 | 2400/1600 | RC | 0.01 |

---

## Referee Report (RR1)

Reconstructing Coupled Time Series in Climate Systems using Three Kinds of Machine Learning Methods

Anonymous Reviewer 2:

I would like to appreciate the effort of the authors for including my specific suggestions and answering my queries. The result and discussion section looks good to me. However, I have some issue with the presentation of the manuscript. There are English language errors in the manuscript, which dilute the impact of the manuscript in some areas and can cause confusion to the reader. Please find my major comments below.

1. In response to my specific comment No. 22, the explanation associates the presence of non-stationarity in time series, a time varying local mean, to the performance of LSTM. However, the same has not been added in the main manuscript, instead it has been associated to the "simple architecture of LSTM". That feel vague. Although, I agree with the authors' point that it would require further analysis to establish non-stationarity as the sole reason for the performance of LSTM. But, it should also be mentioned in the manuscript, as that will pave the way for future research on this topic.

2. In response to my specific comment No 23, 24, and 25, the authors only provided half of the explanation in the section 4.2.3. Through the comments, I wanted to see the performance of LSTM, RC, BP, and LSTM* with a changing coupling strength ($\theta$). As in the previous manuscript this section did not mention LSTM but claimed that LSTM along with RC to be better than BP and LSTM* in reconstructing the lorenz96 system. Currently, the section talks about BP and LSTM*. And a line is added in the Figure caption that, RC overlaps LSTM, i.e., RC and LSTM have the same

performance. However, it seems the earlier explanation about the insensitivity of RC with respect to correlation and its sensitivity to CCM has been removed (430 - 439; from the last reviewed manuscript).

"*However, the RC is not so much restrained by the Pearson correlation. When $\theta$ is equal to 0.7 or 0.3, the values of CCM index are both higher than 0.9, that is to say, the nonlinear coupling strength is not changed by $\theta$. Then, it can be found that the quality of reconstructed $X_1$ by RC is always good. As Fig. 7b shows, the green dots (RC output) in Fig. 7b always overlap with the black line (original target series). Actually, the reconstruction quality of the RC is determined more by the nonlinear coupling strength. The values of CCM index are calculated between $X_1$ and $X_2$, $X_3$ ..., $X_{18}$; meanwhile, $X_1$ is reconstructed from $X_2$, $X_3$ ..., $X_{18}$, respectively. Then, a significant correspondence exists between the nRMSE and CCM index (Fig. 8), especially for the results of RC. This indicates that the reconstruction quality is dependent on the coupling strength between the reconstructed variable and different explanatory variables.*". I would suggest keep this explanation and add LSTM along with RC here too as both seems to be sensitive to CCM and not Pearson's correlation.

3. I would urge the authors to check the English language thoroughly in the manuscript.

Specific Suggestions

1. Lines 43-44: rewrite.

2. Lines 50 - 54: There seem to be a disconnect between these lines and the preceding line (Line 49). have a look at it.

3. Overall, give another look to the presentation of the matter and its English language.

4. Shorten all the Figure captions. Captions should only include the information of the Figure, not its description. Its description should be added in the main body of the manuscript.

5. Line 212: write "governing" instead of "govern".

6. Lines 229 - 231 can be rewritten as: "*The Root Mean Square Error (RMSE) of residuals is used here to evaluate the quality of reconstruction by machine learning. The residual represents the difference between the real series (b(t')) and the reconstructed series ($\hat{b}(t')$).*"

7. Lines 267 - 268 can be rewritten as: "*Sugihara et al. and Tsonis et al. defined the causal inference according to* $\rho_{a \to b}$ *and* $\rho_{b \to a}$ *as: ...*" follow proper reference format (Sugihara et al and Tsonis et al. Are missing its years).

8. Line 440: Instead of "two-directional", use "bi-directional".

9. Lines 472 - 474: It can be rewritten as: "*In this case, performance of the reconstruction through BP and LSTM\* are not good and it is analyzed in section 4.2.3.*"

10. Lines 510 - 512: The sentence is confusing, please rewrite.

11. Line 538: no need to mention: "*… mentioned in the introduction*".

---

## Referee Report (RR2)

Reconstructing coupled time series in climate systems using three kinds of machine learning methods

Anonymous Reviewer #2

The authors have answered all the technical queries and modified the manuscript based on the suggestions. However, I feel the authors need to check for English language corrections. I am mentioning some specific ones below:

Specific Points:

Line No. 474-478: The following lines should be written as

" Chattopadhyay et al. (2019) also suggests that LSTM performs worse than RC in some cases, and this might be related to the use of a simple variant of the LSTM architecture. This variant of LSTM was tasted and it was found that the time-varying local mean in time series would sometimes influence its performance. However further investigation is required for a deeper understanding of the same. "

Line No. 523 - 528: The following lines can be rewritten as:

"However, RC and LSTM are not restrained by the Pearson correlation in this nonlinear system. When $\theta$ is altered from 0.7 to 0.3, although the Pearson correlation changed a lot, the values of CCM index stayed consistently above 0.9. Throughout the alterations in $\theta$, RC is able to produce a good quality reconstruction of X1. Fig. 9b shows that the reconstructed series by RC and LSTM always overlap with the real time series. Thus it can be inferred that the performance of both RC and LSTM is sensitive to the value of CCM index. This has been analyzed in section 4.2.2."

Line No. 47-48: I suggest rewriting this line as:

"For example, chaos is a crucial property of climatic time series (Lorenz, 1963; Patil et al., 2001)."

Line no. 48-50: The following lines can be rewritten as:

"Thus, their is significant concern regarding the ability of machine learning algorithms to reconstruct the temporal dynamics of the underlying complex systems (Pathak et al., 2017; Du et al., 2017; Lu et al., 2018; Carroll, 2018; Watson, 2019)."

---

## Author Response (AR3)

**Response Letter**

Dear Prof. Dhanya,

We sincerely thank you and all reviewers for concerning our manuscript entitled "Reconstructing coupled time series in climate systems using three kinds of machine learning methods" (ID: esd-2019-63). Your comments are very helpful for revising and improving our paper. We have made revision and correction by taking these comments into account carefully, and we hope all of these revisions meet with approval. Revised changes are marked in yellow in the paper. Below you will find the main revisions and corrections in the paper and the point-to-point responses to the reviewers' comments:

**Reply to the comments of Editor:**

Editor's comments to the Author:

Reviewers have expressed their satisfaction over the technical suitability of the article to be published in ESD. However, they raise serious concerns over the presentation style/ grammatical errors in the manuscript. The manuscript is recommended for minor revision; but I suggest authors to please consider these comments seriously and rework on the presentation of the manuscript. Reviewer comments are enclosed.

**Response:** Many thanks for your comments and suggestions. We carefully addressed all issues proposed by the reviewers, and we made point-to-point responses to these comments in the following. In this revised manuscript, we thoroughly inspected the language errors, figures and formulation mistakes, and we have made correction which we hope meet with approval.

**Reply to the comments of Reviewer 2:**

Comments of anonymous Reviewer 2:

The authors have answered all the technical queries and modified the manuscript based on the suggestions. However, I feel the authors need to check for English language corrections. I am mentioning some specific ones below:

**Response:** Many thanks for your comments and suggestions. In this revised manuscript, we thoroughly inspected the language errors, figures and formulation mistakes, and all revisions and corrections are marked in yellow in the paper.

Specific Points:

1.  Line No. 474-478: The following lines should be written as

    *"Chattopadhyay et al. (2019) also suggests that LSTM performs worse than RC in some cases, and this might be related to the use of a simple variant of the LSTM architecture. This variant of LSTM was tasted and it was found that the time-varying local mean in time series would sometimes influence its performance. However further investigation is required for a deeper understanding of the same."*

**Response:** Thank you! We rewrote these lines in the revised manuscript, as the following screenshot shows:

> 474    (nRMSE=0.16) is better than reconstructing from $X_2$ to $X_1$ (nRMSE=0.20). In this case, the
>
> 475    reconstruction quality of LSTM is worse than that of RC, and the reconstruction results by LSTM
>
> 476    are consistent with the indication of the CCM index. Chattopadhyay et al. (2020) also suggests that
>
> 477    LSTM performs worse than RC in some cases, and this might be related to the use of a simple
>
> 478    variant of the LSTM architecture. This variant of LSTM was tested and it was found that the
>
> 479    time-varying local mean in time series would sometimes influence its performance. However further
>
> 480    investigation is required for a deeper understanding of the real reason. In this high-dimensional

2.  Line No. 523 - 528: The following lines can be rewritten as:

    *"However, RC and LSTM are not restrained by the Pearson correlation in this nonlinear system. When θ is altered from 0.7 to 0.3, although the Pearson correlation changed a lot, the values of CCM index stayed*

*consistently above 0.9. Throughout the alterations in θ, RC is able to produce a good quality reconstruction of X1. Fig. 9b shows that the reconstructed series by RC and LSTM always overlap with the real time series. Thus it can be inferred that the performance of both RC and LSTM is sensitive to the value of CCM index. This has been analyzed in section 4.2.2.''*

**Response:** Many thanks for your suggestion! We rewrote these lines in the revised manuscript, as the following screenshot shows:

| | |
|---|---|
| 525 | However, RC and LSTM are not restricted to the Pearson correlation in this nonlinear system. |
| 526 | When $\theta$ is altered from 0.7 to 0.3, although the Pearson correlation is changed a lot, the values of |
| 527 | the CCM index are kept consistently above 0.9. For all values of $\theta$, RC is able to equally produce a |
| 528 | good quality reconstruction of $X_1$. Fig. 9b shows that the reconstructed series through RC and LSTM |
| 529 | always overlap with the real time series. These results indicate that the performance of both RC and |
| 530 | LSTM is sensitive to the value of CCM index, which is in line with the results given in section 4.2.2. |

3. Line No. 47-48: I suggest rewriting this line as:

*"For example, chaos is a crucial property of climatic time series (Lorenz, 1963; Patil et al., 2001).''*

**Response:** Many thanks for your suggestion! We rewrote these lines in the revised manuscript, as the following screenshot shows:

| | |
|---|---|
| 47 | series. For example, chaos is a crucial property of climatic time series (Lorenz, 1963; Patil et al., |
| 48 | 2001). Thus, there is significant concern regarding the ability of machine learning algorithms to |
| 49 | reconstruct the temporal dynamics of the underlying complex systems (Pathak et al., 2017; Du et al., |

4. Line no. 48-50: The following lines can be rewritten as:

*"Thus, there is significant concern regarding the ability of machine learning algorithms to reconstruct the temporal dynamics of the underlying complex systems (Pathak et al., 2017; Du et al., 2017; Lu et al., 2018; Carroll, 2018; Watson, 2019).''*

**Response:** Thank you! We have rewritten these sentences, as the following screenshot shows:

series. For example, chaos is a crucial property of climatic time series (Lorenz, 1963; Patil et al.,

2001). Thus, there is significant concern regarding the ability of machine learning algorithms to reconstruct the temporal dynamics of the underlying complex systems (Pathak et al., 2017; Du et al.,

**Other main corrections:**

1.  In the lines 72-74, for the better presentation, we rewrote the sentences, as the following screenshot shows:

might be nonlinearly coupled. For instance, the linear cross-correlations of sea air temperature series observed in different tropical areas are overall weak, but they can be strong locally and vary with time (Ludescher et al., 2014), and such time-varying correlation is an indicator of non-linear correlation (Sugihara et al., 2012). These non-linear correlations of the sea air temperature series

2.  Line 663 and line 713, the two cited papers were from the arXiv preprint. Now their corresponding published articles are available , and we update their information in the reference, as the following screenshot shows:

Chattopadhyay, A., Hassanzadeh, P., and Subramanian, D.: Data-driven predictions of a multiscale Lorenz 96
chaotic system using machine-learning methods: reservoir computing, artificial neural network, and long
short-term memory network, Nonlin. Processes Geophys., 27, 373–389, 2020.

Kratzert F., Herrnegger M., Klotz D., Hochreiter S., Klambauer G.: NeuralHydrology – Interpreting LSTMs in
Hydrology. In: Samek W., Montavon G., Vedaldi A., Hansen L., Müller KR. (eds) Explainable AI:
Interpreting, Explaining and Visualizing Deep Learning. Lecture Notes in Computer Science, vol 11700.
Springer, Cham, 2019.

3.  We also carefully inspected the sentences, figures, figure captions, tables and formulas for avoiding any possible typos and errors. Here we did not list the changes but marked in yellow in revised paper.

We have improved the manuscript and made some changes in the manuscript. These changes will not alter the content and framework of the paper.

We appreciate the Editor's and Reviewers' work earnestly, and hope that the correction will meet with approval.

Once again, thank you very much for your comments and suggestions.

Best regards.

Yours sincerely,

Yu Huang, Lichao Yang, Zuntao Fu

[revised manuscript text omitted]